# Neural mechanisms underlying the effects of physical fatigue on effort-based choice

Patrick S. Hogan[1], Steven X. Chen[1], Wen Wen Teh[1] & Vikram S. Chib [1,2,3 ✉]

Physical fatigue crucially influences our decisions to partake in effortful action. However, there is a limited understanding of how fatigue impacts effort-based decision-making at the level of brain and behavior. We use functional magnetic resonance imaging to record markers of brain activity while human participants engage in uncertain choices for prospective physical effort, before and after bouts of exertion. Using computational modeling of choice behavior we find that fatiguing exertions cause participants to increase their subjective cost of effort, compared to a baseline/rested state. We describe a mechanism by which signals related to motor cortical state in premotor cortex influence effort value computations, instantiated by insula, thereby increasing an individual's subjective valuation of prospective physical effort while fatigued. Our findings provide a neurobiological account of how information about bodily state modulates decisions to engage in physical activity.

[1] Department of Biomedical Engineering, Johns Hopkins School of Medicine, Baltimore, MD 21205, USA. [2] Kavli Neuroscience Discovery Institute, Johns Hopkins University, Baltimore, MD 21205, USA. [3] Kennedy Krieger Institute, 707 North Broadway, Baltimore, MD 21205, USA. ✉email: vchib@jhu.edu

How fatigued we feel has a marked influence on our decisions. For example, after a particularly effortful bicycle commute, we might feel very tired and decide not to partake in our regularly scheduled post-work exercise routine. Physical fatigue can result in a diminished willingness to exert effort that might be performed in a more rested state. Previous work has established a network of brain regions, including the anterior cingulate cortex (ACC), bilateral anterior insula, and ventromedial prefrontal cortex (vmPFC) in computing the value of effortful options and making effort-based decisions[1–10]. However this work has primarily focused on valuation and decision-making while in a rested state, and did not consider how physical fatigue might influence decisions to exert effort. Consequently, there is a limited understanding of how the value of effort, and associated neural processes, are influenced by fatigue-induced changes in bodily state.

Previous studies have reported neural correlates of physical fatigue in motor regions, with motor cortex and somatosensory cortex exhibiting decreased activity following fatiguing exertion[11–13]. These results are further corroborated by transcranial magnetic stimulation studies that found motor cortical excitability decreased following fatiguing exertion[14,15]. It has been suggested that such fatigue-induced decreases in motor cortical activity are a reflection of a reduced capacity for the recruitment of motor pathways within the central nervous system. These previous works examined motor cortical physiology following fatiguing exertion and did not investigate how such motor cortical changes might influence the valuation of prospective effort and decisions to engage in physical activity.

Recently it has been proposed that feelings of fatigue may arise from inconsistencies between beliefs about the consequences of actions and actual sensory inputs and motor outputs[16–18]. With this in mind, it has been suggested that brain networks that process proprioceptive and exteroceptive signals from muscles and interoceptive signals from the internal state of the body and visceral organs, could be critical for generating feelings of effort and fatigue. These brain regions include somatosensory regions and the posterior insula[19–23]. Notably, a series of studies by Meyniel and colleagues examined how individuals performed bouts of physical exertion and choose to take rests, and found that portions of the posterior insula encoded signals that followed the time course of proprioceptive feelings associated with exertion and rest[22,24].

The findings of posterior insula encoding proprioceptive signals, and anterior insula encoding effort value[4,7,25], hint at the possibility that exteroceptive and interoceptive feelings could mediate effort values encoded by the insula. However, studies that examined neural signals at the time of exertion and rest were not designed to examine prospective valuation of effort, while studies of effort valuation have not investigated the influence of fatigue. Therefore it is not clear how brain signals related to fatigue might influence effort valuation. It should be noted that while these studies included model parameters related to fatigue[22,24,26], fatigue was not experimentally manipulated, making it difficult to infer how the state of fatigue influences effort valuation and associated brain activity. Furthermore, while recent studies have examined how physical[27] and cognitive[28] fatigue influence cognitive control during a temporal discounting task, they did not evaluate how fatigue influenced effort valuation. To our knowledge there have been no studies that have directly tested how changes in an individual's bodily state, through bouts of physical exertion, influence prospective valuation of effort and resulting decisions. Accordingly, there is a limited mechanistic understanding of how neural signals related to perceptions and sensations of fatigue influence effort valuation and decisions to exert.

Studies of motor control have also begun to examine how internal models of effort value are generated and how these representations influence decisions between potential movements and movement generation. However these studies have focused on behavior and did not examine the brain activity that encodes effort value. Furthermore, these motor control studies did not examine how physical fatigue influences effort valuation[29–31].

Here, we investigate the influence of fatigue on behavioral representations of subjective effort value, and the neural mechanisms by which fatigue interacts with the brain's valuation and decision-making circuitry. Behaviorally we hypothesize that fatigue, arising from repeated physical exertion, results in individuals having an exaggerated subjective valuation of effort that manifests as diminished risk preferences for prospective effort. Essentially, when individuals are faced with the option of exerting a certain amount of effort, versus a risky option involving either a greater amount of effort or no effort at all, they are less willing to choose the risky option while in a fatigued state. This hypothesis has its basis in previous studies of effort-based decision-making that found individuals exhibit increased sensitivity to changes in subjective effort as objective effort levels increase (i.e., risk aversion for effort)[6,7,9]. Neurally we hypothesize that decisions about prospective effort exertion have their basis in a value signal encoded in the ACC and insula. This hypothesis is informed by a number of neuroimaging studies that found a correlation between activity in these brain regions and behavioral measures of effort value[1–8,25]. Given recent studies which found that insula encodes feelings of effort during bouts of exertion and rest[22], we predict that the insula is sensitive to changes in effort value as a function of fatigue. Together these hypotheses form a neurobehavioral account of fatigue, which recruits regions of the brain responsible for effort valuation and motor exertion, to inform decisions about prospective effort while in an altered bodily state arising from prolonged exertion.

## Results

**Behavioral evidence of subjective effort value**. To study how decisions about effort are influenced by physical fatigue we scanned participants' brains with functional magnetic resonance imaging (fMRI) while they made risky choices about prospective effort before, and interspersed with, bouts of fatiguing exertion. The first session of choices was used to characterize participant-specific subjective valuation of effort in a baseline, rested state (Fig. 1a; effort-based choice). After this baseline choice phase participants performed a block of exertion trials until exhaustion (Fig. 1a; exertion trial). Participants then alternated between blocks of effort choice trials and exertion trials (Fig. 1b). These blocks of exertion trials were meant to maintain participants in a fatigued state throughout choice, and minimize the possibility of physical recovery during choice. All of the choices were for prospective effort and at the end of the experiment 10 trials were randomly selected to be played out so that participants' decisions had actual consequences. It should be noted that all participants were right-handed and made choices with a button box in the left hand, and hand-grip exertions were performed with the right hand. This allowed us to laterally dissociate motoric brain signals related to choice selection and exertion.

Our effort choice paradigm exploits the theoretical equivalence between risk preferences and subjective valuation in order to measure subjective valuation via the presentation of risky choices, a widely accepted practice in economics and decision neuroscience[32,33]. Moreover, choice prospects only involved varying amounts of physical effort (i.e., no prospective rewards were involved), which allowed us to examine how fundamental processes related to effort valuation were influenced by fatigue-

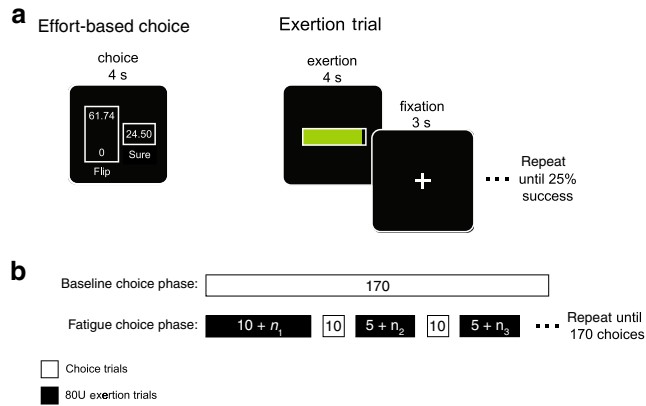

**Fig. 1 Experimental design. a** During effort-based choice trials participants were presented with a series of choices between a risky option that could result in either a large level of exertion or no exertion with equal probability (Flip), or exerting a lower level of effort with certainty (Sure). The effort amounts were presented on a 0 to 100 scale, which participants were trained on during an association phase prior to choice. An effort level of zero corresponded to no exertion and 100 to 80% of a participant's maximum exertion. To study the effects of fatigue on effort-based decision-making, blocks of exertion trials were interspersed with blocks of effort-based choice trials. During exertion trials, participants exerted grip force on a hand dynamometer to fill a horizontal bar, where the full bar corresponded to an effort level of 80. **b** Experiment schedule. The experiment was divided into baseline and fatigue choice phases, which were both scanned with fMRI. The baseline choice phase consisted of 170 effort-based choices presented in a randomized order. Following the baseline choice phase, participants performed the fatigue choice phase of the experiment, in which they underwent repeated exertion trials (indicated in black) to bring them into a fatigued state. The fatigue choice phase began with an initial exertion block, comprised of a minimum of ten repetitions of exertion, followed by alternating choice (shown in white) and exertion blocks. Exertion trials in a block were repeated until 75% of trials within an exertion block were counted as failures ($n_i$ indicates the variable additional number of repetitions before participants reached the failure criteria). Each choice block consisted of 10 effort-based choices randomly sampled from the same set used in the baseline choice phase. Subsequent exertion blocks were identical to the first, differing only in that they were comprised of a minimum of 5 trials. Completion of the fatigue choice phase consisted of 17 back-to-back exertion and choice blocks.

induced changes in bodily state, separate from effects associated with reward valuation or effort/reward trade-offs. We have previously used this choice set to extract computational parameters that capture an individual's subjective valuation of physical effort in a state of rest[9].

Consistent with our previous findings[9], prior to the fatiguing exertions, the majority of participants exhibited $\rho_{baseline} > 1$, indicating increasing sensitivity to changes in subjective effort cost as objective effort level increases (mean $\rho_{baseline} = 1.39$ (SD = 0.56); two-tailed one-sample $t$-test against the null hypothesis that the mean of $\rho_{baseline} = 1$: $t_{19} = 3.17$, $p = 0.005$). $\rho_{baseline} > 1$ corresponds to participants being risk averse for effort. As in our previous work, there was considerable individual variability in participants' $\rho_{baseline}$, reflecting individual differences in baseline subjective preferences for effort.

**Repeated physical exertion results in fatigue.** Behavioral results from exertion trials indicated that participants' ability to successfully achieve the performance criteria diminished between the first and second exertion blocks (Fig. 2a; average reduction in exertion repetitions between the first and second block: 9.26, two-

tailed paired-sample $t_{18} = 4.69$, $p = 0.0002$; one participant was excluded from this analysis, because the number of exertion trials in their second block was <3 SD above the mean, but is included in all subsequent analyses), and that participants consistently performed the same number of exertion trial repetitions (~5 trials; the required minimum) after the second fatigue block. These data show that it took ~15 trials for participants to reach a level of fatigue that caused them to meet the failure threshold (i.e., 75% of trials within the block were failed exertions). Subsequent exertion blocks lasted for ~5 repetitions, indicating that participants remained in a fatigued state and met the failure threshold after fewer trials than in the initial exertion block.

To provide a more continuous metric of motor performance we analyzed participants' mean exertion (in terms of effort level) within exertion blocks. We found that mean exertion decreased between the initial and final trials in the first exertion block (Fig. 2b; average reduction in mean exertion force: 10.14 effort units; two-tailed paired-sample $t_{19} = 3.36$, $p = 0.003$), suggesting that repetitive exertion rapidly reduced the capacity for effortful exertion. Furthermore, participants' mean exertion in subsequent blocks never recovered to its level in the first exertion block (first five trials), and all subsequent blocks were significantly reduced compared to the initial exertion bock (two-tailed paired-sample $t_{19} > 3.60$, $p < 0.05$ for all exertion blocks after the first) (see Supplementary Fig. 1 for all participants' mean exertion data). It should be noted that mean exertion levels never diminished to zero, implying that on average participants were always trying to exert effort and their capacity was reduced over the course of repeated exertion blocks. Together these results illustrate that participants' capacity for exertion was decreased by repeated exertion, and that the exertion blocks interspersed throughout the fatigue choice phase maintained participants in a fatigued state.

**Evidence of fatigue-induced changes in effort value.** Analyzing choice trials in the fatigue choice phase, we found that exertion-induced fatigue resulted in participants exhibiting more risk averse choice behavior for effort compared to the baseline choice phase; participants were less willing to take the chance of having to exert large amounts of effort, indicating an increased marginal sensitivity to effort cost (Fig. 2c shows group-average cost functions for effort for the baseline and fatigue choice phases). This manifested as a significant increase in $\rho_{fatigue}$ compared to $\rho_{baseline}$ (Fig. 2d; mean $\Delta\rho = 0.49$ (SD = 0.86), two-tailed paired-sample $t_{19} = 2.54$, $p = 0.02$), corresponding to an increase in marginal costs for effort while in a fatigued state. A model-free analysis of the change in choice behavior (proportion of risky options accepted) corroborated this finding of decreased risk-taking while fatigued, and illustrated that this result was not simply a byproduct of our model-based analysis (Supplementary Fig. 2). We also examined how changes in choice behavior varied over the course of the fatigue choice phase and found that the change in risk-taking behavior remained relatively constant over the course of the choice blocks (Supplementary Fig. 3). It is important to mention that a subset of participants ($n = 5$) exhibited a decrease in $\rho_{fatigue}$ compared to $\rho_{baseline}$, corresponding to a decrease in the marginal cost of effort while in a fatigued state. Such behavioral change corresponds to participants exhibiting an increased risk tolerance in order to avoid effort exertion—these individuals are willing to take more risks for the chance of obtaining the zero effort component of the risky prospect. Notably the temperature parameter $\tau$, which represents the stochasticity of an individual's choices, was not significantly different between the baseline and fatigue choice phases ($\Delta\tau = -0.08$ (SD = 0.25), two-tailed paired-sample $t_{19} = -1.42$, $p = 0.17$). In addition, we re-estimated subjectivity parameter $\rho_{fatigue}$ while holding $\tau_{fatigue}$ constant and

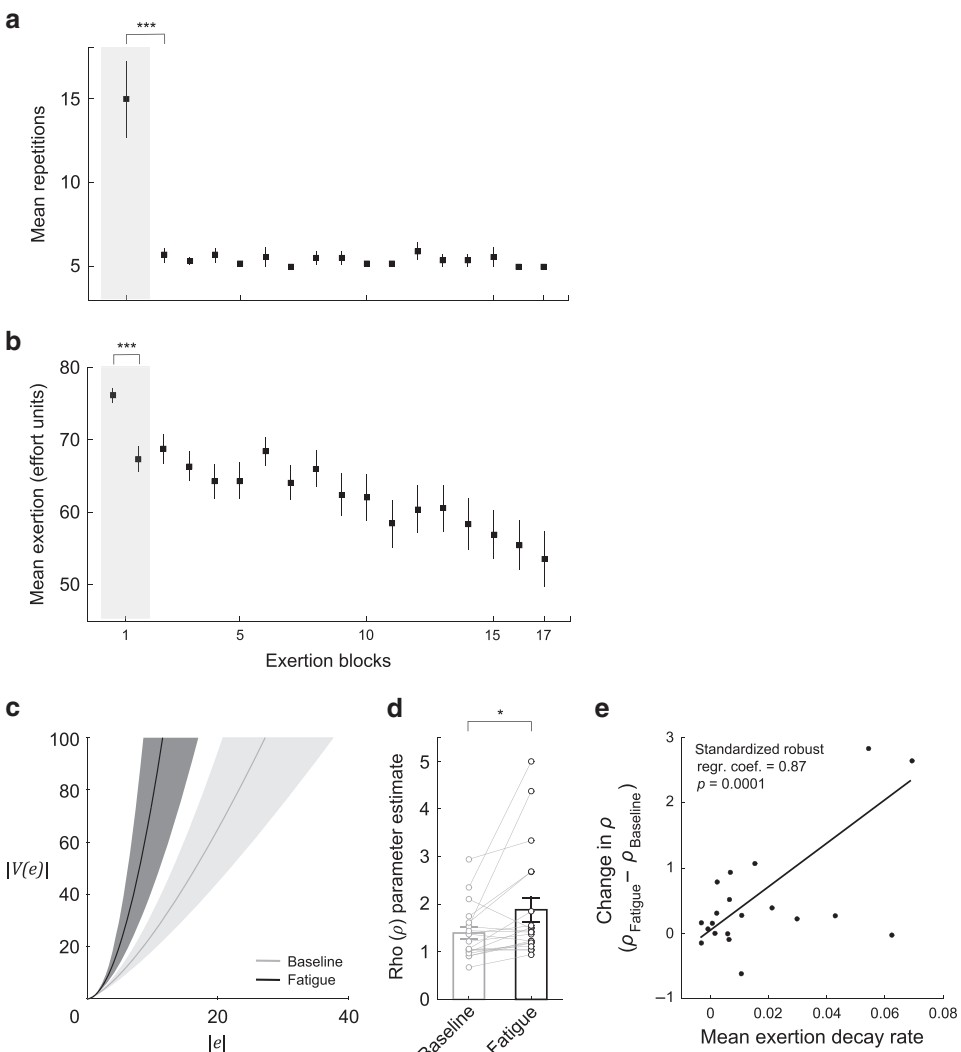

**Fig. 2 Behavioral results ($n = 20$). a** Exertion trial performance during the fatigue choice phase. Performance is represented as the mean number of repetitions until the exertion block termination criterion was met, across participants. The initial exertion block is indicated in the shaded gray region. Participants' ability to successfully achieve the performance criteria diminished across exertion blocks, and participants consistently performed the same number of exertion trial repetitions after the second exertion block. Error bars indicate SEM. Paired $t$-test (two-tailed): ***$p = 0.0002$. **b** Mean exertion during the fatigue choice phase. Mean exertion is presented in units of effort. For the initial exertion block (shaded gray), the mean exertion for the first and last five trials is shown. For subsequent exertion blocks the mean exertion over the full exertion block is shown. Mean exertion decreased between the initial and final trials in the first exertion block and mean exertion never recovered to its initial level during the fatigue choice phase. Error bars indicate SEM. Paired $t$-test (two-tailed): ***$p = 0.003$. **c** The function used to model the subjective cost of effort. This function has the form $V(x) = -(-x)^\rho$. Effort cost functions using mean values of the $\rho$ estimates are indicated by the solid lines (baseline: light gray; fatigue: dark gray), with SEM indicated by the shaded regions. Undergoing fatiguing exertions increases the marginal cost of effort. In order to better illustrate the cost functions, the $x$- and $y$-axes shown are not to the same scale. **d** The effort subjectivity parameter ($\rho$) significantly increased between the baseline and fatigue choice phases. This indicates that when compared to baseline, exertion-induced fatigue makes the subjective value of effort even more costly to participants. Error bars indicates SEM. Paired $t$-test (two-tailed): *$p = 0.02$. **e** Between-participant regression considering the relationship between fatigue-induced decay in mean exertion, and changes in effort subjectivity parameters between the baseline and fatigue choice phases. This relationship illustrates that individuals who experience a steeper reduction in motor performance (larger decay parameter) exhibit greater increases in the subjective cost of effort, between the baseline and fatigue choice phases.

equal to its baseline value, and still found a significant increase in $\rho_{\text{fatigue}}$ compared to $\rho_{\text{baseline}}$ (two-tailed paired sample $t_{19} = 3.22$, $p = 0.005$). These results indicate that fatigue had an impact on subjective effort valuation that was separate from variability in choice behavior, between the two phases.

If repeated exertion trials play a role in the subjective valuation of effort in the fatigue choice phase, we would expect that fatigue-induced decreases in exertion should modulate participants' changes in subjective effort parameters between the baseline and fatigue choice phases. To test this, we fit an independent

parameter that quantified participants' mean exertion decay rate over the first ten exertion trials (Supplementary Fig. 4, see "Methods" for details). This parameter captures how quickly a participant's exertion diminished over the course of the initial exertion block—larger metrics correspond to a steeper drop-off in performance (representative of the fatiguing effect of exertion). We found that the degree of participants' changes in effort subjectivity were correlated with mean exertion decay rate during the initial exertion block (standardized robust regression coefficient = 0.87, $p < 0.001$; Fig. 2e), suggesting that fatigue-

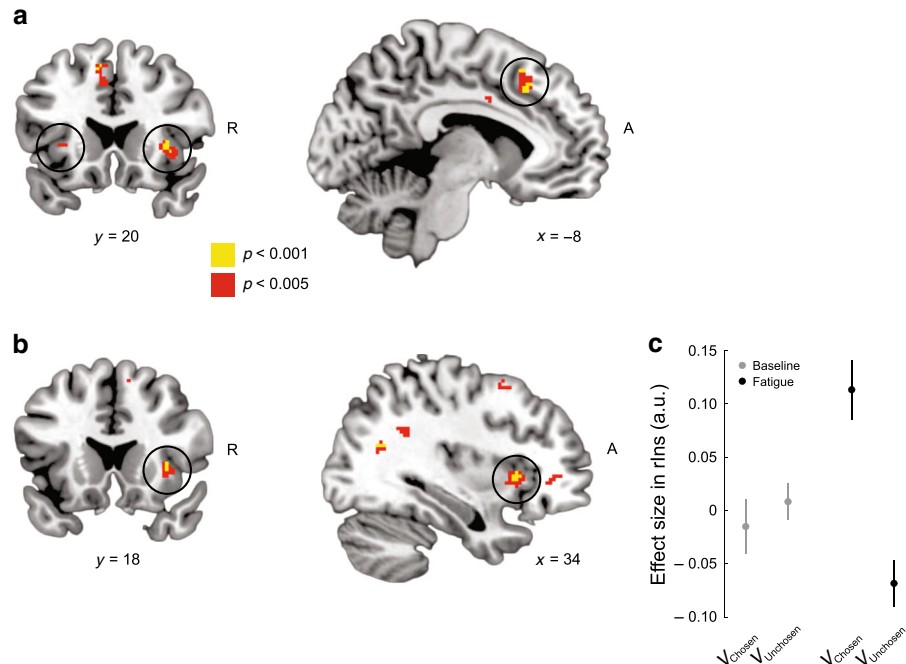

**Fig. 3 Neural signatures of chosen effort value ($n = 20$). a** Overall effort value encoding. Whole brain results thresholded at voxelwise $p < 0.005$ (red) and $p < 0.001$ (yellow). Activity in bilateral insula (rIns: peak = [34, 20, 2]; lIns: peak = [−32, 24, 2]; small volume corrected $p < 0.05$ in a priori ROI) reflects the difference in chosen and unchosen effort value at the time of choice, across both the baseline and fatigue choice phases. Activity was also observed in ACC (MNI coordinate: peak = [−8, 24, 42]), although it does not survive small volume correction in our a priori ACC ROI. **b** Effort value encoding in rIns increases with fatigue. Activity in rIns (peak = [34, 18, 2]; small volume corrected $p < 0.05$ in a priori ROI) reflects the difference of chosen and unchosen effort value between the baseline and fatigue choice phases. **c** Effects in rIns (5 mm sphere centered at [34, 18, 2]) for chosen and unchosen effort value in the baseline and fatigue choice phases are shown. This plot was not used for statistical inference (which was performed in the SPM framework) and is shown solely to illustrate the pattern of the BOLD signal. Error bars indicate SEM.

induced changes in subjective effort value were significantly influenced by the diminished physical capacity imparted by the initial exertion block.

To exclude the possibility that changes in choice behavior between the choice phases were simply the result of mere exposure to the effort choices, and not an effect of exertion-induced fatigue, we performed a control experiment in which a separate group of participants performed two phases of effort choices where no fatiguing exertion blocks were included (see "Methods", Control Experiment 1). There was no significant difference in choice behavior in the control group, while the main experimental group exhibited a significant change (Supplementary Fig. 5). These results suggest that the observed changes in behavior between the baseline and fatigue choice phases in the main experiment were specifically related to fatigue induced by the exertion blocks, and not repeated exposure to the effort choices.

It is important to note that our use of the effort utility function $V(x) = -(-x)^\rho$ was motivated by previous work in which we found this formulation best described effort-based decision-making in a rested state, in the absence of reward[9]. Previous studies have proposed a variety of effort utility models[6,7] and we tested if these models better described fatigue-induced changes in subjective effort valuation (details in Supplementary Experimental Methods). A formal model comparison showed that $V(x) = -(-x)^\rho$ provided the best description of participants' decisions across both the baseline and fatigue choice phases (Supplementary Fig. 6).

**Neural encoding of effort value.** We found that BOLD signal in a network of brain regions, including dorsal anterior cingulate and bilateral insula, was modulated by the difference between chosen and unchosen effort value across both the baseline and fatigue choice phases (Fig. 3a; Supplementary Table 1). Activity in these brain regions exhibited increased activity for the chosen compared to the unchosen option. This finding is consistent with a multitude of studies of effort-based decision-making that implicated these brain regions as part of an effort valuation network[1–8,22].

To test for regions of the brain that were sensitive to changes in effort value induced by fatigue, we contrasted the difference between the chosen and unchosen options between the baseline and fatigue choice phases. We found a region of the right anterior insula (rIns) was modulated by effort value during the fatigue choice phase (Fig. 3b; Supplementary Table 2), and appeared to be insensitive to chosen and unchosen effort value in the baseline/rested state (Fig. 3c). This suggests that activity in rIns is sensitive to changes in effort value resulting from fatigue, which aligns with previous studies of effortful exertion that have suggested this brain region encodes representations of bodily state that influence decisions regarding bouts of exertion and rest[22,24].

Since regions of insula have also been implicated in representations of risk preferences, it is possible that the signals we observed could be driven by differences in risk preferences between choices in the baseline and fatigue choice phases, and not effort value per se. To test this possibility we examined the correlation between individual differences in risk preferences between the fatigue and baseline choice phases and rIns effects shown in Fig. 3b ($r = 0.03$, $p = 0.91$). We failed to find a significant correlation between brain activity in rIns and fatigue-induced changes in risk preferences, suggesting that these brain signals best represented fatigue-induced effort valuations and were not simply the byproduct of fatigue-induced changes in risk preferences.

**Fatigue-induced changes in motoric brain activity**. We next reasoned that in order to make informed decisions about effort, while in a fatigued state, the brain should encode information about the state of the motor system at the time of choice. To test this idea we used the aforementioned general linear model (GLM) to examine differences in brain activity between the baseline and fatigue choice phases, irrespective of the effort values under consideration. Consistent with this idea, we found that regions of left premotor (PM) and primary motor cortices exhibited significant decreases in activity at the time of choice while individuals were in the fatigue choice phase, compared to baseline (Fig. 4a, b; Supplementary Table 3). These results align with previous studies of physical fatigue which have shown that motor regions exhibit diminished activity and cortical excitability following repeated exertion[11–15], and go further by illustrating that such fatigue-induced reductions in motor activity are observed even when prospective decisions are being made about effort.

Considering our prediction that motor state would be an integral factor in representations of fatigue-induced changes in subjective value, we tested if changes in subjective effort preferences ($\rho_{fatigue}-\rho_{baseline}$) were indexed by fatigue-induced changes in activity in PM. It has been suggested that fatigue might arise as the result of discrepancies between expectations about the consequences of actions and actual sensory and motor outputs[16–18]. In the context of our experiment, if an individual's motor system does not appropriately adjust its resting state in response to repeated exertions, one might feel that effort is particularly costly because of the discrepancy between the motor production that one believes they can achieve and their actual motor capacity following fatigue. In contrast, it is possible that fatigue arises from an accurate representation of an individual's bodily state and that changes in subjective preferences are simply a reflection of the altered motor cortical state that occurs following fatiguing physical exertion[11–15]. Using each participant's fatigue-induced change in $\rho$ ($\rho_{fatigue}-\rho_{baseline}$) as a second-level covariate in our imaging model, we found that those individuals with greater fatigue-induced changes in subjective effort parameters exhibited less change in PM activity (Fig. 4c, d; Supplementary Table 4). These findings are consistent with the idea that a miscalibration of motor regions is related to fatigue-induced changes in subjective effort preferences—those participants that find effort to be particularly costly following fatigue may be those who do not modify their motor cortical activity to accommodate the reduced motor capacity that results from repeated exertion.

To further explore the role of PM in fatigue-induced changes in subjective effort value, we performed an analysis to test the hypothesis that PM activity has an influence on the relationship between fatigue-induced reductions in motor performance during exertion trials and changes in subjective effort valuation (illustrated in Fig. 2e). To test this hypothesis we used a moderation analysis, a form of linear modeling in which correlations observed in the data are explained assuming that a specific set of causal influences exists among the variables[34]. This model revealed that the degree of PM BOLD deactivation (in the fatigue choice phase compared to baseline) had a significant moderating influence on the relationship between fatigue-induced decreases in performance (during exertion trials) and the increases in subjective valuation of effort (during choice) (Fig. 4e, f). Those participants that exhibited less fatigue-induced deactivation in PM activity had a stronger relationship between motor performance decay and changes in subjective valuation of effort, whereas individuals who experienced greater fatigue-induced BOLD deactivation in PM showed a weaker relationship.

We also considered an alternative causal model in which PM deactivations mediated the relationship between fatigue-induced

motor performance decays and changes in subjective valuation of effort. However, there was not a significant correlation between fatigue-induced motor performance decays and PM deactivations ($r = -0.20$, $p = 0.40$), which precluded the performance of a formal mediation analysis[35]. These findings lend further support to the specificity of PM deactivations moderating the relationship between fatigue-induced changes in motor performance and changes in subjective effort value.

**Different levels of exertion modulate effort value**. While our main experiment was designed to examine how changes in fatigue influenced effort valuation, we did not directly poll participants' subjective ratings of effort to confirm that the fatiguing exertions did in fact influence perceptions of fatigue. Furthermore, the experiment did not collect any measures of muscle activation, so there was no data to show that exertions resulted in physiological signatures of physical fatigue. Moreover, the main experiment tested only one level of exertion (80U), making it unclear whether different levels of exertion modulate subjective effort preferences.

To address these limitations, we designed a comprehensive control experiment in which participants exerted different levels of effort, interspersed with blocks of effort choices, and were queried on their feelings of fatigue throughout the experiment (Fig. 5a, b). We also monitored muscle activity with electromyography (EMG) to confirm that repeated exertions lead to physiological signatures of muscle fatigue. This experimental design allowed us to assess how self-reported measures of fatigue, changes in muscle physiology, and effort preferences were influenced by experimentally varying levels of exertion.

Participants performed a relatively consistent number of repetitions (~5 trials) across effort exertion blocks (Fig. 6a) (similarly to Fig. 2a, one participant was excluded from this figure because the number of exertion trials in their sixth block—the first block of the 60U section—was greater than 3 SD above the group mean, but is included in all subsequent analyses). The first exertion block of the 60U exertion session had an increased number of repetitions because participants were not cued when the exertion blocks switched from low to high effort exertions. The first block of exertion trials in the new section were intentionally unanticipated, resulting in increased variability in performance. However, participants quickly understood the exertion criteria of the section and their behavior became more stereotypical. Participants successfully reached the target levels of exertion in the low and high exertion blocks, and maintained consistent levels of mean exertion matching the cued effort levels of a given effort block (Fig. 6b). Together these results indicate that the modified success criteria resulted in consistent, and experimentally controlled, amounts of exertion for both the low and high effort sections of the modified fatigue choice phase.

When initially performing low effort exertions, participants did not self-report as feeling fatigued, indicating that the mere presence of exertion trials did not induce feelings of fatigue (Fig. 6c; average self-reported rating during the first low effort (10U) exertion section: 1.81 (SD = 0.66); results of a two-tailed one-sample $t$-test against the null hypothesis that the average self-reported rating in the first low effort (10U) exertion section was 3 (corresponding to "Unsure"): $t_{16} = = 7.48$, $p < 0.001$). When performing repeated high effort exertions, participants' fatigue ratings increased (mean increase in within-block self-reported rating between the first low effort (10U) and high effort (60U) sections: 1.14 (SD = 0.71), two-tailed paired-sample t-test comparing the average ratings between first low effort and high effort sections: $t_{16} = 6.58$, $p < 0.001$). Furthermore, an increase in fatigue ratings persisted even after the two-minute rest period, as well as during the second low effort exertion blocks (comparing the

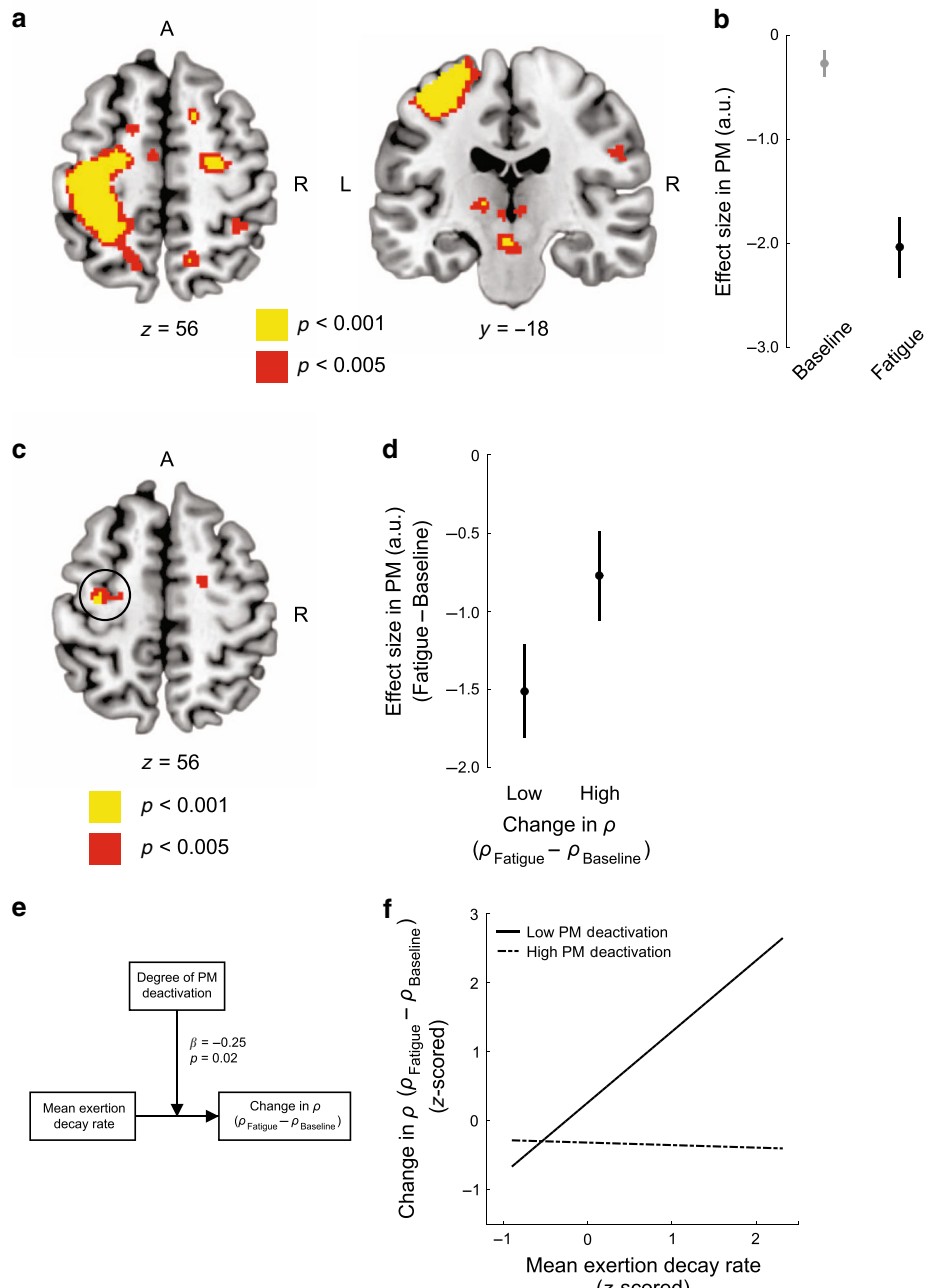

**Fig. 4 Neural signatures of exertion-induced fatigue ($n = 20$). a** Exertion-induced changes in brain activity. Whole brain results thresholded at voxelwise $p < 0.005$ (red) and $p < 0.001$ (yellow). Activity in primary motor cortex (peak = [−38, −30, 52]) and PM (peak = [−36, −18, 54]) was reduced in the fatigue condition, relative to baseline, at the time of choice. These activations are significant small-volume corrected $p < 0.05$ within our a priori premotor ROI. **b** Illustration of the PM (5 mm sphere centered at [−36, −18, 54]) effects reported in (**a**). This plot was not used for statistical inference, which was performed in the SPM framework. Error bars indicate SEM. **c** Bold signal in PM is positively modulated by fatigue-induced increases in $\rho_{\text{fatigue}}$ relative to $\rho_{\text{baseline}}$. Between-participant regression analysis entering the change in effort subjectivity $\rho_{\text{fatigue}} - \rho_{\text{baseline}}$ as a covariate for the decreasing activity between the fatigue and baseline choice phases in PM (peak = [−32, −14, 58]; small volume corrected $p < 0.05$ in a priori ROI). **d** Relationship between activity in PM (5 mm sphere centered at [−32, −14, 58]) and the degree of fatigue-induced change in effort subjectivity parameters. The low/high groups are defined by the median split of the change in $\rho$ metric. Individuals who exhibit greater fatigue-induced increases in the subjective effort parameters have less PM deactivation. This plot is shown solely to illustrate the pattern of the BOLD signal and statistical inference was performed in the SPM framework. Error bars indicate SEM. **e** Moderation analysis. We tested the extent to which changes in PM activity moderates the relationship between exertion-induced decreases in effort capacity (indexed by mean exertion decay rate parameters) and changes in subjective effort value (indexed by $\rho_{\text{fatigue}} - \rho_{\text{baseline}}$). **f** Illustrative plot of the moderating influence of PM deactivation on the relationship between fatigue-induced reduction in motor performance and increases in the subjective cost of effort. The lines represent regressions between mean exertion decay and fatigue-induced changes in $\rho$, separated by median split PM activity. The less PM deactivation between the baseline and fatigue phases, the stronger the relationship between the fatigue-induced decay in motor performance and fatigue-induced increases in the subjective cost of effort.

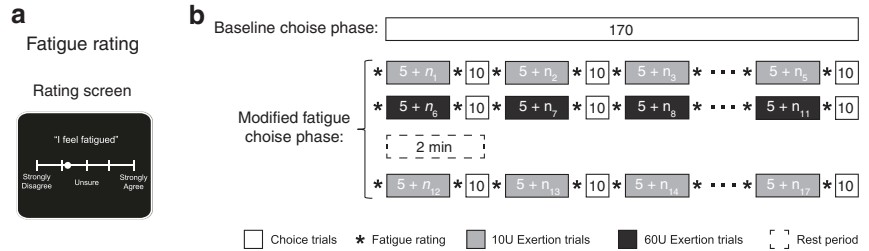

**Fig. 5 Experimental Design of Control Experiment 2. a** At the beginning and end of each exertion block, participants were queried on their level of agreement ("Strongly disagree", "Somewhat disagree", "Unsure", "Somewhat agree", "Strongly agree") with the statement "I feel fatigued". Participants were free to select anywhere on the scale (not just the indicated points), and there was no time constraint for them to select their rating. **b** Experiment schedule. The control experiment was divided into baseline and modified fatigue choice phases. The baseline choice phase consisted of the same 170 effort-based choices used in the main experiment. Following the baseline choice phase, participants performed the modified fatigue choice phase, in which they alternated between exertion trials and choice trials. The modified fatigue choice phase began with participants alternating between low effort exertions (10U; light gray) and blocks of choice trials (white). Following this sequence of exertion/choice, participants alternated between high effort exertions (60U; dark gray) and blocks of choice trials. Participants were then given a two minute period of rest in which they did not make any exertions or choice, and then alternated between low effort exertions and blocks of choice trials. Each exertion block lasted until participants exerted five successful trials by achieving the target level ($n_i$ indicated the variable additional number of unsuccessful repetitions in an exertion block). At the beginning and end of each exertion blocks, participants were queried as to their feelings of fatigue as shown in **a** (indicated by asterisks).

average within-block self-reported fatigue ratings between the first and second sections of low effort exertions with a two-tailed paired-sample $t$-test: $t_{16} = 5.69$, $p < 0.001$). These data show that after repeated high effort exertions, participants' self-reported feelings of fatigue persist even after a period of rest and when performing subsequent low-effort exertions.

To assess if muscle activity reflected physical fatigue across different exertion sections in the modified fatigue choice phase, we compared the mean frequency of the power spectral density of the EMG signal between the average of successful trials in the first and second low effort exertion sections. Muscle fatigue is associated with a down-shift in the mean frequency of the power spectrum of the EMG signal (when exerting the same level of effort)[36–38]. We found a significant decrease in the mean frequency of the power spectrum between the first and second low effort (10U) exertion sections (Fig. 6d; average change in mean frequency between the first and second sections of low effort trials: −4.39 Hz; two-tailed paired-sample $t$-test: $t_{16} = -2.25$, $p = 0.04$). We also found a significant decrease in mean frequency across trials in the 60U exertion section (Supplementary Fig. 7). These EMG results are consistent with the idea that repeated effortful exertions result in muscle fatigue.

To examine how effort preferences were modulated by effort level we used a hierarchal Bayesian approach to estimate subjective effort parameters ($\rho$, $\tau$) from choices in the first low effort, high effort, and second low effort sections of the experiment, providing insight into the differences between the preferences in these sections. We found that there was a significant increase in subjective effort parameters between choices in the first low effort and high effort sections (95% highest-density interval for the Bayesian posteriors of $\rho_{60U}-\rho_{10U,1}$ excludes zero; Fig. 6e), reflecting an increase in the marginal cost of effort with increasing levels of exertion. These results show that effort parameters are modulated by the level of effortful exertion and are in concert with participants' increased self-report ratings of fatigue between the first low effort section and the high effort section. Interestingly, this shift of effort preferences persisted in the second low effort section of the experiment (95% highest-density interval for the Bayesian posteriors of $\rho_{10U,2}-\rho_{10U,1}$ excludes zero), which aligns with our observation that participants' self-reported ratings of fatigue remained elevated in the second low effort section of the experiment, despite a period of rest and a decrease in the target level of exertion. Together these findings illustrate that increasing levels of fatiguing exertion result

in increased marginal utility of effort, fatigue-induced changes in muscle physiology, and increased self-report ratings of fatigue.

It is possible that changes in effort preference could be a byproduct of the experience of exertion, and not actually related to physical fatigue per se. We performed an additional analysis to directly test if changes in effort preferences were related to the level of effortful exertion and associated physical fatigue, or if changes were simply the result of the experience of repeated exertions (regardless of the level exerted). For this analysis we compared choice behavior during the first 25 fatiguing exertions between the main experimental group (first 3 exertion and choice blocks) and the control experimental group (first 5 exertion and choice blocks). Critically, the main difference between these portions of the experiments was the level of effortful exertion. In the main experiment participants exerted effort at a level of 80U, and in the control experiment participants exerted 10U. Despite performing the same number of exertion trials in each group (Fig. 7a), participants exerted significantly more effort in the high effort group (main experimental group) compared to the low effort group (control experimental group) ($t_{34} = 38.83$, $p < 0.001$; Fig. 7b). It is important to note that the choice trials used in this analysis were not specifically designed with this exertion control in mind, and there were fewer trials available for this control analysis (30–50 trials) than in the full main and control experiments. The limited number of trials make it difficult to obtain reliable parameter estimates as in the previous analyses of choice behavior. Instead, we used a model-free choice similarity metric (see "Methods" for details) to compare participants' choices between the baseline state and during exertion. We found a significant change in participants' choice behavior when comparing the high effort and low effort groups ($t_{34} = 2.60$, $p = 0.013$; Fig. 7c). Consistent with our previous results, participants in the high effort group became more risk averse for effort, while those in the low effort group did not exhibit a significant change in choice behavior. These results provide direct evidence that changes in choice preferences are specifically related to fatigue, induced through high intensity physical exertion, and are not simply a result of the experience of repeated exertion trials.

## Discussion
Here we show that fatigue inflates the subjective value of effort (making it more costly), that decisions about effort while fatigued emerge from value signals in rIns, and that information about

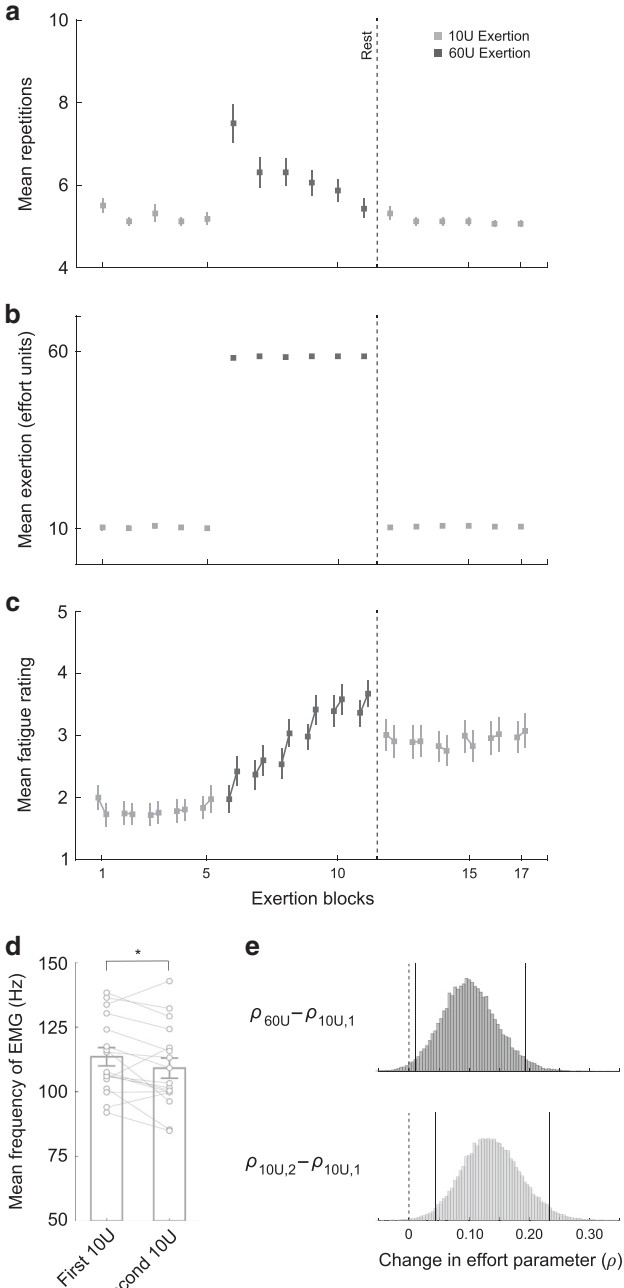

**Fig. 6 Results from Control Experiment 2 (Control Experiment 2; n = 17).**
**a** Exertion trial performance during the modified fatigue choice phase. Performance is represented as the mean repetitions required to successfully perform five exertions (successful trials need not be consecutive); higher values indicate more exertion trials were required in order to successfully exert at the target effort five times. Error bars indicate SEM. **b** Mean exertion during the modified fatigue choice phase. Mean exertion is presented in units of effort. Error bars indicate SEM. **c** Self-reported measures of participants' state of fatigue over the course of the modified fatigued choice phase. The plot shows mean self-reported ratings of fatigue at the beginning and end of the 17 exertion blocks. Self-reported fatigue persisted when the exertion task returned to low effort trials after a two minute rest. Error bars indicate SEM. **d** Mean frequency of the power spectrum of EMGs between first and second low effort sections of the modified fatigue choice phase. The plot shows mean frequency measures from EMGs for all successful exertions averaged within the first and second low effort (10U) sections. Mean frequency of the EMG power spectrum decreased between first and second low effort sections (average change in mean frequency: −4.39 Hz; two-tailed paired-sample *t*-test: $t_{16} = -2.25$, $p = 0.04$). Error bars indicate SEM. *$p = 0.04$. **e** Histograms of the parameter-space posteriors from the hierarchical Bayesian model estimating changes in effort subjectivity parameters ($\Delta\rho$) between the different sections of the modified fatigue choice phase. The population exhibited an increase in effort subjectivity parameters ($\rho$) during choices intermixed with high effort exertions, as compared to the first session of 10U low effort exertions (60U; $\rho_{60U}$–$\rho_{10U,1}$; dark gray). This shift in effort subjectivity remained even after returning to low effort exertions (10U; $\rho_{10U,2}$–$\rho_{10U,1}$; light gray). The solid black lines indicate the bounds of the 95% highest-density interval for each distribution, both of which exclude 0 (indicated with the dashed black line).

In order to make informed decisions about exertion, it is essential to have an idea of one's own physiological state (e.g., capacity for descending motor drive, motor unit recruitment, etc.). A number of neuroimaging and TMS studies have found that repeated physical exertion results in deactivations of motor and premotor brain regions, which have been suggested to reflect the reduced capacity for recruitment of motor pathways in the central nervous system[11–15]. However, it is unknown how such fatigue-induced changes in motor cortical activity are related to decisions about exertion. Here we show that at the time of choice, even when exertion is not being performed, neural deactivations in contralateral PM are related to fatigue-induced changes in subjective preferences for effort. Specifically, those individuals that exhibit the greatest fatigue-induced increases in subjective effort value exhibit the least fatigue-induced deactivation in PM BOLD activity. These results are consistent with the idea that fatigue-induced inflations of effort may be associated with dys-homeostatic representations in PM[16–18]; those individuals who do not reduce their motor cortical activity following fatiguing exertion find prospective effort to be more costly. Effort might feel particularly costly to these individuals because their motor system continues to recruit the same level of descending drive as in a baseline/rested state, in spite of their muscles being fatigued and physiologically incapable of generating the same level of motor output. This idea aligns with recent theoretical accounts of fatigue that suggest discrepancies between perceptions of ability and actual sensorimotor capacity may give rise to feelings of fatigue[16–18].

Notably, an important component of such a dyshomeostatic account of fatigue is the ability to sense one's bodily state[16]. There has been a great deal of work showing that rIns is integral in computations of interoceptive sense[20,21,23]. Across all effort choice conditions, we found a network of brain activity (including

motor cortical state in PM influences the subjective assessment of effort value. Our neural findings are consistent with an established body of work showing effort value signals are coded by a network of brain regions including the rIns[1–4,6–8], and that activity in this brain region follows the time course of feelings associated with exertion and rest[22,24]. However, previous studies investigated prospective effort valuation in a rested state and did not examine how fatigue-induced changes in bodily state influence the valuation of prospective effort and decisions to exert. Our results go beyond these studies by computationally modeling fatigue-induced changes in the subjective valuation of effort, and identifying a network of brain activity that subserves these changes. In so doing, we demonstrate a mechanism by which information about PM state influences effort-based decision-making while fatigued. Our results suggest that fatigue-induced changes in PM influence fatigue-induced changes in subjective effort valuation.

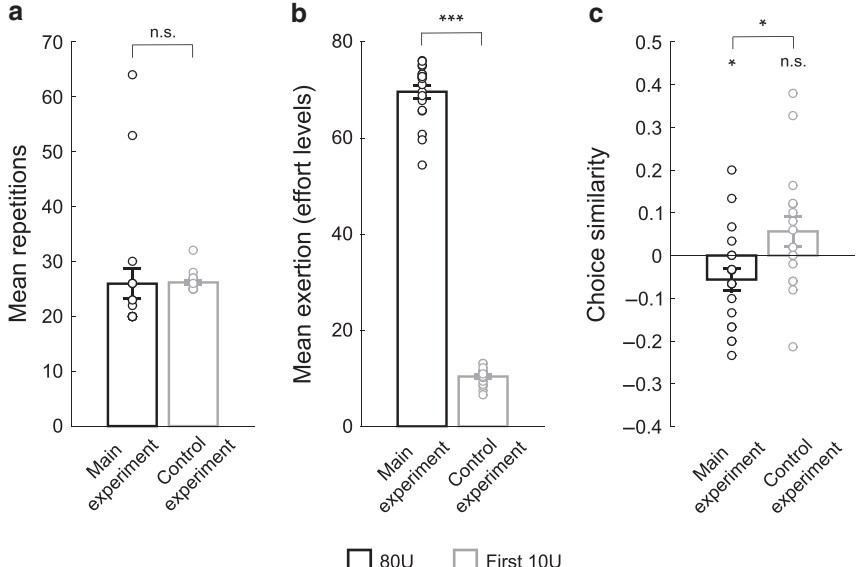

**Fig. 7 Comparison of choice preferences during high and low exertion efforts, controlling for the number of exertion repetitions (participants from the Main Experiment and Control Experiment 2; $n = 37$). a** We compared behavior during the first 25 fatiguing exertions between the main experimental group (first 3 exertion and choice blocks) and the control experimental group (first 5 exertion and choice blocks). In the main experiment participants exerted at a level of 80U, and in the control experiment participants exerted 10U. Participants did not perform a significantly different number of exertion trials between these experimental epochs (Paired $t$-test (two-tailed): $t_{34} = 0.08$, $p = 0.94$). Error bars indicate SEM. **b** Participants exerted significantly more effort in the main experimental group compared to the control experimental group (Unpaired $t$-test (two-tailed) comparing the mean effort produced between groups: $t_{34} = 38.83$, $p < 0.001$). Error bars indicate SEM. ***$p = 9.44 \times 10^{-30}$. **c** Comparison of choice similarity metrics between the main fatigue experiment and control experiment, averaged across the selected epoch. Relative to the baseline choice phase, positive values indicate more risk-seeking behavior in the subsequent choice phase, whereas negative values indicate more risk-averse behavior. There was a significant difference between choice behavior in the main and control groups (Unpaired $t$-test (two-tailed), $t_{34} = 2.60$, $p = 0.013$). Participants in the control experimental group did not significantly alter their choice behavior during 10U exertion (Unpaired $t$-test (two-tailed) $t_{16} = 1.61$, $p = 0.126$), while those in the main experimental group became significantly more risk-averse for effort during 80U exertions (two-tailed one-sample $t_{18} = 2.13$, $p = 0.047$). These results indicate that changes in choice preferences are not simply the result of the experience of repeated exertions, but instead are induced by fatigue resulting from high intensity exertion. Error bars indicate SEM. *$p < 0.05$.

bilateral insula and ACC) that has been shown to be correlated with effort value[1–8,10], and we also found that rIns was sensitive to chosen effort value in the fatigue condition but not the baseline condition. One potential interpretation of our rIns finding could be that this region is particularly sensitive to chosen effort value in the fatigue condition because, while in a fatigued state, individuals rely more on proprioceptive and interoceptive feelings in order to make inferences about the functional state of the motor system, which serves to guide effort valuation. Resolving this possibility is beyond the design of our current study and will require techniques that allow for the precise query of an individual's interoceptive and proprioceptive sensations during choice, dissociated from effort valuation.

It is important to mention that physical fatigue may not only arise from central mechanisms in the cortex, but also from peripheral mechanisms at the level of muscles[39]. Our neuroimaging results speak to the mechanisms of central fatigue by showing how motor cortical state changes following exertion, while also drawing a link between these fatigue-induced changes in PM activity and subjective preferences for effort. While peripheral processes were certainly a factor in effort valuations, and our control experiment clearly shows muscle-related physiological signatures of fatigue, our imaging experiment was not designed to study such contributions to fatigue representations in the central nervous system. In the future, experiments that record from muscles and motor units, in concert with functional neuroimaging, could elucidate the interactions between the central and peripheral mechanisms that give rise to subjective valuations of effort.

Recent studies of prospective effort valuation have separated the timing of effortful exertion from prospective valuation and decision-making about effort[8–10], and showed that in this context the vmPFC encodes decision-values for prospective effort. In contrast, many earlier studies of effort valuation[1–6] had intermixed exertions and effort choices, and these studies reliably identified a network of brain activity that included the ACC and bilateral insula. It has been suggested that in such experimental designs, ACC and insula activity could be related to preparatory motor signals that integrate effort value[5,7,10]. It is important to mention that in a previous experiment that employed a similar paradigm, which completely separated decision-making and exertion, we found that vmPFC encoded effort value[9]. In contrast, participants in the current study performed fatiguing exertions that were intermixed with effort choices and we found that effort value was encoded in ACC and bilateral insula. With this in mind, it is possible that the context of the effort-based decisions in the current paradigm (in which choices were interspersed with fatiguing exertion trials) could have led to the differences in neural results between our previous study of effort valuation and our current study about fatigue.

By integrating computational modeling with neural analysis, we provide evidence that motor cortical state is integral to the brain's representation of fatigue-induced changes in effort value. Our work outlines a mechanism by which information about PM state serves to modulate effort value, supporting effort-based decisions while fatigued. Rather than changes in motor cortical state being a reflection of the fatigue-induced changes in effort value, these modifications to effort value appear to result from a

miscalibration of motor cortical state in response to fatiguing exertions. These results suggest that decisions about effort while one is fatigued rely on motor signals in PM and effort valuation in the rIns.

## Methods

**Experimental setup.** Presentation of visual stimuli and acquisition of behavioral data were achieved using custom MATLAB (http://www.mathworks.com) scripts implementing the PsychToolBox libraries[40]. During fMRI, visual feedback was presented via a projector positioned at the back of the room. Participants viewed a reflection of the projector in a mirror attached to the scanner head coil.

An MRI compatible hand clench dynamometer (TSD121B-MRI, BIOPAC Systems, Inc., Goleta, CA) was used to record grip force effort exertion. During experiments, signals from this sensor were sent to our custom designed software for real-time visual feedback of participants' exertion. Effort exertion was performed while participants held the force transducer in their right hand with arm extended while lying in the supine position.

To record participants' choices we used an MRI compatible multiple button-press response box (Cedrus RB-830, Cedrus Corp., San Pedro, CA) held in the left hand.

**Participants.** All participants were right-handed and were prescreened to exclude those with prior history of neurological or psychiatric illness. The Johns Hopkins School of Medicine Institutional Review Board approved this study, and all participants provided informed consent.

Thirty-healthy individuals participated in the experiment, 10 of which were ultimately excluded from the final analyses for one or a combination of behavioral reasons. Participants were excluded if they were unable to generate salient associations between effort levels and applied effort ($n = 5$; r-squared between reported and actual effort during the recall phase, was <0.5). In addition, participants were excluded if their subjectivity parameter $\rho$, obtained from either choice phase (baseline or fatigue), was beyond two standard deviations of the population mean for that phase and their temperature parameters ($\tau$) were near zero, indicative of random choice ($n = 5$). The final analyses included $N = 20$ participants in total (mean age, 24 years; age range, 18–34 years; 9 females).

A portion of these data, pertaining to only the baseline choice phase ($n = 12$), was reported in one of our previous studies[9]. However, the results from the fatigue choice phase, the comparison between baseline and fatigue choice phases, and moderation analyses reported in the present study are completely distinct from those reported in either the main text or the supplemental materials of that previous paper.

**Experimental paradigm.** Prior to the experiment, participants were informed that they would receive a fixed show-up fee of $50. It was made clear that this fee did not, in any way, depend on performance or behavior over the course of the experiment. The association, recall, and choice phases of the experiment, described below, are identical to those we have previously used[9].

The experiment began by acquiring participants' maximum voluntary contraction (MVC) by selecting the maximum force achieved over the course of three consecutive repetitions on the hand-clench dynamometer. During these repetitions participants did not have knowledge about the subsequent experimental phases, and were instructed (and verbally encouraged) to squeeze with their maximum force.

Next, participants performed an association phase in which they were trained to associate effort levels (defined relative to MVC) with the force they exerted against the dynamometer (Supplementary Fig. 8a). Effort levels were on a scale that ranged from 0 (corresponding to no exertion) to 100 (corresponding to a force equal to 80% of a participant's MVC). A single training block consisted of five trials of training for each target level, where the target levels varied from 10 to 80 in increments of 10, and training blocks were presented in a randomized order. We did not perform association trials at the highest levels of effort to minimize the possibility that participants would become fatigued during this phase. A single trial of a training block began with the numeric display of the target effort level (2 s), followed by an effort task with visual feedback in the form of a black vertical bar, similar in design to a thermometer, which increased in white the harder participants gripped the dynamometer (4 s). The bottom and top of this effort gauge represented effort levels 0 and 100, respectively. Participants were instructed to reach the target zone (defined as ±5 effort levels of the target) as fast as possible and maintain their force within the target zone for as long as possible over the course of 4 s. Visual indication of the target zone was colored green if the effort produced was within the target zone, and red otherwise. At the end of the exertion, if participants were within the target zone for more than two thirds of the total time (2.67 s) during squeezing, the trial was counted a success. These success criteria were meant to ensure that participants were exerting effort for a similar duration across all effort conditions. To minimize participants' fatigue, a fixation cross (2–5 s) separated the trials within a training block, and 60 s of rest were provided between training blocks.

Following the association phase, we performed a recall phase to test if participants successfully developed an association between the effort levels and the actual effort exerted (Supplementary Fig. 8b). Participants were tested on each of the previously trained effort levels (10–80, increments of 10), six times per level, presented in a random order. Each recall trial consisted of the display of a black horizontal bar that participants were instructed to completely fill by gripping the transducer—turning the force-feedback from red to green once the target effort level was reached. For the recall phase, the full bar did not correspond to effort level 100 as in the previous phase, but instead was representative of the target effort level being tested on a particular trial. Participants were instructed to reach the target zone as fast as possible, to maintain their produced force as long as possible, and to get a sense of what effort level they were gripping during exertion (4 s). Following this exertion, participants were presented a number line (from 0 to 100) and told to select the effort level they believed they had just gripped. Selection was accomplished by moving the computer mouse to the point of selection and clicking to finalize the response. Participants had a limited amount of time to make this effort assessment (4 s), and if no effort level was selected within the allotted time the trial was considered missed. No feedback was given to participants as to the accuracy of their selection.

To examine the effects that fatigue has on the behavioral and neural representations of effort valuation, we scanned participants' brains with fMRI while they made decisions about prospective effort, before and after they performed repetitive physical exertions until exhaustion. Before being presented with the effort decisions, participants were told that 10 of their decisions would be selected at random and played out at the end of the experiment, and that they would have to remain in the testing area until they successfully achieved the selected exertions. Since trials were extracted at random at the end of the experiment, participants were instructed that they did not need to spread their exertion over all of their trials and should treat each effort decision individually.

During the baseline choice phase, meant to elicit effort preferences in a rested (unfatigued state), participants were presented with a series of effort choices between two options shown on the screen under a time constraint (4 s): one option entailed exerting a small amount of force (S) with certainty (known as the "sure" option); whereas the other entailed taking a risk which could result in either a large exertion (G) or no exertion, with equal probability (known as the "flip" option) (Fig. 1a, effort-based choice). The effort levels were presented using the 0 to 100 scale that participants were trained on during the association phase. The specifics of these effort choices can be found in our previous study that used this choice set to model subjective value of effort in a rested state[9]. Participants made their choices by pressing one of two buttons on a hand-held button-box with either the first or second digits of the left hand. Gambles were not resolved following choice and participants did not perform the exertion task during this phase of the experiment. One hundred and seventy effort choices were presented consecutively in a randomized order. Participants were encouraged to make a choice on every trial, however there was no penalty for failing to make a decision within the four second time window. Failure to make a choice in time was logged as a missed trial, and was not repeated.

Following the baseline choice phase, participants performed the fatigue choice phase of the experiment, in which they underwent repeated physical exertions to bring them into a fatigued state. After participants were fatigued, they alternated between blocks of prospective effort decisions and exertions, in order to maintain a state of fatigue throughout their subsequent effort choices. Exertion trials consisted of the 4 s presentation of a black horizontal bar (similar to the recall phase), which participants were instructed to fill by gripping the force transducer. During this phase of the experiment the amount of force required to completely fill the bar was always 80 units of effort. If participants were able to maintain their exertion at an effort level of 80 ± 5, for more than two-thirds the total exertion time (2.67 s), it was counted as a successful repetition—though this success/failure classification was not explicitly displayed to participants after performing the exertion. Within a single exertion block, repeated exertion trials were presented, separated by a fixation cross (3 s). Exertion trials were repeated until at least 75% of all trials within a block were counted as failures (Fig. 1a, Exertion Trial). Once a participant had undergone the initial exertion block, which was comprised of a minimum of 10 trials, they were presented with alternating choice and exertion blocks (Fig. 1b). Each choice block consisted of 10 effort decisions randomly sampled from the same choice set used in the baseline choice phase. Subsequent exertion blocks operated in the same way as the first, differing only in that they were comprised of a minimum of five trials. In this way, completion of the fatigue choice phase consisted of 17 back-to-back exertion and choice blocks. The repeated exertion blocks were meant to maintain participants in a fatigued state throughout their choices during the fatigue choice phase. Participants were informed prior to this phase that they would likely become tired/fatigued but that they should try their best to maintain successful exertion on each exertion trial. In addition, participants were explicitly instructed that the performance during the exertion trials was independent of the options and outcomes during choice.

At the end of the scanning phase, 10 choice trials were selected at random to be implemented. These trials could be drawn either from the baseline or fatigue choice phases. Participants remained in the testing area until they achieved the required exertions from the randomly extracted trials.

**Control Experiment 1.** To test if changes in effort preferences, between the first and second choice phases, were the result of a mere exposure effect of the effort

options, and not specifically related to fatigue, we performed a behavioral control experiment in which a new set of participants performed the same experimental phases described above without being exposed to fatiguing exertions. A group of 10-healthy participants, separate from those that performed the main experiment, took part in this experiment. One participant was excluded because they were unable to generate salient associations between effort levels and applied effort ($n = 1$; $r$-squared value between reported effort during the recall phase and perfect reporting was <0.5). These individuals first performed the recall, association, and baseline choice phases. After the baseline choice phase was complete, participants were required to rest for ~5 min and then performed another session of effort choices. This allowed us to compare effort choice behavior between the two groups (and choice phases), and test if the mere exposure of the choices resulted in significant changes in effort preferences. The final analysis included $N = 9$ participants in total (mean age, 19 years; age range, 18–21 years; 6 females).

**Control Experiment 2.** We performed an additional control experiment to test that our fatigue paradigm imparted self-reported increases in feelings of fatigue, associated changes in muscle physiology, and to further examine how choice preferences were modulated by the level of fatiguing exertion. A group of 21-healthy right-handed participants, separate from those in either of the previous two experiments, took part in this experiment. Two participants were unable to complete the experiment after exceeding the specified failure threshold (see below), and were therefore not considered for analysis. Of the remaining participants, one was excluded because they did not generate a salient association between effort levels and applied effort ($r$-squared between reported and actual effort during the recall phase was <0.5); another was excluded because their percentage of accepted effort gambles during the baseline choice phase, was beyond two standard deviations of the mean proportion of acceptance. The final analysis for this experiment included a total of $N = 17$ participants (mean age, 26 years; age range, 21–37 years; 11 females).

Prior to the experiment, participants were informed that they would receive a fixed show-up fee of $50 if they were able to complete the experiment (and $10 otherwise). It was made clear, prior to making any decisions about prospective effort, that this fee did not depend on their choices. The control experiment progressed similarly to the main experiment, first with acquisition of participants' MVC, followed by association and recall phases, and a baseline choice phase. However for this control, we modified the fatigue choice phase to test how varying levels of fatiguing exertion influence self-reported ratings of fatigue, muscle physiology, and subjective valuation of effort.

During the modified fatigue choice phase participants alternated between blocks of prospective effort choices and different levels of repeated physical exertion. As in the main experiment, exertion trials consisted of the 4 s presentation of a black horizontal bar, which participants were instructed to fill by gripping the transducer. However, during this modified fatigue choice phase, the amount of force required to completely fill the bar was either 10 (low effort) or 60 (high effort) units of effort. We also introduced a more stringent success criteria to ensure that a consistent amount of fatiguing exertion trials were performed within and across participants. Participants were required to perform five successful exertions (the successful trials need not be consecutive) before being able to progress. Furthermore, if the total number of failed exertion trials (across all exertion blocks) exceeded 20, the experiment ended. Participants were informed prior to the start of the phase that repeated failure to squeeze at the required level would result in the premature termination of the experiment and that they should try their best to succeed on each exertion trial.

Participants first performed a low effort section, which alternated between blocks of low effort exertion trials (five blocks) and blocks of choices (five blocks), with each block composed of 10 trials. This was followed by a high effort section which alternated between blocks of high effort exertion trials (six blocks) and blocks of effort choices (six blocks). After this high effort section of the experiment, participants were given a two minute period of rest, and subsequently performed a second low effort section where they once more alternated between low effort exertion trials (six blocks) and blocks of effort choices (six blocks). The effort options in the modified fatigue choice phase were pseudo-randomly extracted from the same choice set used in the baseline choice phase and the main experiment, to ensure that options in each choice block sampled a range of gamble and sure values. Overall, this experimental design allowed us to assess how effort preferences were influenced by varying levels of exertion.

To obtain self-report measures of fatigue, before and after each exertion block, participants were queried on their level of agreement ("Strongly disagree", "Somewhat disagree", "Unsure", "Somewhat agree", "Strongly agree") with the statement "I feel fatigued". Participants were free to select anywhere on the scale (not just the indicated points), and there was no time constraint for them to select their rating.

Throughout all exertion trials (for MVC, association, recall, and modified fatigue choice phases), we examined muscle activations using surface electromyograms (sEMGs). Three disposable electrodes (NeuroPlus[TM] A10040 Electrodes; Vermed.com, Buffalo, NY) recorded muscle activity targeting the right flexor digitorum superficialis muscle using a method for standardized EMG electrode placement[41], which has been previously used to study hand grip exertion[37]. EMG signals were amplified (AMT-8; Bortec Biomedical Ltd., Calgary,

Alberta, Canada) and bandpass filtered with high- and low-pass cutoff frequencies of 10 and 1000 Hz, and additionally filtered with a 60 Hz notch-filter. Signals were sampled at 5 kHz by a 16-bit data acquisition system (CED Micro1401-3; Cambridge Electronic Design Ltd., Cambridge, England). EMG acquisition was triggered prior to the onset of an exertion trial and encompassed the full 4 s exertion interval.

This control experiment has several features that are important to mention. First, since we experimentally modulated the level of effort required during fatiguing exertion blocks of the modified fatigue choice phase, and introduced a more stringent success criteria for exertion trials, we could more precisely modulate participants' state of fatigue to examine how different levels of exertion impact valuation of effort. Second, since we acquired measures of participants' self-reported ratings of fatigue we could confirm that participants do in fact feel fatigued following exertion, and were not simply apathetic to participation in the experiment. Third, recording of muscle activity during effortful exertions allowed us to confirm that the paradigm elicited physiological changes in muscle activity associated with fatigue.

**MRI protocol.** A 3 Tesla Philips Achieva Quasar X-series MRI scanner and radio frequency coil was used for all the MR scanning sessions. High resolution structural images were collected using a standard MPRAGE pulse sequence, providing full brain coverage at a resolution of 1 mm × 1 mm × 1 mm. Functional images were collected at an angle of 30° from the anterior commissure-posterior commissure (AC-PC) axis, which reduced signal dropout in the orbitofrontal cortex[42]. Forty-eight slices were acquired at a resolution of 3 mm × 3 mm × 2 mm, providing whole brain coverage. An echo-planar imaging (FE EPI) pulse sequence was used (TR = 2800 ms, TE = 30 ms, FOV = 240, flip angle = 70°).

**Effort choice analysis.** We used a two parameter model to estimate participants' subjective effort cost functions. We assumed a participant's cost function $V(x)$ for effort $x$ as a power function of the form:

$$V(x) = -(-x)^\rho, \quad x \le 0. \tag{1}$$

In this definition of effort cost, the effort level $x$ is defined as negative, with the interpretation being that force production is perceived as a loss. The parameter $\rho$ represents sensitivity to changes in subjective effort value as the effort level changes. A large $\rho$ represents a high sensitivity to increases in absolute effort level. $\rho = 1$ implies that subjective effort costs coincide with objective effort costs.

Representing the effort levels as prospective costs, and assuming participants combine probabilities and utilities linearly, the relative value between the two effort options can be written as follows:

$$\mathrm{RV}_{\mathrm{sure}}(G, S) = \mathrm{Value}(\mathrm{sure}) - \mathrm{Value}(\mathrm{gamble}), \tag{2}$$

$$\mathrm{RV}_{\mathrm{sure}}(G, S) = -(-S)^\rho - (-0.5(-G)^\rho), \tag{3}$$

$$\mathrm{RV}_{\mathrm{sure}}(G, S) = 0.5(-G)^\rho - (-S)^\rho, \tag{4}$$

where $\mathrm{RV}_{\mathrm{sure}}$ denotes the difference in value between the two options, and both $G < 0$ and $S < 0$ for all trials.

We then assume that the probability that a participant chooses the sure option for the $k^{\mathrm{th}}$ trial is given by the softmax function:

$$P_k(\mathrm{RV}_{\mathrm{sure}}(G, S)) = 1/[1 + \exp(-\tau \mathrm{RV}_{\mathrm{sure}}(G, S))], \tag{5}$$

where $\tau$ is a non-negative temperature parameter representing the stochasticity of a participant's choice ($\tau = 0$ corresponds to random choice).

We used maximum likelihood to estimate parameters $\rho$ and $\tau$ for each participant, using 170 trials of effort choices ($G, S$) with a participant's choice denoted by $y\epsilon \{0, 1\}$. Here, $y = 1$ indicates that the participant chose the sure option. This estimation was performed by maximizing the likelihood function separately for each participant:

$$\sum_{k=1}^{170} y_i \log(P_k(G, S)) + (1 - y_i)\log(1 - P_k(G, S)). \tag{6}$$

Parameter estimation was performed separately for choices in the baseline and fatigue choice phases. In this way, we obtained $\rho_{\mathrm{baseline}}$, $\tau_{\mathrm{baseline}}$, $\rho_{\mathrm{fatigue}}$, and $\tau_{\mathrm{fatigue}}$ parameters for each participant.

**Hierarchical Bayesian effort choice analysis.** For the second control experiment, we fit a prospect theory-inspired model of the process underlying valuation and choice. The basic model $V(x)$ was the same as that used in the main experiment. However, we used a hierarchical Bayesian approach to fit this model. This method gives us a statistical advantage by explicitly modeling and fitting parameters at the level of the participant (e.g., participant's preferences when making choices interspersed with low and high effort exertions) as well as at the level of the group (e.g., the mean population effort preferences). Using such a model, and fitting all participants' data simultaneously, reduced the influence of outliers and noise, and maximizes the ability to detect experimental fluctuations in effort preferences. This procedure also has the benefit of allowing us to directly model the effect of interest: the influence of low and high effort fatiguing exertions on changes in the valuation

and decision processes, at the population level. Notably, each of the low/high effort sections in the modified fatigue choice phase of the control experiment has fewer trials (50/60/60) than the main experiment (170) which makes it beneficial to use such a hierarchical Bayesian approach, since it leverages all the choice data and maximizes the possibility of identifying choice 'signals' of interest. Importantly, because of the limited set of choices in each fatiguing section, this methodology allows us to estimate parameters in sparsely sampled regions by modeling intrinsic structure within and between participants.

The underlying model used to fit subjectivity parameters ($\rho$ and $\tau$) from the choice data was structured similarly to a general linear mixed effects random intercept model, allowing for participant-specific intercepts but estimating population level effects of the experimental section. Parameters from the baseline choice phase and first 10U section (10U,1) of the modified fatigue choice phase are estimated in the following form:

$$\rho_i^S = A*\Phi(r_i^S), \tag{7}$$

$$\tau_i^S = B*\Phi(t_i^S), \tag{8}$$

$$\begin{bmatrix} r_i^{\text{Bas}} \\ t_i^{\text{Bas}} \\ r_i^{10U,1} \\ t_i^{10U,1} \end{bmatrix} \sim \mathcal{N}(M, \Sigma) \tag{9}$$

In this formulation, both $\rho$ and $\tau$ for participant ($i$) and experimental section ($S$) (baseline, first low effort section (10U, 1)), are represented and estimated as a participant-specific parameter ($r$ and $t$), drawn from parameter-specific normal distributions with estimated means ($M$) and covariance matrix ($\Sigma$), which are estimated population-level hyper-parameters. $\Phi$ is the unit Gaussian cumulative distribution function bounding the supported parameters to (0, 1), where $A$ and $B$ reflect imposed bounding constraints on the resulting estimates[43].

Changes in effort subjectivity parameters for the other two sections of the modified fatigue choice phase are modeled with a population-level section-specific (60U and 10U, 2) shared offset ($\delta$), which takes the following form:

$$\rho_i^S = A*\Phi(r_i^{10U,1} + \delta_r^S), \tag{10}$$

$$\tau_i^S = B*\Phi(t_i^{10U,1} + \delta_t^S). \tag{11}$$

The model then generates posterior estimates of the population's distributions for M, $\Sigma$, $\delta^{60U}$, and $\delta^{10U,2}$, which can be transformed into parameter-space using the above expressions. The probability that a participant chooses the sure option for the $k^{\text{th}}$ trial is given by the softmax function (in which $RV_{\text{sure}}$ is now contingent upon participant and exertion section-specific parameters, drawn from the population distribution):

$$P_k(RV_{\text{sure}}(G, S)) = 1/[1 + \exp(-\tau RV_{\text{sure}}(G, S))]. \tag{12}$$

This parameter estimation procedure was implemented using Monte-Carlo Markov Chain sampling methods provided by Stan version 2.19[44] and implementing a similar methodology as described by the hBayesDM package[43]. Standard hierarchical Bayesian methods were used, with constraints on the fit parameters of $\rho \in [0, 10]$ and $\tau \in [0, 6]$, and weakly informative distributions were chosen for the parameter priors in order to facilitate model convergence[45].

**Choice similarity measure**. As a secondary method to investigate how risk attitudes for effort change between conditions, we compared choices between conditions by computing a choice similarity metric. This metric is model-free, in that it does not assume an effort utility function and does not require the fitting of a model to the behavioral data. Since each effort gamble was presented twice (once per condition), it is possible to examine if choice behavior for identical effort options changed between experimental phases. To generate this metric, a value of 0 was assigned to a choice trial in the fatigue choice phase if the participant made the same choice as in the baseline choice phase; +1 was assigned to a choice if the participant accepted an effort gamble in the fatigue choice phase that they rejected in the baseline choice phase (i.e., more risk seeking behavior); and −1 was assigned if the participant rejected an effort gamble in the fatigue choice phase that they accepted in the baseline choice phase (i.e., more risk averse behavior).

**Exertion measure**. To quantify performance during exertion trials we calculated the mean exertion during the final 3 s of a trial. We excluded data from the first second of trials to remove variability in performance arising from different response times. This metric is presented in units of effort, relative to an individual participant's maximum exertion.

**Parameterization of decreases in effortful exertion**. To characterize the rate at which participants' capacity for effortful exertion decreases in the first block of exertion trials, we used an exponential decay model of the form:

$$F = F_0 e^{-\alpha k} \tag{13}$$

$F_0$ is the mean exertion force on the first exertion trial ($k = 0$) and $\alpha$ represents the rate that a participant's mean exertion decays. Larger values of $\alpha$ represent a more rapid decrease in mean exertion over trials, capturing how quickly an individual fatigues. We fit this model to each participants' exertion data separately, using performance over the first 10 trials of the first exertion block of the fatigue choice phase.

**Moderation analysis**. Moderation analysis is a form of linear modeling in which correlations observed in experimental data are explained by assuming that specific causal influences exist among the variables[34]. Specifically, moderation is said to occur when the relationship between two variables of interest depends on a third moderating variable (referred to as the moderator). The effect of the moderating variable is characterized statistically as an interaction that affects the relationship between the two other variables.

We performed a moderation analysis of our data to test the possibility that the relationship between fatigue-induced reductions in motor performance during exertion trials and changes in subjective effort valuation were moderated through neural deactivation in PM BOLD signal, at the time of choice. For this analysis we performed a between-participant multiple linear regression with the independent variable being mean exertion decay rate $k$ from the first exertion block, the moderating variable of change in PM activity between the baseline and fatigue choice phases, and the interaction between these two variables. The interaction was created by multiplying the independent variable and moderator after both were first Z-scored. The dependent variable of the regression was the fatigue-induced change in $\rho$ ($\rho_{\text{fatigue}} - \rho_{\text{baseline}}$). All of the variables were Z-scored before being entered into the regression. It is important to note that the ordering of the moderation analysis (i.e., the causal relationship) was informed by the temporal structure of the experiment; fatigue-induced exertion decays in the first exertion block preceded fatigue-induced changes in effort value and PM activity recorded at the time of choice. If the interaction term of the regression is significant, the moderation of the relationship between fatigue-induced decreases in performance and changes in subjective effort valuation is supported.

**Image processing and fMRI statistical analysis**. Image preprocessing: The SPM12 software package was used to analyze the fMRI data (Wellcome Trust Centre for Neuroimaging, Institute of Neurology; London, UK). A slice-timing correction was applied to the functional images to adjust for the fact that different slices within each image were acquired at slightly different time-points. Images were corrected for participant motion by registering all images to the first image, spatially transformed to match a standard echo-planar imaging template brain, and smoothed using a 3D Gaussian kernel (8 mm FWHM) to account for anatomical differences between participants. These set of data were then analyzed statistically.

General linear model: A GLM was used to estimate participant-specific (first-level), voxel-wise, statistical parametric maps (SPMs) from the fMRI data. The GLM included categorical box-car regressors beginning at the time of trial presentation and ending when a choice was indicated, for both the baseline and fatigue choice phase, for the chosen and unchosen effort options. Each of these categorical regressors included unorthogonalized parametric modulators corresponding to the objective value of the risky (Flip) and sure effort options. Trials with missing responses were modeled as a separate nuisance regressor. The fatigue choice phase included an additional nuisance regressor, modeled as a 4 s block, corresponding to exertion trials between choice blocks. Finally, regressors modeling the head motion as derived from the affine part of the realignment procedure were included in the model.

The regressors included in our imaging model were as follows:

1. Trials during the baseline choice phase in which the sure option was chosen (Box-car categorical regressor beginning at the time of choice presentation and ending at the time of response)

   a. Parametric modulator: Value of the sure, chosen option
   b. Parametric modulator: Value of the risky, unchosen option

2. Trials during the baseline choice phase in which the risky option was chosen (Box-car categorical regressor beginning at the time of the choice presentation and ending at the time of response)

   a. Parametric modulator: Value of the risky, chosen option
   b. Parametric modulator: Value of the sure, unchosen option

3. Trials during the fatigue choice phase in which the sure option was chosen (Box-car categorical regressor beginning at the time of choice presentation and ending at the time of response)

   a. Parametric modulator: Value of the sure, chosen option
   b. Parametric modulator: Value of the risky, unchosen option

4. Trials during the fatigue choice phase in which the risky option was chosen (Box-car categorical regressor beginning at the time of choice presentation and ending at the time of response)

   a. Parametric modulator: Value of the risky, chosen option
   b. Parametric modulator: Value of the sure, unchosen option

5. Exertion trials during the exertion block (Box-car categorical regressor beginning at the time of exertion trial presentation and lasting 4 s)

 i. Parametric modulator: Mean exertion (in terms of effort level) of the trial

 ii. Parametric modulator: Exertion trial number

6. Trials in which no choice was made in the allotted time (i.e., missed trials)
7. Regressors modeling the head motion as derived from the affine part of the realignment procedure were included in the model.

With these first-level models we created group models (second-level) to test brain areas that were generally sensitive to effort value. This was done by creating contrasts with the aforementioned parametric modulators for chosen and unchosen effort values, at the time of choice (i.e., difference between the value of the chosen and unchosen options, across both the baseline and fatigue phases). To test for areas of the brain sensitive to decision values for effort, irrespective of fatigue state, we created a contrast that captured the difference between chosen and unchosen effort. This contrast was created by subtracting the parametric modulator for the chosen risky and sure options (1.b, 2.b, 3.b, 4.b) from the chosen risky and sure options (1.a, 2.a, 3.a, 4.a). We also tested for regions of the brain in which decision value was sensitive to changes in bodily state induced by fatigue, by taking the difference between the value of the chosen and unchosen options, between the fatigue and baseline choice phases ($\{ [3.a + 4.a] - [3.b + 4.b] \} - \{ [1.a + 2.a] - [1.b + 2.b] \}$). In addition, we tested for changes in regions of the brain that were more generally sensitive to changes in bodily state induced by fatigue by examining the difference between the baseline and fatigue choice conditions, regardless of the effort values in question.

In a separate model using the same structure, we also included log(response time) as a parametric modulator in order to validate that chosen-minus-unchosen effort value signals were unrelated to potential choice difficulty effects. Region of interest (ROI) analysis of the chosen-minus-unchosen contrast between fatigue and baseline conditions indicated that the effort value effect was preserved.

Statistical inference: There is a degree of heterogeneity in the precise locations of the brain activations reported in previous studies of effort-based decision-making. However, these studies consistently implicate regions of dorsal ACC and bilateral insula in effort valuation. With this in mind we analyzed brain signals related to chosen effort value within independent ROIs taken at peak coordinates from Neurosynth.org[46] when using the term "effort": rInsula MNI coordinates (x, y, z) = [36, 24, 0]; lInsula MNI coordinates (x, y, z) = [−36, 24, 0]; ACC (x, y, z) = [0, 14, 48]. To analyze motor signals related to fatigue we used coordinates for premotor cortex reported in an independent study of fatiguing physical grip exertion (dorsal premotor cortex: (x, y, z) = [−36, −14, 64])[13]. Whole-brain contrasts for all figures are displayed at $p < 0.001$ (in yellow) and $p < 0.005$ with a 10-voxel extent threshold (in red), and statistical inference was performed within the SPM framework using small-volume correction, family-wise error corrected within these independently identified ROIs. These are standard methods used in affective neuroimaging[47]. Future studies will be required to test whether these results hold with whole-brain statistical corrections.

To clarify the signal patterns in our contrasts we created plots of effect sizes at the peak of activity (Figs. 3c, 4b, d). It is important to note that these signals are not statistically independent[48] and these plots were not used for statistical inference, but rather shown solely for illustrative purposes. Again, statistical inference was carried out within the SPM framework by small volume correcting in 5 mm spheres within our a priori coordinates.

**EMG analysis**. In order to compare EMG signals from homogenous exertion profiles, only successful exertion trials (spending at least 2.67 s of the exertion time within ±5 effort levels of the target) were considered. We considered the mean frequency of the power spectrum of the sEMG signal as a physiological measure of fatigue, which has been well documented in numerous EMG studies investigating fatigue[36–38]. Mean frequency measures for each trial were computed via the MATLAB 'meanfreq' function examining a frequency interval of 10–500 Hz. To examine differences between the first 10U and second 10U sections of the modified fatigue choice phase, we averaged mean frequency measures across all successful trials in each section within each participant, and compared these values between the two sections (Fig. 6d).

**Reporting summary**. Further information on research design is available in the Nature Research Reporting Summary linked to this article.

## Data availability

A reporting summary for this Article is available as a Supplementary Information file. The source data underlying Figs. 2a–e, 3c, 4b, d–f, 5c, d, 6a–e, 7a–c and Supplementary Figs. 1–7; and the statistical parametric maps for Figs. 3a, b, 4a, c are available for download at [https://osf.io/w2rdm/]. Fully anonymized raw behavioral and neuroimaging data files are available from the corresponding author upon reasonable request.

## Code availability

The code that support the findings of this study are available from the corresponding author upon reasonable request.

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

## Acknowledgements
This work was supported by the Eunice Kennedy Shriver National Institute of Child Health & Human Development of the National Institutes of Health under Award Number R01HD097619 and the National Institutes of Mental Health under Award Numbers R56MH113627 and R01MH119086 to V.S.C.

## Author contributions
V.S.C. and P.S.H. conceptualized the study. P.S.H. developed the methodology under supervision of V.S.C.; and P.S.H., S.X.C. and W.W.T. conducted the experiments. P.S.H. analyzed the data under supervision of V.S.C.; and V.S.C. and P.S.H. wrote the first draft of the manuscript.

## Competing interests
The authors declare no competing interests.
