## [Peer Review File · Nature Communications]

Reviewers' comments:

Reviewer #1 (Remarks to the Author):

In this manuscript, Hogan and colleagues used model-based fMRI to uncover some neural correlates of fatigue following physical effort exertion. The claim is that fatigue translates into a change in premotor cortex activity, which informs the estimation of effort cost performed by the insula and thereby modifies risk preference regarding effort exertion.

There are several features that make the study novel and interesting. I particularly appreciated the task that proposes choices between sure medium effort and a 50/50 chance of high/zero effort. This avoids the usual confound with reward estimation, and isolates the estimation of effort cost.

However, the solid findings are not so surprising. In particular, the notion that fatigue enhances the perception of effort cost and hence provides a bias on decision-making has been established in motor control theory. Furthermore, it has already been incorporated in models used in neuroscience studies (e.g., Le Bouc et al. 2016). Also, the idea that insular regions represent effort cost has been suggested many times, as the authors acknowledge (to justify their ROI selection).

The novel findings relate to the role of the premotor cortex (PMC), but this part is less solid, in my opinion. The main (within-subject) effect of fatigue is a decrease in PMC activity, but between-subject, the lesser the decrease the lesser the impact of fatigue on choices. This is interpreted as a “dyshomeostatic” representation in PMC: fatigue increases effort cost when the PMC does adjust to the physiological state of the muscles. But this is explaining a change with no change, which makes little sense to me. Besides, it seems like a post-hoc interpretation, as the opposite result would be more straightforward (bigger change in PMC activity associated to bigger change in choices).

The moderation model is interesting but it is just one possible pathway among many others. In particular, it is likely that the mean exertion rate also depends on PMC activity, so mediation model could be more appropriate. The different models could be compared with proper Bayesian model selection or any equivalent procedure. Besides, if I am not mistaken, there must be a negative link in the moderation model, which does not appear in the figure.

The PPI model is difficult to interpret. PPI analysis was made for testing the interaction between orthogonal physiological and psychological factors. Here they are not orthogonal, since the physiological regressor (insular time series) is precisely reflecting the psychological regressor (chosen value). The fact that the weight of the interaction in PMC activity changes between baseline and fatigue states could be driven by the fact that the insula reflects chosen value in the fatigue and not in the baseline state. In any case, it cannot be interpreted as a change in the transfer of information from PMC to insular cortex. I would suggest removing the PPI analysis altogether.

I have some other concerns:

- 1) The introduction should make the difference between posterior insula, where proprioceptive signals have been reported, and anterior insula, where representations of risk and effort costs have been found. This has consequences for the choice of the insula ROI.
- 2) The prediction about risk preference for effort is rather obscure in the introduction. I understood what the authors mean when seeing the behavioral task and model, but still, the description can be improved.
- 3) Many participants have been excluded from the analysis, which may be a sign that the task is not intuitive for a significant number of people. It would be reassuring to know that the results hold when including these participants.
- 4) It would also be reassuring to have a model-free behavioral result, such as a choice rate (in favor of the sure option), to check that the effect does not rely on a particular way of modeling choices.
- 5) The relationship between number of repetitions and mean exertion is not straightforward, it should be better explained. Besides, one effect is step-like and the other linear, so we have no illustration of the exponential decay fitted with the model, which occurs within the first block if I correctly understood.
- 6) Along the same line, we have no time course for the increase in effort cost, or the related brain activity. It would be informative to show whether the change in choices follows the decrease in mean exertion.
- 7) I do not get the difference between model 1 and model 5 in the comparison shown in supplementary material. Having $v = xp$ with an inverse temperature in the logistic regression model is the same thing as having $v = k xp$. I suspect model 5 is just model 1 with a redundant parameter, which would explain why the AIC is higher.
- 8) I am not sure about which model was fitted for the illustration in Figure 2C. It looks like a logistic function with both slope and bias parameters, but what we want to see is the model used to analyze fMRI data. This should be clarified. Besides, it would be more informative to show a group average and not just a (best) typical subject.
- 9) It is nice to have a control experiment, for the effect of time or repetition of choices, but we need a significant difference with the main experiment, not just an absence of effect, which may be due to low statistical power ($n=7$).
- 10) The exclusion of apathy based on RT is somewhat arbitrary. One could equally say that increasing RT would reflect fatigue and not apathy. I would remove this part.
- 11) The fMRI results are rather weak and heavily rely on ROI. In that regard I was surprised not to see

the vmPFC, which was central in the authors' previous study.

12) It is problematic that the difference between chosen and unchosen value is not significant in baseline insular activity. One cannot conclude from this observation that the insula encodes effort cost in a normal state. What fatigue seems to be doing is increasing the sensitivity to effort cost.

13) In the end, did subjects manage to perform the effort level that they had selected? My concern is that they may have avoided efforts that they believe they would not be able to reach, and not efforts that were too costly.

Reviewer #2 (Remarks to the Author):

This paper investigated the effect of fatigue on behaviors and fMRI signal in a choice task of probabilistic efforts. The authors concluded that fatiguing exertions increases participant's subjective cost of effort compared to the baseline/rest state and that motor cortical state in premotor cortex influence effort value computations in insula and increased subjective valuation of prospective physical effort when fatigued. The relationship between body physical state and valuation of prospective effort is an important and new topic which can provide insights into our real life. Therefore, I think this paper is potentially suitable and timely for the readership of Nature Communications.

Having said that, my enthusiasm for this study is weakened by several major concerns listed below, mostly regarding the task design and partly the way how fMRI data was analyzed.

1 The order of baseline choice phase and fatigue choice phase

After practice (Association phase + Recall phase), participants conducted baseline choice phase. Then, they were exposed to exertion trials and fatigue choice phase. Although the authors attributed behavioral changes in the fatigue choice phase to fatigue, I am not convinced of this conclusion. It is possible that participant's experience of exertion trials not fatigue changed his/her behavior. To rule out this possibility, the authors need to confirm at least the followings.

(a) It is necessary to ask the same participants to do the baseline choice task a few hours or days after fatigue choice condition and compare the two baseline choice outcomes.

(b) It is preferable to split fatigue choice data into two parts, and compare the former and latter behaviors. If behavioral change truly reflects fatigue, the authors should be able to observe the shift in logistic curves similar to Fig. 2c.

(c) It is desirable to collect subjective evaluation of fatigue to justify that participants felt fatigue.

2 Learning or fatigue

In Figure 2b, the authors showed the decrease in the mean exertion and interpreted it as an index of fatigue. However, as the criteria for each block is 75 % of failures and a minimum number of trials, participants can easily and gradually understand that they can adapt and finish each block with lower force, which does not necessarily support the authors' interpretation. This may be reflected in a

complicated plot (except upper right two participants) in Fig. 2e. Therefore, some physiological measurements of fatigue or at least a questionnaire data about fatigue would be necessary (related to 1 (c)).

3 Neural correlates of effort value difference

There seems to be activity other than Insula and dACC. For instance, superior temporal area in Figure 3b. A table including all results and correlation analysis of the results with “fatigue” would be necessary.

4 Thresholds for GLM analysis are all different

The authors used $P < 0.05$ small volume corrected, in Fig. 3a and 3b, but $P < 0.05$ FWE corrected for Fig. 4a. These arbitrarily chosen criteria would change the results. For example, if $P < 0.05$ small volume corrected, is applied in Fig. 4a, more brain structures may be revealed in Fig. 4.

5 Decrease in p in fatigue condition

Several participants exhibited decrease in the sensitivity to effort in Fig. 5c. How do you interpret this?

6 Time discount

As participant’s actual cost comes at the end of experiment as randomly selected 10 trials, time discount may also affect behaviors. Isn’t it necessary to consider this? At least discussion might be helpful to readers.

Reviewer #3 (Remarks to the Author):

The authors investigate the effect of fatigue on effort discounting. They find that subjects becomes more risk-averse (higher ρ) as fatigue builds up. A neural circuit consisting of Insula, ACC, and premotor cortex tracks this process. Some comments on the paper are the following:

Model:

- Please define parameters more clearly. If methods come at the end of the paper, the results should provide a quick summary of what parameters mean (perhaps even with relevant equations).
- Is the computational model statistically identified?

Neural results:

- Several of the neural regressors are unclearly described. For example, consider the statement “Each of these categorical regressors included...” This description of the regressors is too compact for me. Did you have box-cars for chosen and unchosen, and on top of that, regressors that were modulated by expected value of each (chosen and unchosen)? Please report this more explicitly and clearly, ideally with a short regressor count. Relatedly, in the chosen - unchosen comparison, what regressors are contrasted exactly?
- I also find the neural regressors not clearly motivated. Why is chosen vs unchosen the relevant

variable? Can the authors use trial-by-trial model-derived regressors and regress them against BOLD signal? This would make the connection between model and data stronger than currently.

- Relatedly, could the difference between chosen and unchosen simply reflect a difficulty effect?

Experimental control:

- Several of the neural results are not properly controlled for multiple comparisons.

- Rho remains the same in a control condition where there is no fatigue induction. This is an important control condition, but there should be an interaction to make this point statistically valid (time*condition).

- I don't get the control analysis for apathy— why would apathy but not fatigue be reflected in response times?

- The correlation in 2E seems entirely driven by 2 outliers.

Minor comment:

- The data in Figure 4f are surprisingly linear; is this a linear model fit?

We thank the reviewers for their comments. We have thoroughly revised our manuscript to include data from a new control experiment, as well as new analyses to address the reviewers' points. We believe the manuscript is substantially strengthened as a result. We now detail each of the changes in a point-by-point reply:

REVIEWER 1

Major Comments:

1.1) The solid findings are not so surprising. In particular, the notion that fatigue enhances the perception of effort cost and hence provides a bias on decision-making has been established in motor control theory. Furthermore, it has already been incorporated in models used in neuroscience studies (e.g., Le Bouc et al. 2016). Also, the idea that insular regions represent effort cost has been suggested many times, as the authors acknowledge (to justify their ROI selection).

While studies of motor control have examined how varying motor costs influence perceptual decisions to exert (Morel et al. 2017), to our knowledge, none directly investigated how decision-making changes as the result of physical fatigue. Moreover, although models of effort-based decision-making have incorporated parameters related to fatigue (Le Bouc et al. 2016; Meyniel 2013), these studies have not experimentally manipulated participants' state of fatigue and subsequently tested how these changes influence both decisions to exert and the associated neural activity. While the referenced study by Le Bouc and colleagues modeled fatigue, it did not experimentally modulate it and thus was not designed to examine how fatigue interacted with effort/reward valuation. Previous neuroeconomic studies of fatigue have mainly focused on the crosstalk between mental and physical fatigue (Blain et al. 2016; Blain et al. 2019).

To our knowledge, no studies have directly tested how physically fatiguing exertions influence the subjective valuation of effort, associated decision-making about effort, and the underlying neural activity associated with prospective effort valuation while fatigued. Our study was designed to address these questions, and provides a deeper understanding of the neurobiological basis of decisions about effort while in a fatigued state.

In the Introduction, we have now further elaborated on previous literature in the fields of motor control and decision neuroscience, which deal with effort-based decision-making and fatigue, and we have framed our study more clearly in the context of these previous works (page 4, paragraph 1).

Recently it has been proposed that feelings of fatigue may arise from inconsistencies between beliefs about the consequences of actions and actual sensory inputs and motor outputs¹⁶⁻¹⁸. With this in mind, it has been suggested that brain networks that process proprioceptive and exteroceptive signals from muscles and interoceptive signals from the internal state of the body and visceral organs, could be critical for generating feelings of effort and fatigue. These brain regions include somatosensory regions and the posterior insula¹⁹⁻²³. Notably, a

series of studies by Meyniel and colleagues examined how individuals performed bouts of physical exertion and choose to take rests, and found that portions of the posterior insula encoded signals that followed the time course of proprioceptive feelings associated with exertion and rest^{22,24}.

The findings of posterior insula encoding proprioceptive signals, and anterior insula encoding effort value^{4,7,25}, hint at the possibility that exteroceptive and interoceptive feelings could mediate effort values encoded by the insula. However, studies that examined neural signals at the time of exertion and rest were not designed to examine prospective valuation of effort, while studies of effort valuation have not investigated the influence of fatigue. Therefore it is not clear how brain signals related to fatigue might influence effort valuation. It should be noted that while these studies included model parameters related to fatigue^{22,24,26}, fatigue was not experimentally manipulated, making it difficult to infer how the state of fatigue influences effort valuation and associated brain activity. Furthermore, while recent studies have examined how physical²⁷ and cognitive²⁸ fatigue influence cognitive control during a temporal discounting task, they did not evaluate how fatigue influenced effort valuation. To our knowledge there have been no studies that have directly tested how changes in an individual's bodily state, through bouts of physical exertion, influence prospective valuation of effort and resulting decisions. Accordingly, there is a limited mechanistic understanding of how neural signals related to perceptions and sensations of fatigue influence effort valuation and decisions to exert.

Studies of motor control have also begun to examine how internal models of effort value are generated and how these representations influence decisions between potential movements and movement generation. However these studies have focused on behavior and did not examine the brain activity that encodes effort value. Furthermore, these motor control studies did not examine how physical fatigue influences effort valuation²⁹⁻³¹.

1.2) The novel findings relate to the role of the premotor cortex (PMC), but this part is less solid, in my opinion. The main (within-subject) effect of fatigue is a decrease in PMC activity, but between-subject, the lesser the decrease the lesser the impact of fatigue on choices. This is interpreted as a "dyshomeostatic" representation in PMC: fatigue increases effort cost when the PMC does adjust to the physiological state of the muscles. But this is explaining a change with no change, which makes little sense to me. Besides, it seems like a post-hoc interpretation, as the opposite result would be more straightforward (bigger change in PMC activity associated to bigger change in choices).

Our *a priori* hypotheses about the role of PM in effort-based decision-making while fatigued were informed by previous studies. Specifically, neuroimaging and brain stimulation studies have shown that motor regions deactivate (relative to a rested state) following fatiguing exertions (Brasil-Neto et al. 1993; Samii et al. 1996; Benwell et al. 2005; Benwell et al. 2006; van Duinen et

al. 2007), which led us to hypothesize that fatigue-induced changes in subjective preferences could simply be the reflection of this altered PM state. Alternatively a number of recent reviews about fatigue (Stephan et al. 2016; Kuppuswamy 2017), as well as a study about chronic fatigue syndrome (van der Shaaf et al. 2018), have proposed that fatigue could be the result of a disconnect between the consequences of actions and actual sensory and motor outputs. This led us to propose an alternative hypothesis: that fatigue-induced changes in subjective preferences could result from a discrepancy the motor production that one believes they can achieve and their actual motor capacity following fatigue. We would like to stress that this dyshomeostatic hypothesis of PM activity was not a post-hoc interpretation of the data, but informed by previous works (Stephan et al. 2016; Kuppuswamy 2017; van der Shaaf et al. 2018).

Our main PM findings show that in a fatigued state (relative to a rested/baseline state), there is decreased activity in motor cortex, which aligns with previous motor studies of fatigue (Brasil-Neto et al. 1993; Samii et al. 1996; Benwell et al. 2005; Benwell et al. 2006; van Duinen et al. 2007). However, the extent of this PMC decrease is inversely related to changes in individuals' effort subjectivity – those individuals with more pronounced fatigue-induced increases in subjective effort value show less change in PMC BOLD activity as a result of fatigue. This suggests that the inability to reduce motor cortical activity following repeated exertions is associated with fatigue-induced increases in effort value, consistent with the dyshomeostatic account of fatigue that has been suggested in recent literature.

To further test the relationship between fatigue-induced changes in PM activity and effort value, we performed a moderation analysis between motor performance during early fatiguing trials and fatigue-induced changes in effort value. We reasoned that if the inability to appropriately reduce motor cortical activity is associated with increased subjective fatigue, this would manifest through the relationship between fatigue-induced changes in motor performance and subjective valuations of effort. Accordingly, we found that PMC deactivation moderated the relationship between fatigue-induced changes in motor performance and choices about effort; those individuals with less PM deactivation (i.e., who did not adjust motor cortical activity in response to fatigue) had a stronger relationship between changes in motor performance and subjective valuations of effort. This result gives further credence to the dyshomeostasis hypothesis by showing that the extent of PM deactivations (or lack thereof) is related to how motor performance is connected to effort-based choice.

We have now more clearly elaborated on our *a priori* hypotheses regarding PM activity in the Results (page 15, paragraph 1).

Considering our prediction that motor state would be an integral factor in representations of fatigue-induced changes in subjective value, we tested if changes in subjective effort preferences ($\rho_{Fatigue} - \rho_{Baseline}$) were indexed by fatigue-induced changes in activity in PM. It has been suggested that fatigue might arise as the result of discrepancies between expectations about the consequences of actions and actual sensory and motor outputs^{16–18}. In the context of our experiment, if an individual's motor system does not appropriately adjust its

resting state in response to repeated exertions, one might feel that effort is particularly costly because of the discrepancy between the motor production that one believes they can achieve and their actual motor capacity following fatigue. In contrast, it is possible that fatigue arises from an accurate representation of an individual's bodily state and that changes in subjective preferences are simply a reflection of the altered motor cortical state that occurs following fatiguing physical exertion¹¹⁻¹⁵. Using each participant's fatigue-induced change in ρ ($\rho_{Fatigue} - \rho_{Baseline}$) as a second-level covariate in our imaging model, we found that those individuals with greater fatigue-induced changes in subjective effort parameters exhibited less change in PM activity (Figure 4c, d; Supplementary Table 4). These findings are consistent with the idea that a miscalibration of motor regions is related to fatigue-induced changes in subjective effort preferences – those participants that find effort to be particularly costly following fatigue may be those who do not modify their motor cortical activity to accommodate the reduced motor capacity that results from repeated exertion.

We have also further expanded upon this point in our Discussion section (page 22, paragraph 3).

In order to make informed decisions about exertion, it is essential to have an idea of one's own physiological state (e.g., capacity for descending motor drive, motor unit recruitment, etc.). A number of neuroimaging and TMS studies have found that repeated physical exertion results in deactivations of motor and premotor brain regions, which have been suggested to reflect the reduced capacity for recruitment of motor pathways in the central nervous system¹¹⁻¹⁵. However, it is unknown how such fatigue-induced changes in motor cortical activity are related to decisions about exertion. Here we show that at the time of choice, even when exertion is not being performed, neural deactivations in contralateral PM are related to fatigue-induced changes in subjective preferences for effort. Specifically, those individuals that exhibit the greatest fatigue-induced increases in subjective effort value exhibit the least fatigue-induced deactivation in PM BOLD activity. These results are consistent with the idea that fatigue-induced inflations of effort may be associated with dyshomeostatic representations in PM¹⁶⁻¹⁸; those individuals who do not reduce their motor cortical activity following fatiguing exertion find prospective effort to be more costly. Effort might feel particularly costly to these individuals because their motor system continues to recruit the same level of descending drive as in a baseline/rested state, in spite of their muscles being fatigued and physiologically incapable of generating the same level of motor output. This idea aligns with recent theoretical accounts of fatigue that suggest discrepancies between perceptions of ability and actual sensorimotor capacity may give rise to feelings of fatigue¹⁶⁻¹⁸.

1.3) The moderation model is interesting but it is just one possible pathway among many others. In particular, it is likely that the mean exertion rate also depends on PMC activity, so mediation model could be more appropriate. The different models could be compared with proper Bayesian

model selection or any equivalent procedure. Besides, if I am not mistaken, there must be a negative link in the moderation model, which does not appear in the figure.

The reviewer makes a good point that it is important to consider the full space of alternative models. Our model space was constrained by the causal nature of our experiment – decays in mean exertion preceded changes in premotor activity and subjective valuation of effort. In order to run a full mediation analysis, all regressors must be significantly correlated (Judd and Kenny, 1981). Since motor performance decay and the degree of PM deactivations were not significantly correlated ($r = -0.20$, $p = 0.40$), our data did not meet the criteria to run a formal mediation analysis. Importantly, the lack of a significant relationship between the motor performance decay parameter and PM deactivation precludes even a partial mediation, although if it were only the direct path that were insignificant (i.e., the relationship between motor performance decay and fatigue-induced changes in effort parameters), a partial mediation would still be possible (Shrout and Bolger 2002).

We have now referenced that our data does not allow for a formal mediation analysis in the Results section (page 17, paragraph 2).

We also considered an alternative causal model in which PM deactivations mediated the relationship between fatigue-induced motor performance decays and changes in subjective valuation of effort. However, there was not a significant correlation between fatigue-induced motor performance decays and PM deactivations ($r = -0.20$, $p = 0.40$), which precluded the performance of a formal mediation analysis³⁵. These findings lend further support to the specificity of PM deactivations moderating the relationship between fatigue-induced changes in motor performance and changes in subjective effort value.

The reviewer is correct that there is a negative moderating influence of PM deactivation on the relationship between fatigue-induced decrease in exertion and subjective effort value. We inadvertently omitted a negative sign in Figure 4e, which depicted the moderation model. We have now clarified the description of our moderation analysis (page 16, paragraph 2).

To further explore the role of PM in fatigue-induced changes in subjective effort value, we performed an analysis to test the hypothesis that PM activity has an influence on the relationship between fatigue-induced reductions in motor performance during exertion trials and changes in subjective effort valuation (illustrated in Figure 2e). To test this hypothesis we used a moderation analysis, a form of linear modeling in which correlations observed in the data are explained assuming that a specific set of causal influences exists among the variables³⁴. This analysis alone does not establish causality but tests the hypothesis that a moderator variable alters the strength of the causal relationship between experimental variables. We fit a between-participant model that assumed that decreases in motor performance during exertion trials influenced changes in subjective valuation of effort, and that the extent of PM deactivation moderated

the relationship between these variables. This model revealed that the degree of PM BOLD deactivation (in the Fatigue Choice Phase compared to Baseline) had a significant moderating influence on the relationship between fatigue-induced decreases in performance (during exertion trials) and the increases in subjective valuation of effort (during choice) (Figure 4e, f). Those participants that exhibited less fatigue-induced deactivation in PM activity had a stronger relationship between motor performance decay and changes in subjective valuation of effort, whereas individuals who experienced greater fatigue-induced BOLD deactivation in PM showed a weaker relationship. These results further support the idea that decreased fatigue-induced PM deactivations are related to fatigue-induced changes in subjective effort preferences.

We have also updated Figures 4e and 4f.

Figure 4e, Moderation analysis. We tested the extent to which changes in PM activity moderates the relationship between exertion-induced decreases in effort capacity (indexed by mean exertion decay rate parameters) and changes in subjective effort value (indexed by $\rho_{\text{Fatigue}} - \rho_{\text{Baseline}}$).

Figure 4f, Illustrative plot of the moderating influence of PM deactivation on the relationship between fatigue-induced reduction in motor performance and increases in the subjective cost of effort. The lines represent linear regressions between mean exertion decay and fatigue-induced changes in ρ , separated by median split PM activity. The less PM deactivation between the Baseline and Fatigue Phases, the stronger the relationship between the fatigue-induced decay in motor performance and fatigue-induced increases in the subjective cost of effort.

1.4) The PPI model is difficult to interpret. PPI analysis was made for testing the interaction between orthogonal physiological and psychological factors. Here they are not orthogonal, since the physiological regressor (insular time series) is precisely reflecting the psychological regressor (chosen value). The fact that the weight of the interaction in PMC activity changes between baseline and fatigue states could be driven by the fact that the insula reflects chosen value in the fatigue and not in the baseline state. In any case, it cannot be interpreted as a change in the transfer of information from PMC to insular cortex. I would suggest removing the PPI analysis altogether.

The reviewer makes a good point that it is difficult to interpret the PPI analysis as it is currently structured, given the correlation between the seed activity and psychological variable. We included the psychological variable in the main model in the hopes of accounting for the main effect of the psychological condition, however we acknowledge that the correlation between the insula seed and the psychological condition may still lead to difficulty in interpreting the results. At the reviewer's suggestion, we have now removed the PPI analysis.

Minor Comments:

1.5) The introduction should make the difference between posterior insula, where proprioceptive signals have been reported, and anterior insula, where representations of risk and effort costs have been found. This has consequences for the choice of the insula ROI.

We have clarified the distinction between the posterior insula's encoding of proprioceptive signals and the anterior insula's encoding of risk and effort cost (page 4, paragraph 2).

The findings of posterior insula encoding proprioceptive signals, and anterior insula encoding effort value^{4,7,25}, hint at the possibility that exteroceptive and interoceptive feelings could mediate effort values encoded by the insula. However, studies that examined neural signals at the time of exertion and rest were not designed to examine prospective valuation of effort, while studies of effort valuation have not investigated the influence of fatigue. Therefore it is not clear how brain signals related to fatigue might influence effort valuation. It should be noted that while these studies included model parameters related to fatigue^{22,24,26}, fatigue was not experimentally manipulated, making it difficult to infer how the state of fatigue influences effort valuation and associated brain activity. Furthermore, while recent studies have examined how physical²⁷ and cognitive²⁸ fatigue influence cognitive control during a temporal discounting task, they did not evaluate how fatigue influenced effort valuation. To our knowledge there have been no studies that have directly tested how changes in an individual's bodily state, through bouts of physical exertion, influence prospective valuation of effort and resulting decisions. Accordingly, there is a limited mechanistic understanding of how neural signals related to perceptions and sensations of fatigue influence effort valuation and decisions to exert.

1.6) The prediction about risk preference for effort is rather obscure in the introduction. I understood what the authors mean when seeing the behavioral task and model, but still, the description can be improved.

We have now clarified our predictions regarding the risk preferences for effort (page 5, paragraph 3).

Here, we investigated the influence of fatigue on behavioral representations of subjective effort value, and the neural mechanisms by which fatigue interacts with the brain's valuation and decision-making circuitry. Behaviorally we hypothesized that fatigue, arising from repeated physical exertion, would result in individuals having an exaggerated subjective valuation of effort that would manifest as diminished risk preferences for prospective effort. Essentially, when individuals are faced with the option of exerting a certain amount of effort, versus a risky option involving either a greater amount of effort or no effort at all, they will be less willing to choose the risky option while in a fatigued state. This hypothesis has its basis in previous studies of effort-based decision-making that found individuals exhibited increased sensitivity to changes in subjective effort as objective effort levels increased (i.e., risk aversion for effort)^{6,7,9}. Neurally we hypothesize that decisions about prospective effort exertion have their basis in a value signal encoded in the anterior cingulate cortex (ACC) and insula. This hypothesis was informed by a number of neuroimaging studies that found a correlation between activity in these brain regions and behavioral measures of effort value^{1-8,25}. Given recent studies which found that insula encoded feelings of effort during bouts of exertion and rest²², we predicted that the insula would be sensitive to changes in effort value as a function of fatigue. Together these hypotheses form a neurobehavioral account of fatigue, which recruits regions of the brain responsible for effort valuation and motor exertion, to inform decisions about prospective effort while in an altered bodily state arising from prolonged exertion.

1.7) Many participants have been excluded from the analysis, which may be a sign that the task is not intuitive for a significant number of people. It would be reassuring to know that the results hold when including these participants.

We utilized a two-step exclusion criteria to ensure that we analyzed data from participants that (1) developed strong effort associations and (2) were making consistent choices. To be sure that we analyzed data from participants who developed salient associations between units of effort and applied effort, we excluded participants ($n = 5$) whose r-squared value between reported and actual effort during Recall was less than 0.5. In the remaining participants, we excluded those in which the subjectivity parameter ρ , obtained from either Choice Phase (Baseline or Fatigue) was beyond two standard deviations of the population mean for that phase or whose temperature

parameters (τ) were near zero, indicative of random choice ($n = 5$). This second exclusion criteria ensured that the remaining participants were making consistent choices.

As the reviewer notes, our conservative exclusion criteria resulted in a number of participants being removed from our main analyses. We have now re-analyzed our data, only excluding the participants who did not generate salient associations ($N = 25$). We maintained our exclusion of participants without salient effort associations, reasoning that their choices would be difficult to interpret if they did not understand the effort levels about which they were making choices. In this analysis, all of our primary behavioral and imaging results held. It is important to mention that we were unable to perform our covariate and moderation analyses in this new population because not all of these participants exhibited reliable parameter estimates.

Below we have included the results from the new analysis which included all participants except those that did not develop strong effort associations ($N = 25$). Importantly, all of the behavioral and imaging results remained significant in this expanded cohort of participants.

(Above: reproduction of Figure 2a, b) Behavioral results ($N = 25$) a. Performance is represented as the mean number of repetitions until the exertion block termination criterion was met (75% of trials within an exertion block failed), across participants. Higher values indicate more exertions were required to reach exhaustion. The initial exertion block is indicated in the shaded gray region. Participants' ability to successfully achieve the performance criteria diminished across exertion blocks, and participants consistently performed the same number of exertion trial repetitions (~ 5 ; the required minimum) after the second exertion block. Error bars indicate

SEM. *** $p < 0.001$. **b.** Mean exertion during the Fatigue Choice Phase. Mean exertion is presented in units of effort. For the initial exertion block (shaded gray), the mean exertion for the first and last five trials is shown. For subsequent exertion blocks the mean exertion over the full exertion block is shown. Mean exertion decreased between the initial and final trials in the first exertion block and mean exertion never recovered to its initial level during the Fatigue Choice Phase. Error bars indicate SEM. *** $p < 0.001$.

Consistent with our previously reported behavioral results, participants' ability to successfully achieve the performance criteria diminished between the first and second exertion blocks (average reduction in high-intensity hand-clench repetitions between the first and second block: 7.96, two-tailed paired-sample $t_{23} = 4.31$, $p < 0.001$; one participant was excluded from this analysis because the number of exertion trials in their second block was greater than 3 S.D. above the mean, but is included in all subsequent analyses); and participants consistently performed the same number of exertion trial repetitions (about 5 trials; the required minimum) after the second fatigue block. Additionally, mean exertion decreased between the initial and final trials in the first exertion block (average reduction in mean exertion force: 7.97 effort units; two-tailed paired-sample $t_{24} = 4.94$, $p < 0.001$), suggesting that repetitive exertion rapidly decreased the capacity for effortful exertion. Finally, participants' mean exertion in subsequent blocks never recovered to its level in the first exertion block (first five trials), and all subsequent blocks were significantly reduced compared to the initial exertion block (two-tailed paired-sample $t_{24} > 4.07$, $p < 0.01$ for all exertion blocks after the first).

Since there were a number of participants ($N = 5$) in this expanded cohort did not exhibit reliable model parameter estimates from the Choice Phase, we examined a model-free metric of fatigue-induced changes in effort preference: the acceptance rate of effort gambles between the Baseline and Fatigue Choice Phases. We observed a significant reduction in the proportion of accepted effort gambles in the Fatigue Choice Phase, compared to Baseline (two-tailed paired sample t-test: $t_{24} = 2.17$, $p = 0.04$). The fatigue-induced decrease in risk-taking exhibited in this model-free analysis is consistent with the increased ρ reported in our original findings.

(Above: reproduction of Figure 3) Neural signatures of effort value ($N = 25$). Consistent with our previous results, activity in rIns and ACC was significantly correlated with the difference in chosen and unchosen effort value at the time of choice, across both the Fatigue and Baseline Choice Phases (panel a). Furthermore, rIns activity correlated with the difference in chosen and unchosen effort value in the Fatigue Choice Phase as compared to the Baseline Phase (panel b).

(Above: reproduction of Figure 4a, b) Neural signatures of exertion-induced fatigue ($N = 25$). Consistent our previous results, we found that brain activity (irrespective of value) in primary motor cortex and PM is reduced in the fatigue condition, relative to baseline at the time of choice (panel a; panel b shows illustration from ROI centered at peak of activation). We did not perform our covariate and moderation analyses in this expanded cohort because not all of these participants exhibited reliable parameter estimates.

1.8) It would also be reassuring to have a model-free behavioral result, such as a choice rate (in favor of the sure option), to check that the effect does not rely on a particular way of modeling choices.

We compared the acceptance rate of effort gambles, between the Baseline and Fatigue Choice Phases, as a model-free measure of choice behavior. We observed a significant reduction in the proportion of accepted effort gambles in the Fatigue Choice Phase, compared to Baseline (two-tailed paired sample t-test: $t_{19} = 2.73$, $p = 0.01$). The fatigue-induced decrease in risk-taking exhibited in this model-free analysis aligns with the increased ρ reported in our model-based analysis. Larger ρ values are associated with more risk-averse effort-choice behavior. We have now referenced this model-free analysis in the main text (page 10, paragraph 1) and included the results in the Supplementary Information.

Supplementary Figure 2. Effort gamble acceptance rate between Baseline and Fatigue Choice Phases. We observed a statistically significant decrease in the percentage of effort gamble choices made between the Fatigue and Baseline Choice Phases (mean change: 5.73% (SD = 9.38%), two-tailed paired sample $t_{19} = 2.73$, $p = 0.01$); on average participants choose the risky option less frequently when fatigued.

1.9) The relationship between number of repetitions and mean exertion is not straightforward, it should be better explained. Besides, one effect is step-like and the other linear, so we have no illustration of the exponential decay fitted with the model, which occurs within the first block if I correctly understood.

During the Fatigue Choice Phase, participants made effort-based choices interspersed with bouts of exertion. The first bout of exertion was meant to severely fatigue participants, and subsequent blocks were intended to maintain them in a fatigued state. If participants were able to maintain their exertion at an effort level of 80 ± 5 on an exertion trial, for more than two thirds the total exertion time (2.67 s), the trial was counted as a successful repetition. Exertion trials were repeated until at least 75% of all trials within a block were counted as failures. Once a participant had undergone the initial exertion block, comprised of a minimum of 10 trials, they were presented with alternating choice and exertion blocks. Subsequent exertion blocks were structured in the same way as the first, differing only in that they were comprised of a minimum of 5 trials.

Figure 2a illustrates the average number of repetitions, across all participants, in each exertion block of the Fatigue Choice Phase. The first exertion block had an average of 14.95 repetitions, indicating that it took approximately 15 trials for participants to reach a level of fatigue that caused them to fail the success criteria (i.e., 75% failure). Subsequent exertion blocks lasted for approximately 5 repetitions on average, indicating that participants were fatigued and were not able to successfully exert for many repetitions. This discrete success criteria led to the step-like effect in the number of repetitions until failure.

Figure 2b illustrates the mean exertion force (in terms of effort level) across all participants, in each exertion block of the Fatigue Choice Phase. This is a more continuous metric of task performance than the average number of repetitions, and captures participants' mean exertion over all trials within each block. The gray region of the figure shows the mean exertion of the first and last 5 trials of the first exertion block, which illustrates that repeated exertions lead to a decrease in capacity for exertion. Moreover, participants' mean exertion decreased through subsequent exertion blocks, showing that repeated exertions lead to fatigue and a decrease in motor output.

We have now clarified the distinction between the figures illustrating the number of repetitions until failure (Figure 2a) and mean exertion (Figure 2b) (page 8, paragraph 3).

Behavioral results from exertion trials indicated that participants' ability to successfully achieve the performance criteria diminished between the first and second exertion blocks (Figure 2a; average reduction in exertion repetitions between the first and second block: 9.26, two-tailed paired-sample $t_{18} = 4.69$, $p < 0.001$; one participant was excluded from this analysis because the number of exertion trials in their second block was greater than 3 SD above the mean, but is included in all subsequent analyses), and that participants consistently performed the same number of exertion trial repetitions (~ 5 trials; the required minimum)

after the second fatigue block. These data show that it took approximately 15 trials for participants to reach a level of fatigue that caused them to meet the failure threshold (i.e., 75% of trials within the block were failed exertions). Subsequent exertion blocks lasted for approximately 5 repetitions, indicating that participants remained in a fatigued state and met the failure threshold after fewer trials than in the initial exertion block. This discrete failure threshold led to a step-like effect in the number of repetitions until failure.

To provide a more continuous metric of motor performance we analyzed participants' mean exertion (in terms of effort level) within exertion blocks. We found that mean exertion decreased between the initial and final trials in the first exertion block (Figure 2b; average reduction in mean exertion force: 10.14 effort units; two-tailed paired-sample $t_{19} = 3.36$, $p = 0.003$), suggesting that repetitive exertion rapidly reduced the capacity for effortful exertion. Furthermore, participants' mean exertion in subsequent blocks never recovered to its level in the first exertion block (first five trials), and all subsequent blocks were significantly reduced compared to the initial exertion block (two-tailed paired-sample $t_{19} > 3.60$, $p < 0.05$ for all exertion blocks after the first) (see Supplementary Figure 1 for all participants' mean exertion data). It should be noted that mean exertion levels never diminished to zero, implying that on average participants were always trying to exert effort and their capacity was reduced over the course of repeated exertion blocks. Together these results illustrate that participants' capacity for exertion was decreased by repeated exertion, and that the exertion blocks interspersed throughout the Fatigue Choice Phase maintained participants in a fatigued state.

The reviewer is correct that we fit a decay model to the initial exertion block of the Fatigue Choice Phase in order to characterize fatigue-induced changes in mean exertion before any choices were made. The reviewer raises a good point that we did not include an illustration of the fits of this decay model, and we have now included a figure in the Supplementary Information to illustrate this data (Supplementary Figure 4).

Supplementary Figure 4. Illustration of the group-level mean exertion decay across all participants over the course of the first ten exertion trials, within the first exertion block of the Fatigue Choice Phase. The dashed line represents the results of the exponential decay model fit to the group-level means. Error bars indicate SEM.

1.10) Along the same line, we have no time course for the increase in effort cost, or the related brain activity. It would be informative to show whether the change in choices follows the decrease in mean exertion.

The reviewer raises a good point regarding the time course for increases in effort cost during the Fatigue Choice Phase. We began the phase with a prolonged exertion block requiring high-effort exertions (80U) to thoroughly fatigue participants, and subsequent exertion blocks were meant to hold participants in this fatigued state. As the reviewer noted in their previous comment, the number of exertions until failure remained relatively constant throughout the phase, while mean exertion levels decayed. To examine how effort cost might change over the course of the Fatigue Choice Phase, we compared choices between the Baseline and Fatigue Choice Phases. Since each effort gamble was presented twice (once in each Choice Phase), it was possible to examine if choice behavior for identical effort options changed between the experimental phases and evolved over the course of repeated exertion blocks. To address this question, we computed a choice similarity metric that captured how much participants' choice behavior changed between the Baseline and Fatigue Phases: a value of 0 was assigned to a choice trial in the Fatigue Choice Phase if the participant made the same choice as in the Baseline Choice Phase; +1 was assigned to a choice if the participant accepted an effort gamble in the Fatigue Choice Phase that they rejected in the Baseline Choice Phase (i.e., more risk-seeking behavior); and -1 was assigned if the participant rejected an effort gamble in the Fatigue Choice Phase that they accepted in the Baseline Choice Phase (i.e., more risk-averse behavior). In this way, we can look at the choice similarity metric within choice blocks (10 choices/block), over the course of the Fatigue Choice Phase to see how participants' choices change compared to Baseline. Below we have included a

plot across the $N = 20$ participants, indicating the mean and standard error of the choice similarity metric across exertion blocks.

Supplementary Figure 3. Choice similarity metric over the course of the Fatigue Choice Phase in the main experiment. The plot shows mean choice similarity (see Supplementary Methods for details) across participants for each of the 17 choice blocks after the initial fatiguing exertion block. On average, participants' choice similarity metric was negative (mean choice similarity: -0.06 ($SD = 0.02$); two-tailed one-sample t-test against the null hypothesis that choice similarity is zero: $t_{16} = -10.32$, $p < 0.001$), indicative of more risk averse decision-making, consistent with what we observed in our primary effort utility analysis. Notably, there was not a significant effect of exertion block on choice similarity (results of a general linear model on the effect of exertion block on choice similarity: $\beta_{\text{Block}} = -0.001$, $p = 0.54$) suggesting that after the initial bout of physical fatigue, participants' change in subjective preference remains relatively constant throughout the Fatigue Choice Phase. Error bars indicate SEM.

After the initial exertion block, which was intended to induce fatigue before any choices were made in the Fatigue Choice Phase, the average choice similarity metric was negative. This is indicative of participants exhibiting more risk-averse effort-choice behavior while fatigued. Over all exertion blocks, the mean choice similarity metric was significantly less than zero (mean choice similarity metric across blocks: -0.06 ($SD = 0.02$); two-tailed one-sample t-test against the null hypothesis that choice similarity is zero: $t_{16} = -10.32$, $p < 0.001$), which aligns with the fatigue-induced increases in ρ identified in our main effort utility analysis. Notably, there was not a significant effect of exertion block on the choice similarity metric (results of a general linear model on the effects of exertion block on the similarity metric: $\beta_{\text{Block}} = -0.001$, $p = 0.54$) suggesting that participants' subjective preference remained relatively constant throughout the Fatigue Choice Phase. Notably, this stable choice preference persisted even though participants' levels of exertion diminished over exertion blocks (Figure 2b). These results suggest that once a threshold of fatigue is reached, subjective preferences for effort remain relatively stable. We have now included this analysis in the Supplementary Information (Supplementary Figure 3, Supplementary Experimental Methods).

We also performed a new control experiment in which we fatigued participants at different levels of exertion sequentially (10U, 60U, and 10U again) to examine how effort preferences shift in response to different levels of exertion. We also acquired self-reported ratings of fatigue and physiological measures of muscle activation during exertion. Interestingly, we found a similar persistent heightening of effort valuation after participants exerted 60U of effort such that even after a period of rest and while exerting lower effort (10U), participants' subjective valuation of effort remained elevated. These heightened effort valuations were coincident with increased self-report ratings of fatigue and fatigue-related changes in muscle physiology. These control results support the idea that the effects of fatigue on subjective valuation of effort may persist even as the capacity for exertion varies.

We have outlined these new results in the Results section (page 17, paragraph 3).

Different Levels of Exertion Modulate Ratings of Fatigue, Muscle Activation, and Effort Preferences

While our main experiment was designed to examine how changes in fatigue influenced effort valuation, we did not directly poll participants' subjective ratings of effort to confirm that the fatiguing exertions did in fact influence perceptions of fatigue. Furthermore, the experiment did not collect any measures of muscle activation, so there was no data to show that exertions resulted in physiological signatures of physical fatigue. Moreover, the main experiment tested only one level of exertion (80U), making it unclear whether different levels of exertion modulate subjective effort preferences.

To address these limitations, we designed a comprehensive control experiment in which participants exerted different levels of effort, interspersed with blocks of effort choices, and were queried on their feelings of fatigue throughout the experiment (Figure 5a). We also monitored muscle activity with electromyography (EMG) to confirm that repeated exertions lead to physiological signatures of muscle fatigue. During the experiment, participants first completed a Baseline Choice Phase as in the main experiment (Figure 5b). Following this phase, participants performed a Modified Fatigue Choice Phase in which they first alternated between blocks of low effort (10U) exertion trials and blocks of effort choices. This was followed by alternating blocks of high effort (60U) exertion trials and blocks of effort choices. After this high effort section of the experiment, participants were given a two-minute rest period, and subsequently alternated between a second set of low effort (10U) exertion trials and choice blocks. At the beginning and end of each exertion block of the Modified Fatigue Choice Phase, we queried participants' self-report feelings of fatigue. For this control experiment, we also introduced a more stringent success criteria to ensure that a consistent number of exertions were performed within and across participants. Participants were required to perform five successful exertions (the successful trials need not be consecutive) before progressing to a block of effort choices. Overall, this experimental design allowed us to assess how self-reported measures

of fatigue, changes in muscle physiology, and effort preferences were influenced by experimentally varying levels of exertion.

Participants performed a relatively consistent number of repetitions (~ 5 trials) across effort exertion blocks (Figure 6a) (similarly to Figure 2a, one participant was excluded from this figure because the number of exertion trials in their sixth block – the first block of the 60U section – was greater than 3 SD above the group mean, but is included in all subsequent analyses). The first exertion block of the 60U exertion session had an increased number of repetitions because participants were not cued to when the exertion blocks switched from low to high effort exertions. The first block of exertion trials in the new section were intentionally unanticipated, resulting in increased variability in performance. However, participants quickly understood the exertion criteria of the section and their behavior became more stereotypical. Participants successfully reached the target levels of exertion in the low and high exertion blocks, and maintained consistent levels of mean exertion matching the cued effort levels of a given effort block (Figure 6b). Together these results indicate that the modified success criteria resulted in consistent, and experimentally controlled, amounts of exertion for both the low and high effort sections of the Modified Fatigue Choice Phase.

When initially performing low effort exertions, participants did not self-report as feeling fatigued, indicating that the mere presence of exertion trials did not induce feelings of fatigue (Figure 6c; average self-reported rating during the first low effort (10U) exertion section: 1.81 (SD = 0.66); results of a two-tailed one-sample t-test against the null hypothesis that the average self-reported rating in the first low effort (10U) exertion section was 3 (corresponding to “*Unsure*”): $t_{16} = 7.48$, $p < 0.001$). When performing repeated high effort exertions, participants’ fatigue ratings increased (mean increase in within-block self-reported rating between the first low effort (10U) and high effort (60U) sections: 1.14 (SD = 0.71), two-tailed paired-sample t-test comparing the average ratings between first low effort and high effort sections: $t_{16} = 6.58$, $p < 0.001$). Furthermore, an increase in fatigue ratings persisted even after the two-minute rest period, as well as during the second low effort exertion blocks (comparing the average within-block self-reported fatigue ratings between the first and second sections of low effort exertions with a two-tailed paired-sample t-test: $t_{16} = 5.69$, $p < 0.001$). These data show that after repeated high effort exertions, participants’ self-reported feelings of fatigue persist even after a period of rest and when performing subsequent low-effort exertions.

To assess if muscle activity reflected physical fatigue across different exertion sections in the Modified Fatigue Choice Phase, we compared the mean frequency of the power spectral density of the EMG signal between the average of successful trials in the first and second low effort exertion sections. Muscle fatigue is associated with a down-shift in the mean frequency of the power spectrum of the

EMG signal (when exerting the same level of effort)^{36–38}. We found a significant decrease in the mean frequency of the power spectrum between the first and second low effort (10U) exertion sections (Figure 6d; average change in mean frequency between the first and second sections of low effort trials: -4.39 Hz; two-tailed paired-sample t-test: $t_{16} = -2.25$, $p = 0.04$), consistent with the idea that repeated effortful exertions result in muscle fatigue.

To examine how effort preferences were modulated by effort level we used a Hierarchical Bayesian approach to estimate subjective effort parameters (ρ , τ) from choices in the first low effort, high effort, and second low effort sections of the experiment, providing insight into the differences between the preferences in these sections. Hierarchical Bayesian fitting allowed us to utilize data from all experimental sections in a single model, to detect choice signals that might otherwise have been difficult to discern given the reduced number of trials that were required at each exertion level (see Methods for details). We found that there was a significant increase in subjective effort parameters between choices in the first low effort and high effort sections (95% highest-density interval for the Bayesian posteriors of $\rho_{60U} - \rho_{10U,1}$ excludes zero; Figure 6e), reflecting an increase in the marginal cost of effort with increasing levels of exertion. These results show that effort parameters are modulated by the level of effortful exertion and are in concert with participants' increased self-report ratings of fatigue between the first low effort section and the high effort section. Interestingly, this shift of effort preferences persisted in the second low effort section of the experiment (95% highest-density interval for the Bayesian posteriors of $\rho_{10U,2} - \rho_{10U,1}$ excludes zero), which aligns with our observation that participants' self-reported ratings of fatigue remained elevated in the second low effort section of the experiment, despite a period of rest and a decrease in the target level of exertion. Together these findings illustrate that increasing levels of fatiguing exertion result in increased marginal utility of effort, fatigue-induced changes in muscle physiology, and increased self-report ratings of fatigue.

Two new figures pertaining to this control experiment have also been added (Figures 5 and 6).

Figure 5. Experimental Design of Control Experiment 2.

a, At the beginning and end of each exertion block, participants were queried on their level of agreement (“*Strongly Disagree*”, “*Somewhat Disagree*”, “*Unsure*”, “*Somewhat Agree*”, “*Strongly Agree*”) with the statement “I feel fatigued”. Participants were free to select anywhere on the scale (not just the indicated points), and there was no time constraint for them to select their rating.

b, Experiment schedule. The control experiment was divided into Baseline and Modified Fatigue Choice phases. The Baseline Choice Phase consisted of the same 170 effort-based choices used in the main experiment. Following the Baseline Choice Phase, participants performed the Modified Fatigue Choice Phase, in which they alternated between exertion trials and choice trials. The Modified Fatigue Choice Phase began with participants alternating between low effort exertions (10U; light gray) and blocks of choice trials (white). Following this sequence of exertion/choice, participants alternated between high effort exertions (60U; dark gray) and blocks of choice trials. Participants were then given a two minute period of rest in which they did not make any exertions or choice, and then alternated between low effort exertions and blocks of choice trials. Each exertion block lasted until participants exerted five successful trials by achieving the target level (n_i ; indicated the variable additional number of unsuccessful repetitions in an exertion block). At the beginning and end of each exertion blocks, participants were queried as to their feelings of fatigue as shown in a (indicated by asterisks).

Figure 6

Figure 6. Results from Control Experiment 2.

a, Exertion trial performance during the Modified Fatigue Choice Phase. Performance is represented as the mean repetitions required to successfully perform five exertions (successful trials need not be consecutive); higher values indicate more exertion trials were required in order to successfully exert at the target effort five times. Data shown in light gray represent low effort exertions (10U) whereas dark gray represents high effort exertions (60U). Participants were able to successfully exert at the required effort within ~5-6 trials for each block, indicating that performance was consistent across exertion levels. Error bars indicate SEM.

b, Mean exertion during the Modified Fatigue Choice Phase. Mean exertion is presented in units of effort. Data shown in light gray represent low effort exertions (10U) whereas dark gray represents high effort exertions (60U). Participants' motor performance did not diminish within sections of the Modified Fatigue Choice Phase (first low effort, high effort, and second low effort), but was appropriately modified in concert with the demands of the task. This suggests that participants' performance was consistent across exertion levels. Error bars indicate SEM.

c, Self-reported measures of participants' state of fatigue over the course of the Modified Fatigued Choice Phase. The plot shows mean self-reported ratings of fatigue at the beginning and end of the 17 exertion blocks (1 corresponds to "Strongly Disagree" and 5 to "Strongly Agree" with the statement "I feel fatigued"). At the start of the Modified Fatigued Choice Phase, participants do not self-report as feeling fatigued even after repeated exertions at low effort (10U; light gray). During repeated high effort exertions (60U; dark gray), participants reported feeling more fatigued and furthermore, this increase in self-reported fatigue persisted when the exertion task returned to low effort trials after a two minute rest (comparing the average within-block ratings between the first and second low effort sections with a two-tailed paired-sample t-test: $p < 0.001$, $t_{16} = 5.69$). Error bars indicate SEM.

d, Mean frequency of the power spectrum of EMGs between first and second low effort sections of the Modified Fatigue Choice Phase. Decreases in the mean frequency of the power spectrum of the EMG signal are physiological reflections of muscle fatigue. The plot shows mean frequency measures from EMGs targeting the *flexor digitorum superficialis* for all successful exertions averaged within the first and second low effort (10U) sections. Mean frequency of the EMG power spectrum decreased between first and second low effort sections (average change in mean frequency: -4.39 Hz; two-tailed paired-sample t-test: $t_{16} = -2.25$, $p = 0.04$), consistent with the idea that repeated effortful exertions resulted in muscle fatigue. Error bars indicate SEM. * $p < 0.05$.

e, Histograms of the parameter-space posteriors from the Hierarchical Bayesian model estimating changes in effort subjectivity parameters ($\Delta\rho$) between the different sections of the Modified Fatigue Choice Phase. The population exhibited an increase in effort subjectivity parameters (ρ) during choices intermixed with high effort exertions, as compared to the first session of 10U low effort exertions (60U; $\rho_{60U} - \rho_{10U,1}$; dark gray). This shift in effort subjectivity remained even after returning to low effort exertions (10U; $\rho_{10U,2} - \rho_{10U,1}$; light gray). The solid black lines indicate the bounds of the 95% highest-density interval for each distribution, both of which exclude 0 (indicated with the dashed black line) – which suggests a significant change in effort subjectivity parameters between sections of the Modified Fatigue Choice Phase. These results show that effort preferences are modulated by the level of fatiguing exertion and are coincident with self-report ratings and physiological signatures of fatigue.

The new methods for this control experiment have been added to the Methods section (page 33, paragraph 2; page 39, paragraph 4; page 48, paragraph 2).

Control Experiment 2

We performed an additional control experiment to test that our fatigue paradigm imparted self-reported increases in feelings of fatigue, associated changes in muscle physiology, and to further examine how choice preferences were modulated by the level of fatiguing exertion. A group of 21 healthy right-handed participants, separate from those in either of the previous two experiments, took part in this experiment. Two participants were unable to complete the experiment after exceeding the specified failure threshold (see below), and were therefore not considered for analysis. Of the remaining participants, one was excluded because they did not generate a salient association between effort levels and applied effort (r-squared between reported and actual effort during the Recall Phase was less than 0.5); another was excluded because their percentage of accepted effort gambles during the Baseline Choice Phase, was beyond two standard deviations of the mean proportion of acceptance. The final analysis for this experiment included a total of $N = 17$ participants (mean age, 26 years; age range, 21-37 years; 11 females).

Prior to the experiment, participants were informed that they would receive a fixed show-up fee of \$50 if they were able to complete the experiment (and \$10 otherwise). It was made clear, prior to making any decisions about prospective effort, that this fee did not depend on their choices. The control experiment progressed similarly to the main experiment, first with acquisition of participants' MVC, followed by Association and Recall Phases, and a Baseline Choice Phase. However for this control, we modified the Fatigue Choice Phase to test how varying levels of fatiguing exertion influence self-reported ratings of fatigue, muscle physiology, and subjective valuation of effort.

During the Modified Fatigue Choice Phase participants alternated between blocks of prospective effort choices and different levels of repeated physical exertion. As in the main experiment, exertion trials consisted of the 4 s presentation of a black horizontal bar, which participants were instructed to fill by gripping the transducer. However, during this Modified Fatigue Choice Phase, the amount of force required to completely fill the bar was either 10 (low effort) or 60 (high effort) units of effort. We also introduced a more stringent success criteria to ensure that a consistent amount of fatiguing exertion trials were performed within and across participants. Participants were required to perform five successful exertions (the successful trials need not be consecutive) before being able to progress. Furthermore, if the total number of failed exertion trials (across all exertion blocks) exceeded 20, the experiment ended. Participants were informed prior to the start of the phase that repeated failure to squeeze at the required level would result in the premature termination of the experiment and that they should try their best to succeed on each exertion trial.

Participants first performed a low effort section, which alternated between blocks of low effort exertion trials (five blocks) and blocks of choices (five blocks), with each block composed of 10 trials. This was followed by a high effort section which alternated between blocks of high effort exertion trials (six blocks) and blocks of effort choices (six blocks). After this high effort section of the experiment, participants were given a two minute period of rest, and subsequently performed a second low effort section where they once more alternated between low effort exertion trials (six blocks) and blocks of effort choices (six blocks). The effort options in the Modified Fatigue Choice Phase were pseudo-randomly extracted from the same choice set used in the Baseline Choice Phase and the main experiment, to ensure that options in each choice block sampled a range of gamble and sure values. Overall, this experimental design allowed us to assess how effort preferences were influenced by varying levels of exertion.

To obtain self-report measures of fatigue, before and after each exertion block, participants were queried on their level of agreement (“*Strongly Disagree*”, “*Somewhat Disagree*”, “*Unsure*”, “*Somewhat Agree*”, “*Strongly Agree*”) with the statement “*I feel fatigued*”. Participants were free to select anywhere on the scale (not just the indicated points), and there was no time constraint for them to select their rating.

Throughout all exertion trials (for MVC, Association, Recall, and Modified Fatigue Choice Phases), we examined muscle activations using surface electromyograms (sEMGs). Three disposable electrodes (NeuroPlus™ A10040 Electrodes; Vermed.com, Buffalo, NY) recorded muscle activity targeting the right *flexor digitorum superficialis* muscle using a method for standardized EMG electrode placement⁴¹, which has been previously used to study hand grip exertion³⁷. EMG

signals were amplified (AMT-8; Bortec Biomedical Ltd., Calgary, Alberta, Canada) and bandpass filtered with high- and low-pass cutoff frequencies of 10 and 1000 Hz, and additionally filtered with a 60 Hz notch-filter. Signals were sampled at 5 kHz by a 16-bit data acquisition system (CED Micro1401-3; Cambridge Electronic Design Ltd., Cambridge, England). EMG acquisition was triggered prior to the onset of an exertion trial and encompassed the full 4 second exertion interval.

This control experiment has several features that are important to mention. First, since we experimentally modulated the level of effort required during fatiguing exertion blocks of the Modified Fatigue Choice Phase, and introduced a more stringent success criteria for exertion trials, we could more precisely modulate participants' state of fatigue to examine how different levels of exertion impact valuation of effort. Second, since we acquired measures of participants' self-reported ratings of fatigue we could confirm that participants do in fact feel fatigued following exertion, and were not simply apathetic as it pertains to participation in the experiment. Third, recording of muscle activity during effortful exertions allowed us to confirm that the paradigm elicits physiological changes in muscle activity associated with fatigue.

Hierarchical Bayesian Effort Choice Analysis

For the second control experiment, we fit a prospect theory-inspired model of the process underlying valuation and choice. The basic model $V(x)$ was the same as that used in the main experiment. However, we used a hierarchical Bayesian approach to fit this model. This method gives us a statistical advantage by explicitly modeling and fitting parameters at the level of the participant (e.g., participant's preferences when making choices interspersed with low and high effort exertions) as well as at the level of the group (e.g., the mean population effort preferences). Using such a model, and fitting all participants' data simultaneously, reduced the influence of outliers and noise, and maximizes the ability to detect experimental fluctuations in effort preferences. This procedure also has the benefit of allowing us to directly model the effect of interest: the influence of low and high effort fatiguing exertions on changes in the valuation and decision processes, at the population level. Notably, each of the low/high effort sections in the Modified Fatigue Choice Phase of the control experiment has fewer trials (50/60/60) than the main experiment (170) which makes it beneficial to use such a Hierarchical Bayesian approach, since it leverages all the choice data and maximizes the possibility of identifying choice 'signals' of interest. Importantly, because of the limited set of choices in each fatiguing section, this methodology allows us to estimate parameters in sparsely sampled regions by modelling intrinsic structure within and between participants.

The underlying model used to fit subjectivity parameters (ρ and τ) from the choice data was structured similarly to a general linear mixed effects random intercept model, allowing for participant-specific intercepts but estimating population level

effects of the experimental section. Parameters from the Baseline Choice Phase and first 10U section (10U,1) of the Modified Fatigue Choice Phase are estimated in the following form:

$$\begin{aligned} \rho_i^S &= A * \Phi(r_i^S) \\ \tau_i^S &= B * \Phi(t_i^S) \end{aligned} \quad \begin{bmatrix} r_i^{Bas} \\ t_i^{Bas} \\ r_i^{10U,1} \\ t_i^{10U,1} \end{bmatrix} \sim \mathcal{N}(M, \Sigma)$$

In this formulation, both ρ and τ for participant (i) and experimental section (S) (Baseline, first low effort section (10U,1)), are represented and estimated as a participant-specific parameter (r and t), drawn from parameter-specific normal distributions with estimated means (M) and covariance matrix (Σ), which are estimated population-level hyper-parameters. Φ is the unit Gaussian cumulative distribution function bounding the supported parameters to (0,1), where A and B reflect imposed bounding constraints on the resulting estimates⁴³.

Changes in effort subjectivity parameters for the other two sections of the Modified Fatigue Choice Phase are modeled with a population-level section-specific (60U and 10U,2) shared offset (δ), which takes the following form:

$$\begin{aligned} \rho_i^S &= A * \Phi(r_i^{10U,1} + \delta_r^S) \\ \tau_i^S &= B * \Phi(t_i^{10U,1} + \delta_t^S) \end{aligned}$$

The model then generates posterior estimates of the population's distributions for M , Σ , δ^{60U} , and $\delta^{10U,2}$, which can be transformed into parameter-space using the above expressions. The probability that a participant chooses the sure option for the k^{th} trial is given by the softmax function (in which RV_{sure} is now contingent upon participant and exertion section-specific parameters, drawn from the population distribution):

$$P_k(RV_{sure}(G, S)) = 1/[1 + \exp(-\tau RV_{sure}(G, S))]$$

This parameter estimation procedure was implemented using Monte-Carlo Markov Chain sampling methods provided by Stan version 2.19⁴⁴ and implementing a similar methodology as described by the hBayesDM package⁴³. Standard hierarchical Bayesian methods were used, with constraints on the fit parameters of $\rho \in [0, 10]$ and $\tau \in [0, 6]$, and weakly informative distributions were chosen for the parameter priors in order to facilitate model convergence⁴⁵.

EMG Analysis

In order to compare EMG signals from homogenous exertion profiles, only successful exertion trials (spending at least 2.67 seconds of the exertion time within ± 5 effort levels of the target) were considered. We considered the mean frequency of the power spectrum of the sEMG signal as a physiological measure of fatigue, which has been well documented in numerous EMG studies investigating fatigue³⁶⁻³⁸. Mean frequency measures for each trial were computed via the MATLAB 'meanfreq' function examining a frequency interval of 10-500 Hz. To examine differences between the first 10U and second 10U sections of the Modified Fatigue Choice Phase, we averaged mean frequency measures across all successful trials in each section within each participant, and compared these values between the two sections (Figure 6d).

1.11) I do not get the difference between model 1 and model 5 in the comparison shown in supplementary material. Having $v = x^p$ with an inverse temperature in the logistic regression model is the same thing as having $v = kx^p$. I suspect model 5 is just model 1 with a redundant parameter, which would explain why the AIC is higher.

The reviewer is correct that models 1 and 5 in the model comparison are functionally equivalent. We have now removed model 5 from the model comparison since it is a redundant model.

Supplementary Figure 6. Model comparison of different effort utility functions. We performed a series of maximum likelihood estimations using effort utility models that have been implemented in previous studies of effort-based decision-making (Klein-Flügge et al. 2016; Chong et al. 2017; Hogan et al. 2018), to assess which utility function best captured fatigue-induced changes in subjective effort valuation. Group-level AIC measures are shown above. Lower AIC measures indicate a more descriptive model. The utility function x^p best described the choice data across conditions. It should be noted that the difference in group-level

AIC between the two best models (x^ρ , $kx^{\rho*}$) was 189.7. Details of this analysis are described in the Supplementary Experimental Methods section.

1.12) I am not sure about which model was fitted for the illustration in Figure 2c. It looks like a logistic function with both slope and bias parameters, but what we want to see is the model used to analyze fMRI data. This should be clarified. Besides, it would be more informative to show a group average and not just a (best) typical subject.

The reviewer raises a good point that Figure 2c (in the previous draft of the paper) could lead to confusion. As the reviewer mentioned this was not the model used to analyze the imaging data. We included these representative logistic fits to illustrate that fatigue leads to a shift of the choice indifference point, indicative of increased risk aversion for effort while in a fatigued state. To eliminate this confusion we have removed this figure and instead included effort cost functions, for the Baseline and Fatigue Choice Phases, averaged across participants.

Figure 2c, The function used to model the subjective cost of effort. This function has the form $V(x) = -(-x)^\rho$. Effort cost functions using mean values of the ρ estimates are indicated by the solid lines (Baseline: light gray; Fatigue: dark gray), with SEM indicated by the shaded regions. Undergoing fatiguing exertions increases the marginal cost of effort. In order to better illustrate the cost functions, the x- and y-axes shown are not to the same scale.

1.13) It is nice to have a control experiment, for the effect or time or repetition of choices, but we need a significant difference with the main experiment, not just an absence of effect, which may be due to low statistical power ($n = 7$).

We have now included 2 additional participants in the control group for Control Experiment 1. For the main group and the control group, we calculated a choice similarity measure that assesses the extent participants' choices changed between the repeated Choice Phases. We found a

significant difference between the choice similarity metric in the control and main experimental groups (unpaired t-test comparing choice similarity metrics between the control and main experimental groups: $t_{27} = 2.38$, $p = 0.02$), indicating that fatiguing exertions in the main experiment group induce changes in effort preferences that are specific to the Fatigue paradigm, and not simply a reflection of mere exposure to the effort options.

Supplementary Figure 5. Comparison of choice similarity metrics (see Supplementary Methods for details) between control experiment 1 and the primary fatigue experiment, averaged across choices. Relative to the Baseline Choice Phase, more positive values indicate more risk-seeking behavior in the second choice phase, whereas negative values indicate more risk-averse behavior. Participants' choice behavior in the control experiment was not significantly different comparing the first and second choice phases (two-tailed one-sample $t_8 = 1.40$, $p = 0.20$), whereas choice behavior between the Baseline and Fatigue Choice Phase was significantly different (two-tailed one-sample $t_{19} = -2.59$, $p = 0.02$), consistent with our observed fatigue-induced increase in effort subjectivity parameter ρ . The results of a two-tailed two-sample t-test comparing these choice similarity metrics, between experimental groups indicates that these effects are significantly different ($t_{27} = 2.38$, $p = 0.02$), supporting the idea that the observed change in choice behavior in our main experiment is not merely an effect of choice exposure alone.

We have referenced this analysis in the main text (page 11, paragraph 3).

We have included the description of this choice similarity metric and analysis in the Supplementary Information (page 14, paragraph 1).

Choice Similarity Metric

As a secondary method to investigate how risk attitudes for effort change between conditions, we compared choices between the two phases by computing a choice similarity metric. This metric is essentially model-free, in that it does not assume an effort utility function and does not require the fitting of a model to the behavioral data. Since each effort gamble was presented twice (once per condition), it is possible to examine if choice behavior for identical effort options changed between experimental phases. To generate this metric, a value of 0 was assigned to a choice trial in the Fatigue Choice Phase if the participant made the same choice as in the Baseline Choice Phase; +1 was assigned to a choice if the participant accepted an effort gamble in the Fatigue Choice Phase that they rejected in the Baseline Choice Phase (i.e., more risk seeking behavior); and -1 was assigned if the participant rejected an effort gamble in the Fatigue Choice Phase that they accepted in the Baseline Choice Phase (i.e., more risk averse behavior).

We also performed a new control experiment in which we fatigued participants at different levels of exertion sequentially (10U, 60U, and 10U again) to examine how effort preferences shift in response to different levels of exertion (described in comment 1.10). In this control experiment, we also acquired self-reported ratings of fatigue and physiological measures of muscle activation during exertion. We found that effort preferences significantly increased between the first 10 unit exertion section and the 60 unit section, suggesting that effort preferences are modulated by the level of fatiguing exertion. Moreover, in the second 10 unit exertion section, we observed a persistent heightening of effort preferences. These increases in effort preferences were coincident with increased self-report ratings of fatigue and fatigue-related changes in muscle physiology. Overall, the results from this second control experiment illustrate that subjective preferences for effort, increases in subjective feelings of fatigue, and fatigue-induced changes in muscle physiology, are all modulated by the level of fatiguing exertion.

1.14) The exclusion of apathy based on RT is somewhat arbitrary. One could equally say that increasing RT would reflect fatigue and not apathy. I would remove this part.

At the reviewer's suggestion, we have removed this discussion of apathy from the manuscript.

1.15) The fMRI results are rather weak and heavily rely on ROI. In that regard I was surprised not to see the vmPFC, which was central in the authors' previous study.

Our whole-brain contrasts were displayed at $p < 0.005$ with a 10-voxel extent threshold. Statistical inference was performed using small-volume correction within *a priori* regions of interest, independently identified from a database of previous studies of effort-based decision-making (Neurosynth.org). These are standard methods used in affective neuroimaging (Lieberman and Cunningham 2009).

The reviewer makes an excellent point that it is important to interpret our results in the context of our previous study of prospective effort valuation (Hogan et al. 2018), which showed that

vmPFC activity was related to subjective effort valuation in a rested state. We have tested an independent vmPFC ROI, used in our previous study of effort-based decision-making and failed to find significant vmPFC activity related to prospective effort valuation in the present study.

One important distinction between the current study and our previous work (Hogan et al. 2018) is that participants in the current study were performing bouts of exertion interspersed with choices, whereas our previous study completely isolated the processes of exertion and valuation. Notably, a number of contemporary studies of prospective effort valuation have completely separated choice and exertion (Arulpragasam et al. 2018; Hogan et al. 2018; Aridan et al. 2019), and have reported effort value signals in vmPFC as opposed to ACC. In contrast, the studies which have found value signals in ACC and insula involved effort choices interspersed with exertion (Croxson et al. 2009; Prévost et al. 2010; Kurniawan et al. 2013; Skvortsova et al. 2014; Bonnelle et al. 2016; Klein-Flügge et al. 2016) as in our current experiment. Taken together, these results may indicate that signals in ACC and insula could be related to the interactions of effort valuations with motor preparation preceding exertion, while vmPFC might be more related to prospective valuation of effort alone. However, our present study was not designed to dissociate such preparatory and prospective value signals. We have now included a discussion of the lack of vmPFC activity in the context of our present study (page 24, paragraph 3).

Recent studies of prospective effort valuation have separated the timing of effortful exertion from prospective valuation and decision-making about effort⁸⁻¹⁰, and showed that in this context the ventromedial prefrontal cortex (vmPFC) encodes decision-values for prospective effort. In contrast, many earlier studies of effort valuation¹⁻⁶ had intermixed exertions and effort choices, and these studies reliably identified a network of brain activity that included the ACC and bilateral insula. It has been suggested that in such experimental designs, ACC and insula activity could be related to preparatory motor signals that integrate effort value^{5,7,10}. It is important to mention that in a previous experiment that employed a similar paradigm, which completely separated decision-making and exertion, we found that vmPFC encoded effort value⁹. In contrast, participants in the current study performed fatiguing exertions that were intermixed with effort choices and we found that effort value was encoded in ACC and bilateral insula. With this in mind, it is possible that the context of the effort-based decisions in the current paradigm (in which choices were interspersed with fatiguing exertion trials) could have led to the differences in neural results between our previous study of effort valuation and our current study about fatigue.

1.16) It is problematic that the difference between chosen and unchosen value is not significant in baseline insular activity. One cannot conclude from this observation that the insula encodes effort cost in a normal state. What fatigue seems to be doing is increasing the sensitivity to effort cost.

We agree with the interpretation that insula appears to be increasing its sensitivity to effort cost while in a fatigued state. We have now carefully edited the results and discussion sections of our manuscript to be sure this idea is conveyed.

As mentioned in our earlier response, the previous studies that identified ACC and insular activity for effort values involved experimental designs in which exertions were interspersed with effort choices. However, the Baseline Choice Phase does not interleave choice and exertion blocks as in the Fatigue Choice Phase, therefore it is possible that the observed increase in effort cost sensitivity in rIns during the fatigue condition is a reflection of the neural interaction between effort valuation and motor preparation.

1.17) In the end, did subjects manage to perform the effort level that they had selected? My concern is that they may have avoided efforts that they believe they would not be able to reach, and not efforts that were too costly.

Participants were informed prior to making any effort choices that they would need to remain in the testing area until they had successfully performed the exertions associated with the outcomes of their decisions. With this in mind, participants continued to perform exertion trials until they accomplished the required exertion. Averaging across the outcome trials in which exertion was required, it took participants 1.17 (standard deviation: 0.18) trials to complete a required exertion level. This indicates that participants were able to successfully perform the chosen effort levels within one or two attempts.

We also analyzed the Association Phase data to observe how the ability to achieve the target effort levels was related to subjective effort preferences acquired during choice. We did not find a significant relationship between the level of success during the Association Phase and the baseline effort subjectivity parameter $\rho_{Baseline}$ ($r = -0.08$, $p = 0.75$), $\rho_{Fatigue}$ ($r = -0.18$, $p = 0.46$), or between percent success and $\rho_{Fatigue} - \rho_{Baseline}$ ($r = -0.18$, $p = 0.44$). Together these results suggest that the choice behavior was not simply a reflection of individual's performance during Association exertions, and instead related to the subjective valuation of effort.

REVIEWER 2

Major Comments:

2.1) The order of baseline choice phase and fatigue choice phase: after practice (Association phase + Recall phase), participants conducted baseline choice phase. Then, they were exposed to exertion trials and fatigue choice phase. Although the authors attributed behavioral changes in the fatigue choice phase to fatigue, I am not convinced of this conclusion. It is possible that participant's experience of exertion trials not fatigue changed his/her behavior. To rule out this possibility, the authors need to confirm at least the following:

- a) It is necessary to ask the same participants to do the baseline choice task a few hours or days after fatigue choice condition and compare the two baseline choice outcomes.
- b) It is preferable to split fatigue choice data into two parts, and compare the former and latter behaviors. If behavioral change truly reflects fatigue, the authors should be able to observe the shift in logistic curves similar to Fig. 2c.
- c) It is desirable to collect subjective evaluation of fatigue to justify that participants felt fatigue.

To confirm that changes in choice behavior following exertion were the result of fatigue, we performed a new control experiment in which participants exerted different levels of effort, interspersed with choice, and acquired self-report ratings of fatigue and physiological measures of muscle activation. This control experiment allowed us to directly test if changes in effort valuation are modulated by the level of exertion, and if these changes are coincident with changes in self-reported ratings of fatigue and changes in muscle physiology.

In this control experiment, participants exerted 10U, 60U, and 10U of effort sequentially, and each section of exertion trials was interspersed with blocks of exertion and choice. Before and after exertion blocks, we had participants rate their level of fatigue, and we acquired arm muscle EMG recordings to confirm that physiological signatures of fatigue were present. Participants' self-reported ratings of fatigue increased between the first 10U exertion section and the 60U exertion section, and relatedly we found that participants' subjective valuations of effort increased (i.e., increased ρ parameter). Interestingly, between the 60U exertion section and the second 10U exertion section, participants took a two minute rest, and while this resulted in their self-reported ratings of fatigue decreasing slightly, they still reported feeling fatigued. In concert with these increased feelings of fatigue, we found that participants' subjective preferences for effort remained elevated. We also observed a decrease in the mean frequency of the power spectrum of participants' sEMG recordings, when comparing the first and second 10U exertion sections, which is a hallmark of muscular fatigue. Together these results show that increased levels of exertion increase fatigue in participants (as indicated by self-reported ratings and muscle physiology), and that this fatigue is coincident with changes in effort-choice behavior).

This control experiment comprehensively addresses the reviewer's concerns by showing that self-reported ratings of fatigue are modulated by the exertion levels in our paradigm, and participants' subjective valuations of effort are modulated accordingly. Moreover, we show that

these changes in choice behavior and increased self-reported fatigue ratings occur in concert with physiological signatures of muscle fatigue.

We have outlined these new results in the Results section (page 17, paragraph 3).

Different Levels of Exertion Modulate Ratings of Fatigue, Muscle Activation, and Effort Preferences

While our main experiment was designed to examine how changes in fatigue influenced effort valuation, we did not directly poll participants' subjective ratings of effort to confirm that the fatiguing exertions did in fact influence perceptions of fatigue. Furthermore, the experiment did not collect any measures of muscle activation, so there was no data to show that exertions resulted in physiological signatures of physical fatigue. Moreover, the main experiment tested only one level of exertion (80U), making it unclear whether different levels of exertion modulate subjective effort preferences.

To address these limitations, we designed a comprehensive control experiment in which participants exerted different levels of effort, interspersed with blocks of effort choices, and were queried on their feelings of fatigue throughout the experiment (Figure 5a). We also monitored muscle activity with electromyography (EMG) to confirm that repeated exertions lead to physiological signatures of muscle fatigue. During the experiment, participants first completed a Baseline Choice Phase as in the main experiment (Figure 5b). Following this phase, participants performed a Modified Fatigue Choice Phase in which they first alternated between blocks of low effort (10U) exertion trials and blocks of effort choices. This was followed by alternating blocks of high effort (60U) exertion trials and blocks of effort choices. After this high effort section of the experiment, participants were given a two-minute rest period, and subsequently alternated between a second set of low effort (10U) exertion trials and choice blocks. At the beginning and end of each exertion block of the Modified Fatigue Choice Phase, we queried participants' self-report feelings of fatigue. For this control experiment, we also introduced a more stringent success criteria to ensure that a consistent number of exertions were performed within and across participants. Participants were required to perform five successful exertions (the successful trials need not be consecutive) before progressing to a block of effort choices. Overall, this experimental design allowed us to assess how self-reported measures of fatigue, changes in muscle physiology, and effort preferences were influenced by experimentally varying levels of exertion.

Participants performed a relatively consistent number of repetitions (~ 5 trials) across effort exertion blocks (Figure 6a) (similarly to Figure 2a, one participant was excluded from this figure because the number of exertion trials in their sixth block – the first block of the 60U section – was greater than 3 SD above the group mean, but is included in all subsequent analyses). The first exertion block of the 60U

exertion session had an increased number of repetitions because participants were not cued to when the exertion blocks switched from low to high effort exertions. The first block of exertion trials in the new section were intentionally unanticipated, resulting in increased variability in performance. However, participants quickly understood the exertion criteria of the section and their behavior became more stereotypical. Participants successfully reached the target levels of exertion in the low and high exertion blocks, and maintained consistent levels of mean exertion matching the cued effort levels of a given effort block (Figure 6b). Together these results indicate that the modified success criteria resulted in consistent, and experimentally controlled, amounts of exertion for both the low and high effort sections of the Modified Fatigue Choice Phase.

When initially performing low effort exertions, participants did not self-report as feeling fatigued, indicating that the mere presence of exertion trials did not induce feelings of fatigue (Figure 6c; average self-reported rating during the first low effort (10U) exertion section: 1.81 (SD = 0.66); results of a two-tailed one-sample t-test against the null hypothesis that the average self-reported rating in the first low effort (10U) exertion section was 3 (corresponding to “*Unsure*”): $t_{16} = 7.48, p < 0.001$). When performing repeated high effort exertions, participants’ fatigue ratings increased (mean increase in within-block self-reported rating between the first low effort (10U) and high effort (60U) sections: 1.14 (SD = 0.71), two-tailed paired-sample t-test comparing the average ratings between first low effort and high effort sections: $t_{16} = 6.58, p < 0.001$). Furthermore, an increase in fatigue ratings persisted even after the two-minute rest period, as well as during the second low effort exertion blocks (comparing the average within-block self-reported fatigue ratings between the first and second sections of low effort exertions with a two-tailed paired-sample t-test: $t_{16} = 5.69, p < 0.001$). These data show that after repeated high effort exertions, participants’ self-reported feelings of fatigue persist even after a period of rest and when performing subsequent low-effort exertions.

To assess if muscle activity reflected physical fatigue across different exertion sections in the Modified Fatigue Choice Phase, we compared the mean frequency of the power spectral density of the EMG signal between the average of successful trials in the first and second low effort exertion sections. Muscle fatigue is associated with a down-shift in the mean frequency of the power spectrum of the EMG signal (when exerting the same level of effort)^{36–38}. We found a significant decrease in the mean frequency of the power spectrum between the first and second low effort (10U) exertion sections (Figure 6d; average change in mean frequency between the first and second sections of low effort trials: -4.39 Hz; two-tailed paired-sample t-test: $t_{16} = -2.25, p = 0.04$), consistent with the idea that repeated effortful exertions result in muscle fatigue.

To examine how effort preferences were modulated by effort level we used a Hierarchical Bayesian approach to estimate subjective effort parameters (ρ, τ) from choices in the first low effort, high effort, and second low effort sections of the experiment, providing insight into the differences between the preferences in these sections. Hierarchical Bayesian fitting allowed us to utilize data from all experimental sections in a single model, to detect choice signals that might otherwise have been difficult to discern given the reduced number of trials that were required at each exertion level (see Methods for details). We found that there was a significant increase in subjective effort parameters between choices in the first low effort and high effort sections (95% highest-density interval for the Bayesian posteriors of $\rho_{60U} - \rho_{10U,1}$ excludes zero; Figure 6e), reflecting an increase in the marginal cost of effort with increasing levels of exertion. These results show that effort parameters are modulated by the level of effortful exertion and are in concert with participants' increased self-report ratings of fatigue between the first low effort section and the high effort section. Interestingly, this shift of effort preferences persisted in the second low effort section of the experiment (95% highest-density interval for the Bayesian posteriors of $\rho_{10U,2} - \rho_{10U,1}$ excludes zero), which aligns with our observation that participants' self-reported ratings of fatigue remained elevated in the second low effort section of the experiment, despite a period of rest and a decrease in the target level of exertion. Together these findings illustrate that increasing levels of fatiguing exertion result in increased marginal utility of effort, fatigue-induced changes in muscle physiology, and increased self-report ratings of fatigue.

Two new figures pertaining to this control experiment have also been added (Figures 5 and 6).

Figure 5. Experimental Design of Control Experiment 2.

a, At the beginning and end of each exertion block, participants were queried on their level of agreement ("Strongly Disagree", "Somewhat Disagree", "Unsure", "Somewhat Agree", "Strongly Agree") with the statement "I feel fatigued". Participants were free to select anywhere on the scale (not just the indicated points), and there was no time constraint for them to select their rating.

b, Experiment schedule. The control experiment was divided into Baseline and Modified Fatigue Choice phases. The Baseline Choice Phase consisted of the same 170 effort-based choices used in the main experiment. Following the Baseline Choice Phase, participants performed the Modified Fatigue Choice Phase, in which they alternated between exertion trials and choice trials. The Modified Fatigue Choice Phase began with participants alternating between low effort exertions (10U; light gray) and blocks of choice trials (white). Following this sequence of exertion/choice, participants alternated between high effort exertions (60U; dark gray) and blocks of choice trials. Participants were then given a two minute period of rest in which they did not make any exertions or choice, and then alternated between low effort exertions and blocks of choice trials. Each exertion block lasted until participants exerted five successful trials by achieving the target level (n_i indicated the variable additional number of unsuccessful repetitions in an exertion block). At the beginning and end of each exertion blocks, participants were queried as to their feelings of fatigue as shown in a (indicated by asterisks).

Figure 6

Figure 6. Results from Control Experiment 2.

a, Exertion trial performance during the Modified Fatigue Choice Phase. Performance is represented as the mean repetitions required to successfully perform five exertions (successful trials need not be consecutive); higher values indicate more exertion trials were required in order to successfully exert at the target effort five times. Data shown in light gray represent low effort exertions (10U) whereas dark gray represents high effort exertions (60U). Participants were able to successfully exert at the required effort within ~5-6 trials for each block, indicating that performance was consistent across exertion levels. Error bars indicate SEM.

b, Mean exertion during the Modified Fatigue Choice Phase. Mean exertion is presented in units of effort. Data shown in light gray represent low effort exertions (10U) whereas dark gray represents high effort exertions (60U). Participants' motor performance did not diminish within sections of the Modified Fatigue Choice Phase (first low effort, high effort, and second low effort), but was appropriately modified in concert with the demands of the task. This suggests that participants' performance was consistent across exertion levels. Error bars indicate SEM.

c, Self-reported measures of participants' state of fatigue over the course of the Modified Fatigued Choice Phase. The plot shows mean self-reported ratings of fatigue at the beginning and end of the 17 exertion blocks (1 corresponds to "Strongly Disagree" and 5 to "Strongly Agree" with the statement "I feel fatigued"). At the start of the Modified Fatigued Choice Phase, participants do not self-report as feeling fatigued even after repeated exertions at low effort (10U; light gray). During repeated high effort exertions (60U; dark gray), participants reported feeling more fatigued and furthermore, this increase in self-reported fatigue persisted when the exertion task returned to low effort trials after a two minute rest (comparing the average within-block ratings between the first and second low effort sections with a two-tailed paired-sample t-test: $p < 0.001$, $t_{16} = 5.69$). Error bars indicate SEM.

d, Mean frequency of the power spectrum of EMGs between first and second low effort sections of the Modified Fatigue Choice Phase. Decreases in the mean frequency of the power spectrum of the EMG signal are physiological reflections of muscle fatigue. The plot shows mean frequency measures from EMGs targeting the *flexor digitorum superficialis* for all successful exertions averaged within the first and second low effort (10U) sections. Mean frequency of the EMG power spectrum decreased between first and second low effort sections (average change in mean frequency: -4.39 Hz; two-tailed paired-sample t-test: $t_{16} = -2.25$, $p = 0.04$), consistent with the idea that repeated effortful exertions resulted in muscle fatigue. Error bars indicate SEM. * $p < 0.05$.

e, Histograms of the parameter-space posteriors from the Hierarchical Bayesian model estimating changes in effort subjectivity parameters ($\Delta\rho$) between the different sections of the Modified Fatigue Choice Phase. The population exhibited an increase in effort subjectivity parameters (ρ) during choices intermixed with high effort exertions, as compared to the first session of 10U low effort exertions (60U; $\rho_{60U} - \rho_{10U,1}$; dark gray). This shift in effort subjectivity remained even after returning to low effort exertions (10U; $\rho_{10U,2} - \rho_{10U,1}$; light gray). The solid black lines indicate the bounds of the 95% highest-density interval for each distribution, both of which exclude 0 (indicated with the dashed black line) – which suggests a significant change in effort subjectivity parameters between sections of the Modified Fatigue Choice Phase. These results show that effort preferences are modulated by the level of fatiguing exertion and are coincident with self-report ratings and physiological signatures of fatigue.

The new methods for this control experiment have been added to the Methods section (page 33, paragraph 2; page 39, paragraph 4; page 48, paragraph 2).

Control Experiment 2

We performed an additional control experiment to test that our fatigue paradigm imparted self-reported increases in feelings of fatigue, associated changes in muscle physiology, and to further examine how choice preferences were modulated by the level of fatiguing exertion. A group of 21 healthy right-handed participants, separate from those in either of the previous two experiments, took part in this experiment. Two participants were unable to complete the experiment after exceeding the specified failure threshold (see below), and were therefore not considered for analysis. Of the remaining participants, one was excluded because they did not generate a salient association between effort levels and applied effort (r-squared between reported and actual effort during the Recall Phase was less than 0.5); another was excluded because their percentage of accepted effort gambles during the Baseline Choice Phase, was beyond two standard deviations of the mean proportion of acceptance. The final analysis for this experiment included a total of $N = 17$ participants (mean age, 26 years; age range, 21-37 years; 11 females).

Prior to the experiment, participants were informed that they would receive a fixed show-up fee of \$50 if they were able to complete the experiment (and \$10 otherwise). It was made clear, prior to making any decisions about prospective effort, that this fee did not depend on their choices. The control experiment progressed similarly to the main experiment, first with acquisition of participants' MVC, followed by Association and Recall Phases, and a Baseline Choice Phase. However for this control, we modified the Fatigue Choice Phase to test how varying levels of fatiguing exertion influence self-reported ratings of fatigue, muscle physiology, and subjective valuation of effort.

During the Modified Fatigue Choice Phase participants alternated between blocks of prospective effort choices and different levels of repeated physical exertion. As in the main experiment, exertion trials consisted of the 4 s presentation of a black horizontal bar, which participants were instructed to fill by gripping the transducer. However, during this Modified Fatigue Choice Phase, the amount of force required to completely fill the bar was either 10 (low effort) or 60 (high effort) units of effort. We also introduced a more stringent success criteria to ensure that a consistent amount of fatiguing exertion trials were performed within and across participants. Participants were required to perform five successful exertions (the successful trials need not be consecutive) before being able to progress. Furthermore, if the total number of failed exertion trials (across all exertion blocks) exceeded 20, the experiment ended. Participants were informed prior to the start of the phase that repeated failure to squeeze at the required level would result in the premature termination of the experiment and that they should try their best to succeed on each exertion trial.

Participants first performed a low effort section, which alternated between blocks of low effort exertion trials (five blocks) and blocks of choices (five blocks), with each block composed of 10 trials. This was followed by a high effort section which alternated between blocks of high effort exertion trials (six blocks) and blocks of effort choices (six blocks). After this high effort section of the experiment, participants were given a two minute period of rest, and subsequently performed a second low effort section where they once more alternated between low effort exertion trials (six blocks) and blocks of effort choices (six blocks). The effort options in the Modified Fatigue Choice Phase were pseudo-randomly extracted from the same choice set used in the Baseline Choice Phase and the main experiment, to ensure that options in each choice block sampled a range of gamble and sure values. Overall, this experimental design allowed us to assess how effort preferences were influenced by varying levels of exertion.

To obtain self-report measures of fatigue, before and after each exertion block, participants were queried on their level of agreement (“*Strongly Disagree*”, “*Somewhat Disagree*”, “*Unsure*”, “*Somewhat Agree*”, “*Strongly Agree*”) with the statement “*I feel fatigued*”. Participants were free to select anywhere on the scale (not just the indicated points), and there was no time constraint for them to select their rating.

Throughout all exertion trials (for MVC, Association, Recall, and Modified Fatigue Choice Phases), we examined muscle activations using surface electromyograms (sEMGs). Three disposable electrodes (NeuroPlus™ A10040 Electrodes; Vermed.com, Buffalo, NY) recorded muscle activity targeting the right *flexor digitorum superficialis* muscle using a method for standardized EMG electrode placement⁴¹, which has been previously used to study hand grip exertion³⁷. EMG

signals were amplified (AMT-8; Bortec Biomedical Ltd., Calgary, Alberta, Canada) and bandpass filtered with high- and low-pass cutoff frequencies of 10 and 1000 Hz, and additionally filtered with a 60 Hz notch-filter. Signals were sampled at 5 kHz by a 16-bit data acquisition system (CED Micro1401-3; Cambridge Electronic Design Ltd., Cambridge, England). EMG acquisition was triggered prior to the onset of an exertion trial and encompassed the full 4 second exertion interval.

This control experiment has several features that are important to mention. First, since we experimentally modulated the level of effort required during fatiguing exertion blocks of the Modified Fatigue Choice Phase, and introduced a more stringent success criteria for exertion trials, we could more precisely modulate participants' state of fatigue to examine how different levels of exertion impact valuation of effort. Second, since we acquired measures of participants' self-reported ratings of fatigue we could confirm that participants do in fact feel fatigued following exertion, and were not simply apathetic as it pertains to participation in the experiment. Third, recording of muscle activity during effortful exertions allowed us to confirm that the paradigm elicits physiological changes in muscle activity associated with fatigue.

Hierarchical Bayesian Effort Choice Analysis

For the second control experiment, we fit a prospect theory-inspired model of the process underlying valuation and choice. The basic model $V(x)$ was the same as that used in the main experiment. However, we used a hierarchical Bayesian approach to fit this model. This method gives us a statistical advantage by explicitly modeling and fitting parameters at the level of the participant (e.g., participant's preferences when making choices interspersed with low and high effort exertions) as well as at the level of the group (e.g., the mean population effort preferences). Using such a model, and fitting all participants' data simultaneously, reduced the influence of outliers and noise, and maximizes the ability to detect experimental fluctuations in effort preferences. This procedure also has the benefit of allowing us to directly model the effect of interest: the influence of low and high effort fatiguing exertions on changes in the valuation and decision processes, at the population level. Notably, each of the low/high effort sections in the Modified Fatigue Choice Phase of the control experiment has fewer trials (50/60/60) than the main experiment (170) which makes it beneficial to use such a Hierarchical Bayesian approach, since it leverages all the choice data and maximizes the possibility of identifying choice 'signals' of interest. Importantly, because of the limited set of choices in each fatiguing section, this methodology allows us to estimate parameters in sparsely sampled regions by modelling intrinsic structure within and between participants.

The underlying model used to fit subjectivity parameters (ρ and τ) from the choice data was structured similarly to a general linear mixed effects random intercept model, allowing for participant-specific intercepts but estimating population level

effects of the experimental section. Parameters from the Baseline Choice Phase and first 10U section (10U,1) of the Modified Fatigue Choice Phase are estimated in the following form:

$$\begin{aligned} \rho_i^S &= A * \Phi(r_i^S) \\ \tau_i^S &= B * \Phi(t_i^S) \end{aligned} \quad \begin{bmatrix} r_i^{Bas} \\ t_i^{Bas} \\ r_i^{10U,1} \\ t_i^{10U,1} \end{bmatrix} \sim \mathcal{N}(M, \Sigma)$$

In this formulation, both ρ and τ for participant (i) and experimental section (S) (Baseline, first low effort section (10U,1)), are represented and estimated as a participant-specific parameter (r and t), drawn from parameter-specific normal distributions with estimated means (M) and covariance matrix (Σ), which are estimated population-level hyper-parameters. Φ is the unit Gaussian cumulative distribution function bounding the supported parameters to (0,1), where A and B reflect imposed bounding constraints on the resulting estimates⁴³.

Changes in effort subjectivity parameters for the other two sections of the Modified Fatigue Choice Phase are modeled with a population-level section-specific (60U and 10U,2) shared offset (δ), which takes the following form:

$$\begin{aligned} \rho_i^S &= A * \Phi(r_i^{10U,1} + \delta_r^S) \\ \tau_i^S &= B * \Phi(t_i^{10U,1} + \delta_t^S) \end{aligned}$$

The model then generates posterior estimates of the population's distributions for M , Σ , δ^{60U} , and $\delta^{10U,2}$, which can be transformed into parameter-space using the above expressions. The probability that a participant chooses the sure option for the k^{th} trial is given by the softmax function (in which RV_{sure} is now contingent upon participant and exertion section-specific parameters, drawn from the population distribution):

$$P_k(RV_{sure}(G, S)) = 1/[1 + \exp(-\tau RV_{sure}(G, S))]$$

This parameter estimation procedure was implemented using Monte-Carlo Markov Chain sampling methods provided by Stan version 2.19⁴⁴ and implementing a similar methodology as described by the hBayesDM package⁴³. Standard hierarchical Bayesian methods were used, with constraints on the fit parameters of $\rho \in [0, 10]$ and $\tau \in [0, 6]$, and weakly informative distributions were chosen for the parameter priors in order to facilitate model convergence⁴⁵.

EMG Analysis

In order to compare EMG signals from homogenous exertion profiles, only successful exertion trials (spending at least 2.67 seconds of the exertion time within ± 5 effort levels of the target) were considered. We considered the mean frequency of the power spectrum of the sEMG signal as a physiological measure of fatigue, which has been well documented in numerous EMG studies investigating fatigue³⁶⁻³⁸. Mean frequency measures for each trial were computed via the MATLAB 'meanfreq' function examining a frequency interval of 10-500 Hz. To examine differences between the first 10U and second 10U sections of the Modified Fatigue Choice Phase, we averaged mean frequency measures across all successful trials in each section within each participant, and compared these values between the two sections (Figure 6d).

Regarding the reviewer's specific points listed above, we chose not to perform a paradigm in which participants performed a Baseline Choice Phase prior to a Fatigue Choice Phase, another Baseline Choice Phase a few hours or days after the Fatigue Choice Phase, to compare choice preferences between these Baseline Phases (Point a). We were concerned that such a paradigm would not allow for a controlled fatigue state of participants between the two Baseline Choice Phases. For example, if participants were tested during different times of the day, or following a particularly sleepless night, their state of fatigue could be quite different, even though they were supposed to be in a 'baseline' state. Instead, we designed the aforementioned control paradigm in which we fatigued participants with different levels of exertion, and acquired self-report measures of fatigue (Point c), and muscular activity recordings. This allowed us to directly test if changes in choice behavior are modulated by levels of exertion, and if these changes are related to ratings of fatigue and muscular signatures of fatigue. As described above, we found exactly such a modulation of fatigue ratings, muscle activity, and choice behavior by the fatiguing exertions in our paradigm.

Regarding Point b, determining if there was a shift in choice behavior as participants became fatigued in the main experiment, we divided the 170 trial choice set in the Fatigue Choice Phase into the first and last 85 trials (divided into equal halves, early and late) and found a significant decrease in the gamble acceptance rate between early and late phase choices reported (results of a one-tailed, one-sample t-test: $t_{19} = 1.80$, $p = 0.04$). This shift is consistent with participants making more risk-averse effort choices as fatigue increases throughout the course of the Fatigue Choice Phase. Notably, our new control experiment directly tested how choice preferences shifted with varying levels of fatiguing exertion and showed that participants became more risk-averse for effort (increase in ρ parameter) between the first 10U exertion section and the 60U exertion section, as well as between the first and second 10U exertion sections, and these changes were coincident with increases in self-reported ratings of fatigue (Figure 6c, e). In this way, our control experiment directly shows how increased fatigue modulates choice behavior.

Overall, the results of our control show that the changes in choice behavior, resulting from our exertion paradigm, are the consequence of experimentally induced physical fatigue.

2.2) Learning or fatigue: In Figure 2b, the authors showed the decrease in the mean exertion and interpreted it as an index of fatigue. However, as the criteria for each block is 75% of failures and a minimum number of trials, participants can easily and gradually understand that they can adapt and finish each block with lower force, which does not necessarily support the authors' interpretation. This may be reflected in a complicated plot (except upper right two participants) in Fig. 2e. Therefore, some physiological measurements of fatigue or at least a questionnaire data about fatigue would be necessary (related to 1 (c)).

The reviewer raises a good point that the failure criteria in the Fatigue Choice Phase, of the main experiment, could lead people to “learn to fail” in order to minimize exertions and advance through the experiment. To address this limitation of the main experiment, we developed a more stringent success criteria for the exertion blocks of the control experiment, and tested how this influenced choice behavior. In the new control experiment, we required that participants perform five successful exertions in each block before they progressed to choices, and if the total number of failed exertions exceeded a threshold throughout the experiment (20 trials), the experiment was terminated. Though participants did not know how many failed attempts were tolerated, they were informed that it was in their best interest to try and succeed on each exertion attempt, and if they performed an excess of unsuccessful trials, the experiment would end. Thus, in the control task, participants could not “learn to fail” in order to avoid performing the high-effort exertions. Furthermore, those who were unable to successfully participate ($n = 2$) were not included in the final analysis.

As discussed previously, we observed a fatigue-induced change in choice behavior with this paradigm. We also acquired self-report ratings of fatigue and found that participants reported increased ratings following high exertions (Figure 6c). Moreover, we acquired measures of participants' muscle activations with EMG to examine if physiological measure of fatigue resulted from this exertion paradigm. We found that the mean frequency of the power spectral density of the EMG signal between the first and second 10U exertion sections decreased, despite our participants exerting the same level of effort (Figure 6d). Such a decrease in the mean frequency of the EMG signal (when controlling for the same level of exertion) is a hallmark of muscular fatigue and shows that our paradigm elicits physiological signatures of fatigue.

2.3) Neural correlates of effort value difference: there seems to be activity other than Insula and dACC. For instance, superior temporal area in Figure 3b. A table including all results and correlation analysis of the results with “fatigue” would be necessary.

We have added supplementary tables containing a summary of the whole-brain results for the contrasts shown in Figures 3a, 3b, 4a, and 4c. These data are outlined in Supplementary Tables 1, 2, 3, and 4.

2.4) Thresholds for GLM analysis are all different. The authors used $P < 0.05$ small volume corrected, in Fig. 3a and 3b, but $P < 0.05$ FWE corrected for Fig. 4a. These arbitrarily chosen criteria would change the results. For example, if $P < 0.05$ small volume corrected, is applied in Fig. 4a, more brain structures may be revealed in Fig. 4.

We have now displayed all of our whole-brain contrasts at $p < 0.001$ (yellow) and $p < 0.005$ with a 10 voxel extent threshold (red). Statistical inference was performed using small-volume correction within *a priori* regions of interest, independently identified from a database of previous studies of effort-based decision-making (Neurosynth.org). These are standard methods used in affective neuroimaging (Lieberman and Cunningham 2009).

We had initially reported Figure 4a as FWE-corrected across the whole brain to illustrate the strongest motor results that survived whole-brain correction. Using a standard threshold level of $p < 0.005$ does result in more activations throughout the brain in the contrast, which we have listed in supplementary tables. Our main results, however, still hold (Figure 4a).

Figure 4a. Exertion-induced changes in brain activity. Irrespective of effort value, activity in primary motor cortex (peak = [-38, -30, 52]) and PM (peak = [-36, -18, 54]) was reduced in the fatigue condition, relative to baseline, at the time of choice. These activations are significant small-volume corrected $p < 0.05$ within our *a priori* premotor ROI.

2.5) Decrease in p in fatigue condition: several participants exhibited decrease in the sensitivity to effort in Fig. 5c. How do you interpret this?

Participants could avoid exerting effort by exhibiting a bias towards choosing the certain option, to forgo the risk of having to exert large amounts of effort (risk averse for effort). Alternatively they could exhibit an increased bias towards the risky option with the hope of obtaining the 0 effort component of that prospect, and potentially not having to exert any effort (risk seeking for effort). Those individuals that exhibited a fatigue-induced decrease in sensitivity to effort were the ones who become more risk seeking when fatigued. We have now more carefully described the interpretation of fatigue-induced increases and decreases in sensitivity to effort, parameterized by subjectivity parameter ρ (page 10, paragraph 1).

Analyzing choice trials in the Fatigue Choice Phase, we found that exertion-induced fatigue resulted in participants exhibiting more risk averse choice behavior for effort compared to the Baseline Choice Phase; participants were less willing to take the chance of having to exert large amounts of effort, indicating an increased marginal sensitivity to effort cost (Figure 2c shows group-average cost functions for effort for the Baseline and Fatigue Choice Phases). This manifested as a significant increase in $\rho_{Fatigue}$ compared to $\rho_{Baseline}$ (Figure 2d; mean $\Delta\rho = 0.49$ (SD = 0.86), two-tailed paired-sample $t_{19} = 2.54$, $p = 0.02$), corresponding to an increase in marginal costs for effort while in a fatigued state. A model-free analysis of the change in choice behavior (proportion of risky options accepted) corroborated this finding of decreased risk-taking while fatigued, and illustrated that this result was not simply a byproduct of our model-based analysis (Supplementary Figure 2). We also examined how changes in choice behavior varied over the course of the Fatigue Choice Phase and found that the change in risk-taking behavior remained relatively constant over the course of the choice blocks (Supplementary Figure 3). It is important to mention that a subset of participants ($n = 5$) exhibited a decrease in $\rho_{Fatigue}$ compared to $\rho_{Baseline}$, corresponding to a decrease in the marginal cost of effort while in a fatigued state. Such behavioral change corresponds to these participants exhibiting an increased risk tolerance in order to avoid effort exertion – these individuals are willing to take more risks for the chance of obtaining the zero effort component of the risky prospect. Notably the temperature parameter τ , which represents the stochasticity of an individual's choices, was not significantly different between the Baseline and Fatigue Choice Phases ($\Delta\tau = -0.08$ (SD = 0.25), two-tailed paired-sample $t_{19} = -1.42$, $p = 0.17$). Additionally we re-estimated subjectivity parameter $\rho_{Fatigue}$ while holding $\tau_{Fatigue}$ constant and equal to its baseline value, and still found a significant increase in $\rho_{Fatigue}$ compared to $\rho_{Baseline}$ (two-tailed paired sample $t_{19} = 3.22$, $p = 0.005$). These results indicate that fatigue had an impact on subjective effort valuation that was separate from variability in choice behavior, between the two phases.

2.6) Time discount: as participant's actual cost comes at the end of experiment as randomly selected 10 trials, time discount may also affect behaviors. Isn't it necessary to consider this? At least discussion might be helpful to readers.

A number of decision-making studies have used similar paradigms, in which choice options are not realized immediately following choice, in order to study neural signals related to prospective valuation (Plassmann et al. 2007; Chib et al. 2012; Crockett et al. 2017; Chong et al. 2017; Hogan et al. 2018). These studies do not model temporal discounting effects because the choice options are randomized across participants and because participants are not aware of the number of remaining gambles, any effects of time-discounting are equally present for all trials, assumed as uniformly distributed across all options, and therefore do not cause a systematic temporal bias in decision-making.

REVIEWER 3

Major Comments:

3.1) Model: Please define parameters more clearly. If methods come at the end of the paper, the results should provide a quick summary of what parameters mean (perhaps even with relevant equations).

At the reviewer's request, we have now added a brief description of the effort utility model in the Results section (page 7, paragraph 3).

To characterize the subjectivity of participants' effort choices in the baseline/rested and fatigued states, we used a subjective cost function $V(x) = -(-x)^\rho$, where $x \leq 0$ and V is the subjective cost of an objective effort level x . ρ is a participant-specific parameter that characterizes how an individual subjectively represents the effort level x . In this formulation, ρ is flexible enough to capture increasing, decreasing, or constant marginal utility changes in subjective effort as absolute effort levels increase. The case where $\rho = 1$ indicates that a participant's subjective effort cost is proportional to absolute effort levels; $\rho < 1$ indicates decreasing sensitivity to changes in subjective effort cost as effort level increases; $\rho > 1$ indicates increasing sensitivity to changes in subjective effort cost as effort level increases. We extracted effort utility parameters ρ , separately for the Baseline and Fatigue Choice Phases to test how fatigue influences the subjective valuation of effort.

3.2) Model: Is the computational model statistically identified?

The use of the effort utility function $V(x) = -(-x)^\rho$ was motivated by previous work in which we found this formulation best described effort-based decision-making in a rested state (Hogan et al. 2018). Previous studies have proposed a variety of effort utility models, and we tested if these models better described fatigue-induced changes in subjective effort valuation. A formal model comparison showed that $V(x) = -(-x)^\rho$ provided the best description of participants' decisions across both the Baseline and Fatigue Choice Phases.

We have included a description of this model comparison in the results section, and a detailed description of the alternative models and the results of the model comparison in the Supplementary Information.

3.3) Neural results: Several of the neural regressors are unclearly described. For example, consider the statement "Each of these categorical regressors included..." This description of the regressors is too compact for me. Did you have box-cars for chosen and unchosen, and on top of that, regressors that were modulated by expected value of each (chosen and unchosen)? Please report this more explicitly and clearly, ideally with a short regressor count. Relatedly, in the chosen - unchosen comparison, what regressors are contrasted exactly?

We have now more thoroughly described our imaging analyses, including the general linear models and the contrasts created. We have also included a count of the regressors used (page 44, paragraph 3).

A general linear model (GLM) was used to estimate participant-specific (first-level), voxel-wise, statistical parametric maps (SPMs) from the fMRI data. The GLM included categorical box-car regressors beginning at the time of trial presentation and ending when a choice was indicated, for both the Baseline and Fatigue Choice Phase, for the chosen and unchosen effort options. Each of these categorical regressors included unorthogonalized parametric modulators corresponding to the objective value of the risky ('Flip') and sure effort options. Trials with missing responses were modelled as a separate nuisance regressor. The Fatigue Choice Phase included an additional nuisance regressor, modelled as a 4 second block, corresponding to exertion trials between choice blocks. Finally, regressors modelling the head motion as derived from the affine part of the realignment procedure were included in the model.

The regressors included in our imaging model were as follows:

1. Trials during the Baseline Choice Phase in which the sure option was chosen (*Box-car categorical regressor beginning at the time of choice presentation and ending at the time of response*)
 - a. Parametric modulator: Value of the sure, chosen option
 - b. Parametric modulator: Value of the risky, unchosen option
2. Trials during the Baseline Choice Phase in which the risky option was chosen (*Box-car categorical regressor beginning at the time of the choice presentation and ending at the time of response*)
 - a. Parametric modulator: Value of the risky, chosen option
 - b. Parametric modulator: Value of the sure, unchosen option
3. Trials during the Fatigue Choice Phase in which the sure option was chosen (*Box-car categorical regressor beginning at the time of choice presentation and ending at the time of response*)
 - a. Parametric modulator: Value of the sure, chosen option
 - b. Parametric modulator: Value of the risky, unchosen option
4. Trials during the Fatigue Choice Phase in which the risky option was chosen (*Box-car categorical regressor beginning at the time of choice presentation and ending at the time of response*)
 - a. Parametric modulator: Value of the risky, chosen option
 - b. Parametric modulator: Value of the sure, unchosen option
5. Exertion trials during the exertion block (*Box-car categorical regressor beginning at the time of exertion trial presentation and lasting 4 s*)
 - a. Parametric modulator: Mean exertion (in terms of effort level) of the trial

b. Parametric modulator: Exertion trial number

6. Trials in which no choice was made in the allotted time (i.e., missed trials)
7. Regressors modeling the head motion as derived from the affine part of the realignment procedure were included in the model.

With these first-level models we created group models (second-level) to test brain areas that were generally sensitive to effort value. This was done by creating contrasts with the aforementioned parametric modulators for chosen and unchosen effort values, at the time of choice (i.e., difference between the value of the chosen and unchosen options, across both the Baseline and Fatigue Phases). To test for areas of the brain sensitive to decision values for effort, irrespective of fatigue state, we created a contrast that captured the difference between chosen and unchosen effort. This contrast was created by subtracting the parametric modulator for the unchosen risky and sure options (1.b, 2.b, 3.b, 4.b) from the chosen risky and sure options (1.a, 2.a, 3.a, 4.a). We also tested for regions of the brain in which decision value was sensitive to changes in bodily state induced by fatigue, by taking the difference between the value of the chosen and unchosen options, between the Fatigue and Baseline Choice Phases ($\{ [3.a + 4.a] - [3.b + 4.b] \} - \{ [1.a + 2.a] - [1.b + 2.b] \}$). Additionally, we tested for changes in regions of the brain that were more generally sensitive to changes in bodily state induced by fatigue by examining the difference between the Baseline and Fatigue Choice conditions, regardless of the effort values in question.

3.4) Neural results: I also find the neural regressors not clearly motivated. Why is chosen vs unchosen the relevant variable? Can the authors use trial-by-trial model-derived regressors and regress them against BOLD signal? This would make the connection between model and data stronger than currently. Relatedly, could the difference between chosen and unchosen simply reflect a difficulty effect?

When studying value-based decision-making it is common to represent the decision value (i.e., the value that drives choice) as the difference between the chosen and unchosen options (Bartra et al. 2013; O'Doherty 2014). The rationale being that the subtraction of the two options will inform a decision about which option will ultimately be chosen. Our current regressors model the trial-by-trial value of the chosen and unchosen options and the computed contrasts are the result of a subtraction between parametric modulators related to the effort value of the chosen and unchosen options on given trials.

To test a link between choice difficulty and decision value, we examined the trial-by-trial relationship between response time and the difference between chosen and unchosen value. We used response time as a model-free measure of choice difficulty – reaction time should be longest for the decisions are most difficult, and shorter when the choice is easier. If our measure of decision value is reflective of choice difficulty, we would expect a significant relationship between response time and the difference between the value of the chosen and unchosen options. When removing trials whose response time was beyond two standard deviations of the within-

participant mean, we failed to observe a significant correlation between decision value and choice difficulty (indexed by $\log(\text{response time})$) over the course of both Choice Phases (Wilcoxon signed-rank test on correlation coefficients: $z = 1.64$, $p = 0.10$). This suggests that decision value is not merely a reflection of choice difficulty.

3.5) Several of the neural results are not properly controlled for multiple comparisons.

Our whole-brain contrasts are displayed at $p < 0.001$ (yellow) and $p < 0.005$ with a 10 voxel extent threshold (red). Statistical inference was performed using small-volume correction, family-wise error corrected, within *a priori* regions of interest, independently identified from a database of previous studies of effort-based decision-making (Neurosynth.org). These are standard methods used in affective neuroimaging (Lieberman and Cunningham 2009). We have clarified this in the methods section (page 47, paragraph 2).

There is a degree of heterogeneity in the precise locations of the brain activations reported in previous studies of effort-based decision-making. However, these studies consistently implicate regions of dorsal anterior cingulate cortex and bilateral insula in effort valuation. With this in mind we analyzed brain signals related to chosen effort value within independent regions of interest (ROIs) taken at peak coordinates from Neurosynth.org⁴⁶ when using the term “effort”: rInsula MNI coordinates (x, y, z) = [36, 24, 0]; lInsula MNI coordinates (x, y, z) = [-36, 24, 0]; anterior cingulate cortex (x, y, z) = [0, 14, 48]. To analyze motor signals related to fatigue we used coordinates for premotor cortex reported in an independent study of fatiguing physical grip exertion (dorsal premotor cortex: (x, y, z) = [-36, -14, 64])¹³. Whole-brain contrasts for all figures are displayed at $p < 0.001$ (in yellow) and $p < 0.005$ with a 10-voxel extent threshold (in red), and statistical inference was performed within the SPM framework using small-volume correction, family-wise error corrected within these independently identified ROIs. These are standard methods used in affective neuroimaging⁴⁷.

3.6) Rho remains the same in a control condition where there is no fatigue induction. This is an important control condition, but there should be an interaction to make this point statistically valid (time*condition).

We have now included a formal statistical test that compares the change in choice behavior between the control condition, with no fatigue induction, and the main experimental group (Supplementary Information, Supplementary Figure 5).

Supplementary Figure 5. Comparison of choice similarity metrics (see Supplementary Methods for details) between control experiment 1 and the primary fatigue experiment, averaged across choices. Relative to the Baseline Choice Phase, more positive values indicate more risk-seeking behavior in the second choice phase, whereas negative values indicate more risk-averse behavior. Participants' choice behavior in the control experiment was not significantly different comparing the first and second choice phases (two-tailed one-sample $t_8 = 1.40$, $p = 0.20$), whereas choice behavior between the Baseline and Fatigue Choice Phase was significantly different (two-tailed one-sample $t_{19} = -2.59$, $p = 0.02$), consistent with our observed fatigue-induced increase in effort subjectivity parameter ρ . The results of a two-tailed two-sample t-test comparing these choice similarity metrics, between experimental groups indicates that these effects are significantly different ($t_{27} = 2.38$, $p = 0.02$), supporting the idea that the observed change in choice behavior in our main experiment is not merely an effect of choice exposure alone.

3.7) I don't get the control analysis for apathy— why would apathy but not fatigue be reflected in response times?

We reasoned that if individuals were apathetic towards the exertion task, this would manifest in their response times – apathy would present as an increase in response times to exert over the course of exertion trials, indicative of a lack of motivation in performing the task. However, our supplementary analysis showed that participants' response times remained relatively constant across exertion blocks.

Reviewer 1 also mentioned that this analysis could lead to confusion for the overall interpretation of our results, and since it is not a main facet of our paper, we have removed this analysis at their suggestion.

3.8) The correlation in 2E seems entirely driven by 2 outliers.

We utilized a two-step exclusion criteria in our main analyses to ensure that we analyzed data from participants that (1) developed strong effort associations and those that (2) were making consistent choices. To be sure that we analyzed data from participants who developed salient associations between units of effort and applied effort, we excluded participants ($n = 5$) whose r-squared value between reported and actual effort during Recall was less than 0.5. In the remaining participants, we excluded those in which the subjectivity parameter ρ , obtained from either Choice Phase (Baseline or Fatigue) was beyond two standard deviations of the populations mean for that phase or whose temperature parameters (τ) were near zero, indicative of random choice ($n = 5$). This second exclusion criteria ensured that the remaining participants were making consistent choices.

Given our conservative exclusion criteria, we did not feel justified in excluding participants from the analysis in Figure 2e. To account for the possibility of these individuals skewing this correlation we utilized a robust regression to down-weight their contribution. We still find a significant correlation between fatigue-induced changes in effort preferences and exertion decay parameters (standardized robust regression coefficient: $r = 0.87, p < 0.001$).

Minor Comments:

3.9) The data in Figure 4f are surprisingly linear; is this a linear model fit?

The data in Figure 4f is the result of a linear regression fit onto the data from the moderation analysis. This plot is meant to illustrate the effects of the moderation analysis. We have now updated the figure caption to clarify how this plot was created, and that the lines shown are linear fits, and though they are reflective of the data, are not raw data.

Figure 4f, Illustrative plot of the moderating influence of PM deactivation on the relationship between fatigue-induced reduction in motor performance and increases in the subjective cost of effort. The lines represent linear regressions between mean exertion decay and fatigue-induced changes in ρ , separated by median split PM activity. The less PM deactivation between the Baseline and Fatigue Phases, the stronger the relationship between the fatigue-induced decay in motor performance and fatigue-induced increases in the subjective cost of effort.

Reviewers' comments:

Reviewer #1 (Remarks to the Author):

The authors have thoroughly addressed all my concerns. Their manuscript has been improved, with key clarifications, deletion of some weaker parts, inclusion of new analyses and even one additional experiment. The revised manuscript is an excellent piece of work that deserves publication. I still find the 'dyshomeostatic' interpretation quite convoluted, but this does not harm the interest of the experiments and results. Readers will make their own opinion.

Reviewer #2 (Remarks to the Author):

In the previous review comments, I listed the following as the first minimum requirement for revision.

1 The order of baseline choice phase and fatigue choice phase

After practice (Association phase + Recall phase), participants conducted baseline choice phase. Then, they were exposed to exertion trials and fatigue choice phase. Although the authors attributed behavioral changes in the fatigue choice phase to fatigue, I am not convinced of this conclusion. It is possible that participant's experience of exertion trials not fatigue changed his/her behavior. To rule out this possibility, the authors need to confirm at least the followings.

(a) It is necessary to ask the same participants to do the baseline choice task a few hours or days after fatigue choice condition and compare the two baseline choice outcomes.

(b) It is preferable to split fatigue choice data into two parts, and compare the former and latter behaviors. If behavioral change truly reflects fatigue, the authors should be able to observe the shift in logistic curves similar to Fig. 2c.

(c) It is desirable to collect subjective evaluation of fatigue to justify that participants felt fatigue.

Among these, the point 1(a) is critically important for distinguishing fatigue from experience of exertion because this procedure could potentially provide causal interpretation between behavioral change and fatigue, but not the experience of exertion.

However, what the authors demonstrated in the revised manuscript is only that fatigue and experienced exertion are tightly correlated. Of course, it should have been done in the previous submission, there remain two weakness of this study as they were. First, we see ambiguity between fatigue and experience of exertion as a cause of behavioral change. Second, this study falls into a category of pure "correlational study" which we see these days only in more specialized journals.

Reviewer #3 (Remarks to the Author):

I thank the authors for responding to my comments by extra analyses and clarifications.

Remaining issues are the following:

Regarding my comment 3.4: The authors report a behavioural null effect; and apparently, the effect requires some creative data transformation before a null effect is obtained (RTs > 2 SD removed). More importantly, it does not directly address my concern that the neural chosen - unchosen effect is a difficulty effect. If I see correctly, RT is not included as a PM in the neural GLM? If so, I suggest including that in the model, thus to exclude an effect of choice difficulty. If it is already included in the regression, I suggest simply adding a sentence stating that RT is included and therefore the contrast chosen - unchosen is controlled for difficulty.

Regarding my comment 3.5: It's possible that SVC is the standard in "affective neuro imaging", it's not generally accepted in neuro imaging. I would at least make clear that future studies should test whether this result holds with proper (ie, whole-brain) statistical control.

We thank the reviewers for their comments. We have now revised our manuscript to address these comments and we believe the manuscript is strengthened as a result. We detail each of the changes in a point-by-point reply:

REVIEWER 2

2.1) In the previous review comments, I listed the following as the first minimum requirement for revision:

The order of baseline choice phase and fatigue choice phase. After practice (Association phase + Recall phase), participants conducted baseline choice phase. Then, they were exposed to exertion trials and fatigue choice phase. Although the authors attributed behavioral changes in the fatigue choice phase to fatigue, I am not convinced of this conclusion. It is possible that participant's experience of exertion trials not fatigue change his/her behavior. To rule out this possibility, the authors need to confirm at least the following:

- a) It is necessary to ask the same participants to do the baseline choice task a few hours or days after fatigue choice condition and compare the two baseline choice outcomes.
- b) It is preferable to split fatigue choice data into two parts, and compare the former and latter behaviors. If behavioral change truly reflects fatigue, the authors should be able to observe a shift in logistic curves similar to Fig. 2c.
- c) It is desirable to collect subjective evaluation of fatigue to justify that participants felt fatigue.

Among these, point 1a is critically important for distinguishing fatigue from experience of exertion because this procedure could potentially provide causal interpretation between the behavioral change and fatigue, but not the experience of exertion.

However, what the authors demonstrated in the revised manuscript is only that fatigue and experienced exertion are tightly correlated. Of course, it should have been done in the previous submission, there remain two weaknesses of this study as they were. First, we see ambiguity between fatigue and experience of exertion as a cause of behavioral change. Second this study falls into a category of pure "correlational study" which we see these days only in more specialized journals.

Using our current data we were able to directly test if changes in effort preferences were causally related to the level of effortful exertion and associated physical fatigue, or if changes were simply the result of the experience of repeated exertions (regardless of the level exerted). To do this we compared choice preferences during the first 25 fatiguing exertions, between the main experimental group and the control experimental group. Critically, the only difference between these portions of the experiments was the level of effortful exertion. In the main experiment, participants exerted at an effort level of 80U, whereas in the control experiment participants exerted 10U. As expected, participants exerted significantly more effort in the high effort group

(main experiment group) compared to the low effort group (control experiment group). Despite performing the same number of exertion trials in each group, we found a significant difference between participants' choice behavior when comparing the high effort and low effort groups. Participants in the high effort group became more risk-averse for effort, while those in the low effort group did not exhibit a significant change in choice behavior. These results provide direct causal evidence that changes in choice preferences are specifically related to fatigue, induced through high intensity physical exertion, and are not simply a result of the experience of repeated exertions.

We have now referenced this analysis in the main text (page 22, paragraph 2):

We performed an additional analysis to directly test if changes in effort preferences were causally related to the level of effortful exertion and associated physical fatigue, or if changes were simply the result of the experience of repeated exertions (regardless of the level exerted). For this analysis we compared choice behavior during the first 25 fatiguing exertions between the main experimental group (first 3 exertion and choice blocks) and the control experimental group (first 5 exertion and choice blocks). Critically, the main difference between these portions of the experiments was the level of effortful exertion. In the main experiment participants exerted effort at a level of 80U, and in the control experiment participants exerted 10U. Despite performing the same number of exertion trials in each group (Supplementary Figure 7a), participants exerted significantly more effort in the high effort group (main experimental group) compared to the low effort group (control experimental group) ($t_{34} = 38.83$, $p < 0.001$; Supplementary Figure 7b). We found a significant change in participants' choice behavior when comparing the high effort and low effort groups ($t_{34} = 2.60$, $p = 0.013$; Supplementary Figure 7c). Participants in the high effort group became more risk averse for effort, while those in the low effort group did not exhibit a significant change in choice behavior. These results provide direct causal evidence that changes in choice preferences are specifically related to fatigue, induced through high intensity physical exertion, and are not simply a result of the experience of repeated exertion trials.

We have included the full results from this analysis in the Supplementary Information (Supplementary Figure 7).

Supplementary Figure 7.

Comparison of choice preferences during high and low exertion efforts, controlling for the number of exertion repetitions. We compared behavior during the first 25 fatiguing exertions between the main experimental group (first 3 exertion and choice blocks) and the control experimental group (first 5 exertion and choice blocks). In the main experiment participants exerted at a level of 80U, and in the control experiment participants exerted 10U.

a, Participants did not perform a significantly different number of exertion trials between these experimental epochs (two-tailed two-sample $t_{34} = 0.08$, $p = 0.94$). Error bars indicate SEM.

b, Participants exerted significantly more effort in the main experimental group compared to the control experimental group (two-tailed, two-sample t-test comparing the mean effort produced between groups: $t_{34} = 38.83$, $p < 0.001$). Error bars indicate SEM. *** $p < 0.001$.

c, Comparison of choice similarity metrics (see Supplementary Methods for details) between the main fatigue experiment and control experiment, averaged across the selected epoch. Relative to the Baseline Choice Phase, positive values indicate more risk-seeking behavior in the subsequent choice phase, whereas negative values indicate more risk-averse behavior. There was a significant difference between choice behavior in the main and control groups (two-tailed

two-sample t-test, $t_{34} = 2.60$, $p = 0.013$). Participants in the control experimental group did not significantly alter their choice behavior during 10U exertion (two-tailed one-sample $t_{16} = 1.61$, $p = 0.126$), while those in the main experimental group became significantly more risk-averse for effort during 80U exertions (two-tailed one-sample $t_{18} = 2.13$, $p = 0.047$). The results indicate that changes in choice preferences are not simply the result of the experience of repeated exertions, but instead are induced by fatigue resulting from high intensity exertion. Error bars indicate SEM. * $p < 0.05$.

As we mentioned in our previous responses, we chose not to perform a paradigm in which participants performed a Baseline Choice Phase prior to a Fatigue Choice Phase, and then another Baseline Choice Phase a few hours or days after the Fatigue Choice Phase, to compare choice preferences between these Baseline Phases (Point a). We were concerned that such a paradigm would not allow for a controlled fatigue state of participants between the two Baseline Choice Phases. For example, if participants were tested during different times of the day, or following a particularly sleepless night, their state of fatigue could be quite different, even though they were supposed to be in a 'baseline' state. Instead, we designed a control experiment in which we fatigued participants with different levels of exertion, and acquired self-report measures of fatigue (Point b), and muscle activity recordings. This allowed us to directly test if changes in choice behavior were modulated by levels of exertion, and if these changes were related to ratings of fatigue and muscular signatures of fatigue. We found exactly such a modulation of fatigue ratings, muscle activity, and choice behavior by the fatiguing exertions in our paradigm (Figure 6).

Regarding Point b, determining if there was a shift in choice behavior as participants became fatigued in the main experiment, we divided the 170 trial choice set in the Fatigue Choice Phase into the first and last 85 trials (divided into equal halves, early and late) and found a significant decrease in the gamble acceptance rate between early and late phase choices reported (results of a one-tailed one-sample t-test $t_{19} = 1.80$, $p = 0.04$). This shift is consistent with participants making more risk-averse effort choices as fatigue increases throughout the course of the Fatigue Choice Phase. Notably, our control experiment directly tested how choice preferences shifted with varying levels of fatiguing exertion and showed that participants became more risk-averse for effort (increase in ρ parameter) between the first 10U exertion section and the 60U exertion section, as well as between the first and second 10U exertion sections, and that these changes were coincident with increases in self-reported ratings of fatigue (Figure 6c, e). In this way, our control experiment directly shows how increased fatigue modulates choice behavior.

Overall, the results of our control experiment show that changes in choice behavior, resulting from our exertion paradigm, are the consequence of experimentally induced physical fatigue. Furthermore, the additional analysis we have included in this revision provides causal evidence that changes in choice behavior are the result of high intensity physical exertion and not simply a byproduct of the experience of repeated exertion trials.

REVIEWER 3

3.1) Regarding my comment 3.4: The authors report a behavioral null effect; and apparently, the effect requires some creative data transformation before a null effect is obtained (RTs > 2 SD removed). More importantly, it does not directly address my concern that the neural chosen-unchosen effect is a difficulty effect. If I see correctly, RT is not include as a PM in the neural GLM? If so, I suggest including that in the model, thus to exclude an effect of choice difficulty. If it is already included in the regression, I suggest simply adding a sentence stating that RT is included and therefore the contrast chosen-unchosen is controlled for difficulty.

As the reviewer notes, our initial model did not include RT as a regressor. We have now performed a control analysis in which we included RT as a nuisance regressor in our GLM and we found that the chosen-unchosen effort effect was preserved. We have now included a sentence in the methods section referencing this control analysis (page 48, paragraph 1).

In a separate model using the same structure, we also included log(response time) as a parametric modulator in order to validate that chosen-minus-unchosen effort value signals were unrelated to potential choice difficulty effects. ROI analysis of the chosen-minus-unchosen contrast between Fatigue and Baseline conditions indicated that the effort value effect was preserved.

3.2) Regarding my comment 3.5: It's possible that SVC is the standard in "affective neuroimaging", it's not generally accepted in neuroimaging. I would at least make clear that future studies should test whether this result holds with proper (i.e., whole-brain) statistical control.

At the Reviewer's request, an additional clarification about whole-brain statistical control has been added to the Methods (page 48, paragraph 2).

There is a degree of heterogeneity in the precise locations of the brain activations reported in previous studies of effort-based decision-making. However, these studies consistently implicate regions of dorsal anterior cingulate cortex and bilateral insula in effort valuation. With this in mind we analyzed brain signals related to chosen effort value within independent regions of interest (ROIs) taken at peak coordinates from Neurosynth.org⁴⁶ when using the term "effort": rInsula MNI coordinates (x, y, z) = [36, 24, 0]; lInsula MNI coordinates (x, y, z) = [-36, 24, 0]; anterior cingulate cortex (x, y, z) = [0, 14, 48]. To analyze motor signals related to fatigue we used coordinates for premotor cortex reported in an independent study of fatiguing physical grip exertion (dorsal premotor cortex: (x, y, z) = [-36, -14, 64])¹³. Whole-brain contrasts for all figures are displayed at $p < 0.001$ (in yellow) and $p < 0.005$ with a 10-voxel extent threshold (in red), and statistical inference was performed within the SPM framework using small-volume correction, family-wise error corrected within these independently identified ROIs. These are standard methods used in affective neuroimaging⁴⁷. Future

studies will be required to test whether these results hold with whole-brain statistical corrections.

REVIEWER COMMENTS

Reviewer #2 (Remarks to the Author):

First of all, I would like to thank the authors for the revision. I read the manuscript carefully again with my original concern about the potential link between the experience of high exertions (not the fatigue) and increased risk aversion parameters.

Figure 6c shows that subjects' fatigue rating decreased during the second 10 U exertions, although it still remained high. The corresponding change in the effort parameter (Fig. 6e) demonstrated that it was higher in the second 10 U exertions than in the 60 U exertions. These observations reinforced my concerns that experience of high exertions not fatigue accounts for high effort parameters.

In this revision, the authors conducted additional comparison of choices in the first 25 fatiguing exertions between the main experimental group (first 3 exertion and choice blocks) and the control experimental group (first 5 exertion and choice blocks). They found a significant change in participants' choice behavior when comparing the high effort and low effort groups. However, if I understand the procedures correctly, this new observation cannot rule out the possibility that experience of high exertions is the cause of high effort parameters.

Related to this, it is also desirable to show Mean frequency of EMG during 60 U exertions to infer subject's fatigue.

In summary, although I appreciate the authors' sincere effort made for the revision and do understand the costs for running additional measurements, I still feel it necessary to add a control similar to Figure 6 in which subjects' effort parameters are measured after they rest 25 minutes for instance, and their fatigue rating returns to baseline.

Although the following two issues are small compared with the above, it would also be informative to readers if the authors can clarify these.

1 The results in Figure 2d and 2e look to be heavily dependent on two subjects (top right in Fig. 2e). Does significance remain after removing these?

2 It is surprising that the number of repetitions in Figure 2a and Figure 6a stayed around 5 throughout blocks, consistent with a possibility that subjects understood the stopping criteria and adapted their behaviours. Did the exertions in each block decrease gradually?

Reviewer #3 (Remarks to the Author):

I think the authors have satisfactorily addressed my issues.

However, I think the very valid comment of reviewer 2 has not been addressed in a satisfactory manner.

In particular, I do not understand why the authors would not directly compare the rho values across main and control experiment? They report a similarity analysis instead, but it's not clear to me why this is the appropriate control (rather than just calculate the rho values, as in the main analysis). Maybe it's just a matter of reporting, or maybe it's critical, I'm not sure. But I think it needs clarification, at least.

Relatedly, if it holds up, I think this is sufficiently important to be placed (and the motivation of the analysis explained) in main text, rather than just in Supplementary Methods and Results. But this may be a matter of taste.

We thank the reviewers for their comments. Overall, we believe the findings from our main experiment, control experiment, and control analyses, illustrate that shifts in effort preferences are the result of physical fatigue, and are not simply the result of experience of exertion. We have presented data from self-report ratings of fatigue, and EMG measures of muscle fatigue, which show that fatiguing exertions elicit physical fatigue, and that this fatigue is concomitant with changes in effort preferences. We have revised the manuscript to more clearly convey these ideas. We detail our rationale and the changes to our manuscript in a point-by-point reply:

REVIEWER 2

Major Comments:

2.1) Figure 6c shows that subjects' fatigue rating decreased during the second 10U exertions, although it still remained high. The corresponding change in the effort parameter (Fig. 6e) demonstrated that it was higher in the second 10U exertions than in the 60U exertions. These observations reinforced my concerns that experience of high exertions not fatigue accounts for high effort parameters.

It is important to consider the full set of data collected in the control experiment (Figure 6). As Reviewer 2 suggested in the first round of reviews, we collected self-report ratings of fatigue to assess if participants were fatigued by exertions, and EMG data so that we could directly observe physiological measures of muscle fatigue. Consistent with participants feeling physically fatigued, participants' ratings of fatigue significantly increased with 60U exertions and these elevated ratings persisted in the second 10U exertion section (Figure 6c). Moreover, when comparing EMG signals between the early and late 10U exertion sections, we found a significant decrease in the mean frequency of the EMG signal in the second 10U exertion section, indicating that participants experienced muscle fatigue (Figure 6d). Importantly these markers of increased fatigue were coincident with significant increases in participants' effort subjectivity parameters (Figure 6e). Together these results show that 60U exertions fatigue participants, that this fatigue persists even when performing the second block of low effort exertions, and that fatigue is related to changes in the subjective value of effort.

We take exception to the reviewer's points listed above because they cite observations that are not statistically significant, to justify that our results are driven by the experience of exertion rather than fatigue. While participants' ratings of exertion appear to slightly decrease in the second 10U exertion section, relative to the final exertion block of the 60U section, the decreases between sections were not statistically significant (comparison of mean effort ratings between sections using a two-tailed paired-sample t-test: $t_{16} = 0.10$, $p = 0.92$). Therefore we are unable to say that ratings of fatigue significantly decreased in the second 10U exertion section, relative to the 60U section. Furthermore, the reviewer makes the claim that the change in effort parameters in the second 10U section are greater than the changes in the 60U section, however there was no significant difference between these changes (Figure 6e, the highest density intervals of these differences substantially overlap). Both of the observations the reviewer uses as arguments for

our results being driven by experience of exertion, and not fatigue, are based on effects that are not statistically significant.

2.2) In this revision, the authors conducted additional comparison of choices in the first 25 fatiguing exertions between the main experimental group (first 3 exertion and choice blocks) and the control experimental group (first 5 exertion and choice blocks). They found a significant change in participants' choice behavior when comparing the high effort and low effort groups. However, if I understand the procedures correctly, this new observation cannot rule out the possibility that experience of high exertions is the cause of high effort parameters.

From this comment, it seems that the reviewer might be concerned that the 80U condition does not actually elicit fatigue. The results of this analysis must be considered in the context of the control experiment. Our control experiment showed clear evidence that the repeated exertions resulted in significant increases in self-report ratings of fatigue, and a significant decrease in mean frequency of EMG (an index of muscle fatigue). It is important to mention that the control experiment's high exertion condition was 60U, while the main experiment's exertions were 80U. Given the control experiment result, it is logical to conclude that the 80U exertions will result in even higher levels of physical fatigue than shown in the control experiment. Notably, mean exertion levels in the exertion trials showed that participants were exerting effort throughout the exertion blocks (Figure 2b), suggesting that participants were performing fatiguing effort and not simply abstaining from exertion (see point 2.6 for further discussion). In summary, the comparison of exertions in the main and control experiments illustrates that when controlling for the number of exertions, the *level* of fatiguing exertion (not just the experience of exertion) influences preferences for effort.

The results from the control experiment also run contrary to the idea that experience of high exertions alone drives changes in effort preferences. In the control experiment participants exerted a lower amount of effort – 60U (roughly 48% of maximum exertion capacity). This level of exertion was not as high intensity as the 80U condition of the main experiment, and still led to heightened ratings of fatigue and muscle physiological signatures of fatigue. In spite of the 60U exertions being lower intensity than the 80U level in the main experiment, we observed shifts in effort preferences. Additionally, we found that subjective effort parameters during 60U exertions increased relative to the first 10U exertions, suggesting an increase in the subjective value of effort with intensity of exertion. These results show that changes in effort preferences are modulated by the level of exertion and are not simply the result of experience of particularly high exertions (e.g., 80U exertions).

2.3) Related to this, it is also desirable to show Mean frequency of EMG during 60U exertions to infer subject's fatigue.

Consistent with participants becoming fatigued during 60U exertions, in the control experiment, participants exhibited a significant decrease in mean EMG frequency over the course of repeated exertion trials. Moreover, there was a decrease in the mean frequency of EMG in the 60U section

compared to the 10U sections. Overall these EMG results are consistent with 60U exertions eliciting fatigue-induced changes in muscle physiology.

We have now included these results in the Supplementary Information (Supplementary Figure 7).

Supplementary Figure 7. Mean frequency of the power spectrum of EMGs on a trial-by-trial basis across sections of the Modified Fatigue Choice Phase. Decreases in the mean frequency of the power spectrum of the EMG signal are physiological reflections of muscle fatigue. A general linear mixed effects model revealed a significant decrease of mean frequency of EMGs over trials (within block) for the 60U exertion section (fixed effect $\beta_{trial} = -0.75$ (SE = 0.14), $p < 0.001$). Error bars indicate SEM.

2.4) In summary, although I appreciate the authors' sincere effort made for the revision and do understand the costs for running additional measurements, I still feel it necessary to add a control similar to Figure 6 in which subjects' effort parameters are measured after they rest 25 minutes for instance, and their fatigue rating returns to baseline.

Having participants exert effort, rest for a prolonged period of time, and check that fatigue ratings return to baseline, is not the best control to rule out that changes in effort preferences are the result of exposure to exertion. Such a paradigm cannot rule out that changes in effort preferences are the result of exposure to exertions during the exertion phase and a lack of exposure to exertion during the baseline and post-rest phases, or actually different levels of fatigue between phases.

A more rigorous control is to have participants exert different levels of effort and determine if effort preferences are altered by these different levels of exertion. Such a paradigm ensures that

exposure to exertion is controlled across conditions, the only difference being the level of fatigue these exertions impart. This is precisely the control that we presented in our last revision (Figure 7). We show that preferences are modulated by the level of exertion, and are not the result of generally performing exertions. Furthermore, our control experiment shows that our exertion paradigm induces self-reported fatigue, and physiological changes associated with muscle fatigue (Figure 6). Together, we believe that our results show that changes in effort preferences are the result of fatigue and are not simply a byproduct of exertion.

We have taken the reviewer's comments to heart and understand their concerns regarding experience of exertions and fatigue, and the dissociation between the two. With this in mind, we have now included a discussion of how experience of exertion and context might influence feelings of fatigue and how this idea might influence the interpretation of our results (page 26, paragraph 3).

Another important factor influencing valuation of effort is the context in which fatiguing physical exertions are experienced. In this study we showed that mere exposure to exertion was not enough to elicit changes in effort preferences, and when controlling for the number of exertions, participants' effort preferences were modulated by the magnitude of the fatiguing exertions. However, physically fatiguing tasks are often associated with contextual components that could modulate the subjective value of physical effort. For example, the experience of running a distance race involves exerting a great deal of physical effort, similar to training, however there is added cognitive effort associated with race strategy that could make race performance particularly effortful. There are any number of factors (e.g., boredom, task-framing, social context) that may influence how physical efforts are experienced, and these factors may inflate or diminish the subjective value of physical effort. Investigating the role of contextual experience on effort valuation and fatigue will be an important future direction in understanding how individuals generate a subjective value of effort.

Minor Comments:

2.5) The results in Figure 2d and 2e look to be heavily dependent on two subjects (top right in Fig. 2e). Does significance remain after removing these?

We utilized a two-step exclusion criteria in our main analyses to ensure that we analyzed data from participants that (1) developed strong effort associations and those that (2) were making consistent choices. To be sure that we analyzed data from participants who developed salient associations between units of effort and applied effort, we excluded participants ($n = 5$) whose r -squared value between reported and actual effort during Recall was less than 0.5. In the remaining participants, we excluded those in which the subjectivity parameter ρ , obtained from either Choice Phase (Baseline or Fatigue) was beyond two standard deviations of the populations mean for that phase or whose temperature parameters (τ) were near zero, indicative of random

choice ($n = 5$). This second exclusion criteria ensured that the remaining participants were making consistent choices.

The comparison between Baseline and Fatigue effort parameters in Figure 2d remains significant even after removing the 2 participants that the reviewer mentions ($t_{17} = 2.49$, $p = 0.023$). However, given our conservative exclusion criteria, we did not feel justified in excluding these participants from the analysis.

The correlation in Figure 2e does not remain significant after removing these participants. However, as we mentioned above, given our conservative exclusion criteria, we did not feel justified in excluding participants from this analysis. To account for the possibility of these individuals skewing this correlation we utilized a robust regression to down-weight their contribution. We still find a significant correlation between fatigue-induced changes in effort preferences and exertion decay parameters (standardized robust regression coefficient: $r = 0.87$, $p < 0.001$).

2.6) It is surprising that the number of repetitions in Figure 2a and Figure 6a stayed around 5 throughout blocks, consistent with a possibility that subjects understood the stopping criteria and adapted their behaviors. Did the exertions in each block decrease gradually?

Exertions in each block decreased gradually and we did not see any evidence of participants prematurely stopping exertion.

Figure 2a illustrates the average number of repetitions, across all participants, in each exertion block of the Fatigue Choice Phase in the main experiment. The first exertion block had an average of 14.95 repetitions, indicating that it took approximately 15 trials for participants to reach a level of fatigue that caused them to fail the success criteria (i.e., 75% failure). Subsequent exertion blocks lasted for approximately 5 repetitions on average, indicating that participants were fatigued and were not able to successfully exert for many repetitions. This discrete success criteria led to the step-like effect in the number of repetitions until failure.

Figure 2b illustrates the mean exertion force (in terms of effort level) across all participants, in each exertion block of the Fatigue Choice Phase of the main experiment. This is a more continuous metric of task performance than the average number of repetitions, and captures participants' mean exertion over all trials within each block. The figure shows that repeated exertions (over blocks) lead to a decrease in capacity for exertion. Moreover, participants' mean exertion decreased through subsequent exertion blocks, showing that repeated exertions lead to fatigue and a decrease in motor output. Notably, mean exertion over the blocks never diminished below 50U indicating that participants were always attempting to exert during the exertion blocks and not prematurely stopping.

Below, we include a figure illustrating the mean and standard error of the mean exertion forces of the final five exertion trials, in each exertion block of the Fatigue Choice Phase of the main experiment. The gradual decrease in mean exertion force across block is consistent with the

effect illustrated in Figure 2b – participants attempt to perform high level exertions, rather than adjusting their exertion behavior to meet the stopping criteria.

Figure: Mean exertion force of the last 5 exertion trials in each exertion block of the main experiment.

Figure 6a illustrates the average number of repetitions, across all participants, in each exertion block of the Modified Fatigue Choice Phase of the control experiment. The control experiment had a slightly different success criteria, during the exertion phase, which was meant to tightly control the number exertions participants performed. In the control experiment participants exerted in a block until they achieved 5 trials at the target exertion level. Therefore, if participants wanted to advance to the subsequent choice block, they needed to succeed 5 times. No adaptive behavior could alter this – the only way to advance was to comply with the instructions.

Figure 6b illustrates the mean exertion force across all participants, in each exertion block of the Fatigue Choice Phase of the control experiment. This figure shows that participants mean exertions remained at the target levels throughout exertion blocks, which demonstrates that participants were not prematurely stopping exertion in the control experiment.

Below, we show the last five trials in each exertion block of the control experiment. Participants' exertions remained constant across block, further suggesting that participants were not prematurely stopping exertion during the exertion blocks.

Figure: Mean exertion force of the last 5 exertion trials in each exertion block of the control experiment.

REVIEWER 3

Major Comments:

3.1) I think the authors have satisfactorily addressed my issues. However, I think the very valid comment of reviewer 2 has not been addressed in a satisfactory manner. In particular, I do not understand why the authors would not directly compare the rho values across main and control experiment? They report a similarity analysis instead, but it's not clear to me why this is the appropriate control (rather than just calculate the rho values, as in the main analysis). Maybe it's just a matter of reporting, or maybe it's critical, I'm not sure. But I think it needs clarification, at least.

We apologize for omitting an explanation of our use of a model-free measure for this control analysis. The control analysis involved significantly fewer trials (30-50 trials per section; potentially sampling different regions of effort value space) than were used in the full main experiment (240 trials) and the other control experiment (240 trials), and it is difficult to obtain reliable parameter estimates from choice data with such few samples. To obtain useful parameter estimates it is best to use on the order of 100 choice trials, which sample the full space of effort options. It is important to note that the data used in this control analysis was not specifically designed with this exertion control in mind, and this is why such a limited number of choice trials were present. With this in mind, we used the model-free choice similarity measure that we had used elsewhere in the manuscript. The results from this analysis (increased risk-aversion for effort in the high effort/fatigue condition, compared to the low effort/fatigue condition) align with our previous findings from the model-based and model-free analyses in the main and control experiments. We have now clarified our rationale for using this model-free choice measure, and not model-based measures of ρ (page 22, paragraph 2).

It is possible that changes in effort preference could be a byproduct of the experience of exertion, and not actually related to physical fatigue *per se*. We performed an additional analysis to directly test if changes in effort preferences were causally related to the level of effortful exertion and associated physical fatigue, or if changes were simply the result of the experience of repeated exertions (regardless of the level exerted). For this analysis we compared choice behavior during the first 25 fatiguing exertions between the main experimental group (first 3 exertion and choice blocks) and the control experimental group (first 5 exertion and choice blocks). Critically, the main difference between these portions of the experiments was the level of effortful exertion. In the main experiment participants exerted effort at a level of 80U, and in the control experiment participants exerted 10U. Despite performing the same number of exertion trials in each group (Figure 7a), participants exerted significantly more effort in the high effort group (main experimental group) compared to the low effort group (control experimental group) ($t_{34} = 38.83$, $p < 0.001$; Figure 7b). It is important to note that the choice trials used in this analysis were not specifically designed with this exertion control in mind, and there were fewer trials available

for this control analysis (30-50 trials) than in the full main and control experiments. The limited number of trials make it difficult to obtain reliable parameter estimates as in the previous analyses of choice behavior. Instead, we used a model-free choice similarity metric (see Methods for details) to compare participants' choices between the baseline state and during exertion. We found a significant change in participants' choice behavior when comparing the high effort and low effort groups ($t_{34} = 2.60$, $p = 0.013$; Figure 7c). Consistent with our previous results, participants in the high effort group became more risk averse for effort, while those in the low effort group did not exhibit a significant change in choice behavior. These results provide direct causal evidence that changes in choice preferences are specifically related to fatigue, induced through high intensity physical exertion, and are not simply a result of the experience of repeated exertion trials.

3.2) Relatedly, if it holds up, I think this is sufficiently important to be placed (and the motivation of the analysis explained) in main text, rather than just in Supplementary Methods and Results. But this may be a matter of taste.

At the reviewer's suggestion we have added a more detailed explanation of this analysis, and associated figures, to the main text of the paper (now Figure 7). We have also included a discussion of how experience of exertion might influence feelings of fatigue and how this idea might influence the interpretation of our results (page 26, paragraph 3).

Another important factor influencing valuation of effort is the context in which fatiguing physical exertions are experienced. In this study we showed that mere exposure to exertion was not enough to elicit changes in effort preferences, and when controlling for the number of exertions, participants' effort preferences were modulated by the magnitude of the fatiguing exertions. However, physically fatiguing tasks are often associated with contextual components that could modulate the subjective value of physical effort. For example, the experience of running a distance race involves exerting a great deal of physical effort, similar to training, however there is added cognitive effort associated with race strategy that could make race performance particularly effortful. There are any number of factors (e.g., boredom, task-framing, social context) that may influence how physical efforts are experienced, and these factors may inflate or diminish the subjective value of physical effort. Investigating the role of contextual experience on effort valuation and fatigue will be an important future direction in understanding how individuals generate a subjective value of effort.